# Bedfast and Floating Ice Dynamics of Thermokarst Lakes Using a Temporal Deep Learning Mapping Approach: Case Study of the Old Crow Flats, Yukon, Canada

Maria Shaposhnikova[1], Claude Duguay[1,2], and Pascale Roy-Léveillée[3]

[1]Department of Geography and Environmental Management, University of Waterloo, Waterloo, Ontario, Canada
[2]H2O Geomatics Inc., Waterloo, Ontario, Canada
[3]Département de géographie et Centre d'études nordiques, Université Laval, Québec, Québec, Canada

*Correspondence to*: Maria Shaposhnikova (mshaposh@uwaterloo.ca)

**Abstract**. In light of the recent climate warming, monitoring of lake ice in Arctic and sub-Arctic regions is becoming increasingly important. Many shallow arctic lakes and ponds of thermokarst origin freeze to bed in the winter months, maintaining the underlying permafrost in its frozen state. However, as air temperatures rise and precipitation increases, less lakes are expected to develop bedfast ice. In this work, we propose a novel temporal deep learning approach to lake ice regime mapping from synthetic aperture radar (SAR) and employ it to study lake ice dynamics in the Old Crow Flats (OCF), Yukon, Canada over the 1992/1993 to 2020/2021 period. We utilized a combination of Sentinel-1, ERS-1 and 2, and RADARSAT-1 to create an extensive annotated dataset of SAR time-series labeled as either bedfast ice, floating ice, or land, used to train a temporal convolutional neural network (TempCNN). The trained TempCNN, in turn, allowed to automatically map lake ice regimes. The classified maps aligned well with the available field measurements and ice thickness simulations obtained with a thermodynamic lake ice model. Reaching a mean overall classification accuracy of 95%, the TempCNN was determined to be suitable for automated lake ice regime classification. The fraction of bedfast ice in the OCF increased by 11% over the 29-year period of analysis. Findings suggest that the OCF lake ice dynamics is dominated by lake drainage events, brought on by thermokarst processes accelerated by climate warming, as well as fluctuations in water level and winter snowfall. Catastrophic drainage, and lowered water levels cause surface water area and lake depth to decrease and lake ice to often transition from floating to bedfast ice, while a reduction in snowfall allows for the growth of thicker ice. The proposed lake ice regime mapping approach allowed to assess the combined impacts of warming, drainage, and changing precipitation patterns on transitions between bedfast and floating ice regimes, which is crucial to understanding evolving permafrost dynamics beneath shallow lakes and drained basins in thermokarst lowlands such as OCF.

Keywords: Old Crow Flats, thermokarst lakes, ice regime, synthetic aperture radar, deep learning, temporal convolutional neural network

**1 Introduction**

Lake ice is a fundamental part of the freshwater processes in cold regions, and its sensitivity to air temperatures makes it a robust indicator of climate change (Brown and Duguay, 2010). Arctic and sub-Arctic regions underlain by permafrost, or perennially frozen ground, are rich in lakes that formed as a result of localized ground subsidence attributable to permafrost thaw, also known as thermokarst lakes (Bouchard et al., 2017). Many shallow arctic lakes and ponds of thermokarst origin freeze to bed in the winter months, allowing lake-bottom temperatures to drop below 0°C and frost to penetrate the lake bottom

sediment. Permafrost is sustained beneath the lake bottom where the freezing-degree-days at the ice-sediment interface are sufficient to counterbalance the thawing that takes place while lake-bottom temperatures are above 0°C (Roy-Léveillée and Burn, 2017). Where lake bottom conditions are too warm to sustain permafrost, for instance where ice does not reach the lake bottom or where the period of ice contact is brief, permafrost will degrade and a bulb of unfrozen ground or talik will develop and expand beneath the lake bottom. Such talik development contributes to positive feedbacks as it promotes lake deepening

via subsidence of the lake bottom (Roy-Léveillée and Burn, 2016), further reducing the occurrence of bedfast ice, and increases the ebullition of potent greenhouse gases such as methane from the thawing and decomposition of organic matter beneath the lake bottom (Arp et al., 2012; Engram et al., 2020). However, lake ice thinning and a subsequent decrease in the extent and duration of bedfast ice lakes has been noted by researchers investigating thermokarst lakes of Arctic Alaska (Engram et al., 2018; Surdu et al., 2014). Hence, monitoring and quantifying thermokarst lake ice dynamics is critical for understanding

changes in sub-lake permafrost stability and expected changes in methane ebullition patterns in thermokarst lowlands. Bedfast ice mapping, in particular, has a variety of other applications, including climate monitoring (Arp et al., 2012), permafrost studies (Arp et al., 2011), bathymetric mapping (Duguay and Lafleur, 2003; Kozlenko and Jeffries, 2000), overwintering fish habitat (Brown et al., 2010), and winter water withdrawal (Hirose et al., 2008; Jeffries et al., 1996).

The bedfast and floating ice regimes of Alaskan lakes have been studied extensively. For instance, a study by Surdu et al.

(2014) analyzed 402 lakes, near Barrow, North Slope of Alaska using ERS-1 and 2 synthetic aperture radar (SAR) imagery from 1991 to 2011. The study indicates a decrease of bedfast ice fraction from a maximum of 62% in 1992 to 26% in 2011. A study by Arp et al. (2012) reports significant variability of ice regime changes observed in inner and outer regions of Arctic Coastal Plain of northern Alaska (ACP). Analyzing SAR imagery between 2003-2011 and comparing it to radar-based ice maps from 1980 it was found that 16% of bedfast lakes shifted to floating ice regimes. However, while in the outer ACP only

three lakes shifted from being fully bedfast in 1980 to having only floating ice in the period between 2003 and 2011, in the inner ACP 27% of lakes transitioned to floating ice. Engram et al. (2018) analyzed a 25-year time-series (1992-2016) of C-band SAR images in seven regions of northern Alaska and Seward Peninsula. The authors note that due to high inter-annual variability in floating ice extent, no statistically significant trends could be observed. Nonetheless, one of the study areas, namely Fish Creek region on the inner ACP, exhibited strong trends towards floating ice regimes. Over the 25 years of analysis

an increase of 4.2 % per decade was observed in the areas covered by floating ice, and the number of floating-ice lakes increased by 1.5% per decade. Considering the variability observed between different study areas within northern Alaska, lake

ice trends in other thermokarst lake areas of the Northern Hemisphere characterized by different climate and types of underlying sediments could show different results. Hence, in this study, the Old Crow Flats (OCF), Yukon, Canada is selected as the focus area. No previous study to date has examined the bedfast and floating ice regimes of lakes in this region.

Owing to the vast number of lakes occupying permafrost regions, satellite remote sensing plays a key role in monitoring lake ice. The potential of active microwave remote sensing for bedfast ice mapping has been known since 1975 when Sellmann et al. (1975) noticed a characteristic dark and bright pattern of ice on shallow lakes of Alaskan Coastal Plain when observing them from X-band Side Looking Airborne Radar. The dielectric properties of water and ice display a high contrast in an electromagnetic window between 5-17 GHz making this range sensitive to the presence of liquid water under the ice (Gunn et

al., 2015a). As such, SAR active microwave remote sensors, including X (8-12 GHz)-, L (0.5-1.5 GHz)-, and C (4-8 GHz)-bands, are not only able to penetrate clouds and operate independently of solar illumination, but also benefit from distinct backscatter patterns for floating and bedfast ice. Floating ice in shallow lakes is generally characterized by a high backscatter; surface scattering from the ice-water interface is the largest contributor to backscatter throughout the ice season, followed by volume scattering in the surface ice layer and double bounce from tubular bubbles providing a smaller contribution to total

backscatter (Atwood et al., 2015; Gunn et al., 2018; Murfitt and Duguay, 2021). Bedfast ice, on the other hand, presents a dark SAR signature due to low dielectric contrast between the ice and the underlying sediment which results in signal transmission or absorption by lakebeds (Grunblatt and Atwood, 2014; Jeffries et al., 2005). Traditionally bedfast and floating ice has been mapped from spaceborne C-band SAR (e.g., Kozlenko and Jeffries, 2000; Duguay et al., 2002; Brown et al., 2010), with a more limited use of L-band (Engram et al., 2013) and X-band data (Antonova et al., 2016). A variety of approaches have been

proposed to distinguish between bedfast and floating ice based on SAR backscatter, namely threshold-based classification (Bartsch et al., 2017; Brown and Duguay, 2010; Duguay et al., 2002; Duguay and Wang, 2019a; Engram et al., 2018; Hirose et al., 2008; Kozlenko and Jeffries, 2000; Wakabayashi and Motohashi, 2018), supervised and unsupervised classification approaches (Grunblatt and Atwood, 2014; Pointner et al., 2019; Surdu et al., 2014), as well as one unique method based on data mining (Tsui et al., 2019). It is worth noting that not all bedfast mapping approaches rely directly on the SAR backscatter,

for instance, some sea ice studies identified bedfast ice using SAR interferometry (Dammann et al., 2018) and landfast ice using SAR image pairs (Makynen et al., 2020).

Thresholding is the most widely used method. For instance, Bartsch et al. (2017) using Envisat ASAR C-band imagery produced circumpolar bedfast ice maps for a single winter season based on a threshold function fitted to data collected from the Yamal Peninsula, Russia. The utilized threshold function accounted for incidence angle variability and was applied to two

million Arctic lakes. Engram et al. (2018) reported a 93% overall accuracy achieved through an interactive thresholding algorithm applied to a 25-year time-series of six generations of C-band SAR imagery. The developed algorithm identified a unique threshold for each scene avoiding errors resulting from variations in ice and weather conditions as well as alleviating the need for incidence angle normalization. Finally, Duguay and Wang (2019a) compared thresholding with correction for incidence angle effects to two unsupervised classification techniques (K-means and Iterative Region Growing Using

Semantics) and found that thresholding outperformed the other two algorithms, reaching an overall accuracy of 92.56%, and generalized the best to new geographical regions.

As to the remaining challenges, overestimation of bedfast ice in the middle of deeper lakes, such as Teshekpuk Lake, Alaska, is still observed (Duguay and Wang, 2019a). It has been hypothesized that the darker signatures in the deeper sections of lakes could be caused by such phenomena as cracks in the ice (Pointner et al., 2019), or local ice thinning or complete melt caused by methane ebullition (Engram et al., 2020; Pointner and Bartsch, 2020). Moreover, ice salinity and the presence of wet snow on the ice surface result in reduced backscatter intensities of floating ice (Duguay et al., 2002; Grunblatt and Atwood, 2014). With the aim of improving classification results for deeper lakes, Pointner et al. (2019) compared the threshold method to two novel methods based on pixel connectivity: flood-fill and watershed method. Both methods considered topography and the fact that ice grounding generally takes place in the shelf regions. Visual assessment suggested an improved performance, but no definitive conclusion was reached due to the lack of field measurements.

Although thresholding involves analyzing temporal evolution of backscatter, to the best of our knowledge, the only study that fully exploited the temporal progression is Tsui et al. (2019) who adopted a data mining approach called dynamic time warping (DTW). DTW compares backscatter time-series based on their shape. However, approaches that use temporal similarity measures, such as DTW, are very computationally costly as they require scanning the training set in its entirety in order to make a decision for every test instance (Pelletier et al., 2019).

With the aim of analyzing bedfast and floating lake ice dynamics of OCF over time, in this study we propose a comprehensive automatic classification framework that employs temporal deep learning. A temporal convolutional neural network (TempCNN) architecture proposed by Pelletier et al. (2019) is adopted and modified to fit lake ice classification from SAR. This approach allows to develop a non-linear framework able to automatically distinguish between ice regimes based on their temporal evolution. The deep learning offers a more universal approach that works for different SAR polarizations (HH and VV), and is not sensor-, spatial resolution-, or year-specific. In addition, in contrast to the existing methods, TempCNN does not require a lake mask, as it is able to create a three-class output: bedfast ice, floating ice, and land. The trained network is used to create high-quality ice regime maps of OCF and analyze lake ice dynamics from 1992/1993 to 2020/2021. Documenting transitions between bedfast and floating ice regimes in relation to climatic trends is crucial to understanding permafrost dynamics beneath shallow water in thermokarst plains such as OCF, with potential implications for methane emissions and the regional carbon balance.

The remainder of the paper is organized as follows: Sect. 2 describes the study area; Sect. 3 outlines the methodology used to select SAR imagery, process it, create an annotated dataset (Shaposhnikova et al., 2022), train and test a TempCNN model, create ice maps (Shaposhnikova et al., 2022), and perform ice dynamics analysis in OCF; Sect. 4 presents and discusses the results; finally, Sect. 5 summarizes the key findings including potential causes and implications of the observed dynamics, and suggestions for future research.

## 2 Study Area

While thermokarst lake ice regimes of northern Alaska, USA, Lena River Delta, Russia, and Hudson Bay, Canada, have been studied by multiple researchers (Antonova et al., 2016; Arp et al., 2011; Arp et al., 2012; Duguay and Wang, 2019a; Engram et al., 2018; Mommertz, 2019; Surdu et al., 2014; Wang et al., 2018), ice regimes of the Old Crow Flats, Yukon, Canada (OCF), remain largely unexplored. OCF is a wetland of international significance rich in lakes of thermokarst origin and surrounded by mountains (Fig. 1) (Lantz and Turner, 2015). The 6,000 $km^2$ area contains over 1,050 $km^2$ of flat-bottomed shallow lakes approximately 1.5 m in depth on average (Roy-Léveillée and Burn, 2010; Roy-Léveillée and Burn, 2015). OCF was submerged under glacial lake Old Crow during the Wisconsin stage (Irving and Cinq-Mars, 1974). The glacial lake drained catastrophically about 15,000 years ago leaving behind thick glaciolacustrine deposits and remnant lakes (Lauriol et al., 2009; Zazula et al., 2004). More lakes formed during the early Holocene through thermokarst processes (Ovenden, 1985).

The northern part of the lowland is characterized by polygonal tundra, while subarctic boreal forest is found in the south (Roy-Léveillée, 2014). According to tree-ring climate records, the last few decades have been characterized by warming unseen in any other period of the past 300 years (Porter and Pisaric, 2011). Moreover, members of the Vuntut Gwitch'in First Nation, whose lifestyle is largely sustained by OCF, note significant changes in air temperatures, precipitation, and ice regimes in the past decades (Wolfe et al., 2011). As a result, between 1951 and 2007, OCF has seen a decline in lake area of 60 $km^2$ mainly attributed to climate- and erosion-induced catastrophic lake drainage (38 large lake drainages described by Lantz and Turner, 2015), the rate of which increased five times in recent decades (Labrecque et al., 2009; Turner et al., 2010; Lantz and Turner, 2015). Despite the overall trend of decreasing water surface area brought on by catastrophic drainage events, most lakes are increasing in surface area and new ponds are forming due to rising air temperatures and increased precipitation, which bring on lake ice thinning, deepening of active layers, permafrost thaw, and ground subsidence (Labrecque et al., 2009). The climate of OCF is continental with cold winters (mean January temperature of -31.1°C) and warm summers (mean July temperature of 14.6°C) (Lantz and Turner, 2015). Although OCF is underlain by continuous ice-rich permafrost, unlike other regions such as Mackenzie River Delta, even shallow bedfast-ice areas within lakes can display mean lakebed temperatures above zero due to deep snow and high summer air temperatures (Roy-Léveillée, 2014). Lakes surrounded by boreal forest tend to accumulate more snow, and therefore, develop thinner ice, while tundra lakes are characterized by a thinner, denser snow which results in complete (shallow ponds) or partial ice grounding (Duguay et al., 2003; Roy-Léveillée et al., 2014). In light of this region's unique climate and sediment type, active thermokarst processes, climate-driven vegetation changes, such as shrubification of the tundra area (Wang et al., 2020), and a vast number of lakes which given considerable changes to their ice regime have the potential to significantly influence the underlaying permafrost and subsequently the climate of the area, the analysis carried out in this study make a valuable addition to the existing body of knowledge.

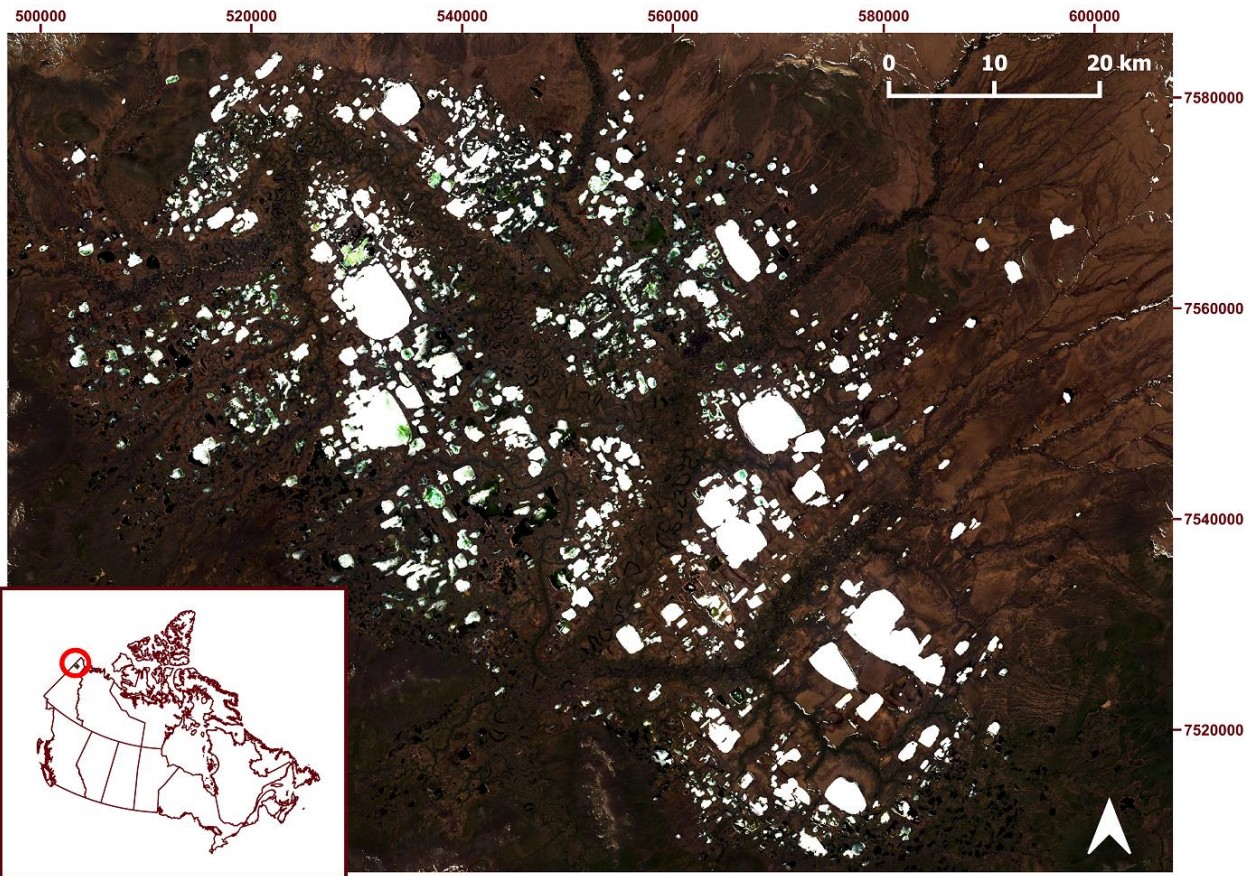

**Figure 1.** Old Crow Flats, Yukon, Canada. The background image is an RGB Landsat 8 of May 31, 2020, downloaded from USGS Earth Explorer (link: https://earthexplorer.usgs.gov/, accessed: July 4, 2021). Most lakes are still ice covered at this time of the year and appear white, open water surface of the river and smaller lakes, as well as some of the ice fringes appear black, tundra has a brownish shade, while areas of boreal forest appear dark green.

## 3 Data and Methods

### 3.1. SAR imagery

The dataset consists of imagery from four C-band SAR spaceborne platforms: Sentinel-1 A (S1) (VV polarization), ERS-1 and 2 (ERS1/2) (VV polarization), and RADARSAT-1 (R1) (HH polarization), which cover the time period between 1992/1993 to 2020/2021. Although ice reaches its maximum thickness in the OCF in late March or early April, the October to mid-March window was selected owing to the classification approach requirement for a consistent time-series end date. For all the dates beyond mid-March, one or more of the years had air temperatures above 0°C, which leads to surface melt

interfering with the discrimination between floating and bedfast ice (Duguay and Lafleur, 2003). Table 1 summarizes the dataset. In total, 18 years of data were chosen as they offered a minimum of two scenes for each month throughout the ice season (555 scenes).

**Table 1. Data used in the project (I.A. stands for Incidence Angle); number of scenes is indicated for each year, where full indicates # of full coverage scenes of OCF.**

| Instrument | Year | Polarization | Imaging Mode | Data Product | Pixel Size (m) | Spatial Resolution (m) | I.A |
|---|---|---|---|---|---|---|---|
| Sentinel - 1 | 2020/2021 (28 full); 2019/2020 (41 full); 2018/2019 (37 full); 2017/2018 (36 full); | VV | IW (Interferometric Wide Swath) IW covers an area of 250 km | L2 RTC product | 10 (30 after RTC) | 5x20 | 20-45 |
| ERS-1/2 | 2009/2010 (32; 7 full); 2008/2009 (26; 7 full); 1995/1996 (52; 9 full); 1994/1995 (34; 6 full); 1993/1994 (25; 4 full); 1992/1993 (30; 6 full) | VV (central frequency 5.6 cm) | STD (SAR Imaging Mode; swath width of 100 km) | L1 amplitude CEOS image | 12.5 | 26x6 | 23 at mid-swath |
| RADARSAT-1 | 2007/2008 (24 full); 2005/2006 (16 full); 2004/2005 (21 full); 2003/2004 (24 full); 2002/2003 (32 full); 2001/2002 (45 full); 2000/2001 (22; 16 full); 1999/2000 (30 full) | HH (central frequency 5.6 cm) | ScanSAR wide (SWB, swath width of 450 km) | L1 amplitude CEOS image | 50 | 100x100 | 20-46 |

The imagery was obtained from Alaska Satellite Facility (ASF). For S1, Radiometrically Terrain Corrected (RTC) level 2 products were downloaded. The RTC processing procedure calibrates the images removing topographic effects and corrects for geometric distortions ensuring precise geolocation. The process involves calibration, multi-looking to six looks, digital elevation model (DEM) matching, radiometric calibration to sigma nought amplitude, and speckle filtering. Images are subsequently terrain corrected and geocoded to UTM Zone 7N, datum WGS84. The pixel size of the RTC products is 30 m. The ERS1/2 and R1 data were available as level 1 products with a respective pixel size of 12.5 m and 50 m. They were processed using MapReady Remote Sensing Tool Kit available from ASF. MapReady allows to perform calibration to sigma nought, terrain correction, and geocoding. Further processing was done using Sentinel Application Platform (SNAP) available

from the European Space Agency (ESA). First, all the scenes were subset to the OCF extent. When working with SAR imagery it is essential to minimize speckle noise for successful analysis. Therefore, to match the RTC S1 products filtered using a 7x7 Lee Filter (the filter kernel covers approximately 44,100 m$^2$) with a dampening factor of 1 and 180 looks, ERS1/2 and R1 were speckle filtered using a 17x17 (45,156 m$^2$) and a 5x5 (62,500 m$^2$) Lee Filter, respectively. Adjusting the filter size allowed to account for the pixel size differences. Finally, the scenes for each year (October to mid-March) were co-registered to create stacks of SAR image time-series and converted to a decibel (dB) logarithmic scale traditionally used for C-band SAR analysis (e.g., Brown et al., 2010; Engram et al., 2018).

It is important to note that the four SAR platforms differ significantly in terms of pixel size, as well as spatial and temporal coverage. S1 provided full coverage of the OCF at least every five days between October and mid-March when data from both descending and ascending overpasses were used. Due to its relatively narrower swath size (100 km), ERS1/2 did not cover the OCF in their entirety and scenes were much sparser in time. The wide swath ScanSAR imaging mode of R1 allowed for full coverage of the OCF. However, the revisit time was variable from month to month and year to year.

Although using different SAR instruments and polarizations within the same classification algorithm presents its challenges, previous research has shown that such combination is suitable for the mapping of bedfast and floating ice regimes. For instance, Duguay and Wang (2019b) have developed a thresholding algorithm for Sentinel-1 adjusted for incidence angle and have demonstrated comparability of VV and HH polarized C-band SAR imagery for the purpose of classifying lake ice regimes. Engram et al. (2018) also proposed an interactive threshold classification method to analyze floating and bedfast lake ice regimes across Arctic Alaska using 25-year time-series (1992-2016) of C-band SAR images from different platforms (ERS1/2, RADARSAT-2, Envisat, and S1) with both HH and VV polarizations.

## 3.2. Annotated dataset creation

Deep learning algorithms require extensive annotated datasets. Therefore, a dataset consisting of 129,000 labeled backscatter time-series was created (Shaposhnikova et al., 2022). Labeling was done in SNAP by manually placing pins at locations identified as either floating ice, bedfast ice, or land through visual assessment of the ice regime/land on the last day of the time-series for a given season. This was done for each of the SAR image stacks (18 years). Due to variable temporal coverage, the dates of labeling ranged from March 4 to March 22. The labeling date was selected as close as possible to mid-March, and care was taken to ensure that the air temperature was below 0°C. Then, the backscatter values at the locations marked by each pin were extracted for each of the scenes in a SAR stack, creating time-series of labeled backscatter values for each year covering the October to mid-March period.

Labels were assigned based on the following three factors: 1) backscatter values, 2) value of the projected incidence angle of the SAR pulse (Duguay and Wang, 2019b), and 3) location of the pixel within the scene. Firstly, due to high dielectric contrast of water and ice, on a gray level scale floating ice appears bright, whereas bedfast ice is dark. Secondly, it has been noted in

previous investigations that the threshold between bedfast and floating ice becomes lower as the value of the projected incidence angle increases (Bartsch et al., 2017; Duguay and Wang, 2019a). Finally, shallower shore areas tend to become bedfast, while deeper middle portions maintain liquid water under the ice (Pointner et al., 2019). In addition, ponds are likely

to freeze to bed in their entirety, while bigger lakes display a combination of the two regimes. To identify land areas, early fall scenes (e.g., October) were used as a reference where water covered by thin ice appears dark as its mirror-like surface leads to specular reflection of the SAR signal away from the sensor, while the land is bright due to the roughness of its surface and vegetation volume scattering (Huang et al., 2018). Later in the season as the floating ice thickens its backscatter increases due to high dielectric discontinuity and roughness of the ice-water interface causing floating ice and land to have similar signatures.

In terms of spatial distribution, floating ice labeled pixels were spread out over each scene as evenly as possible. Labelled pixels with bedfast ice, on the other hand, being less prevalent were less spread out. However, it was ensured that areas in both the northern and the southern parts of the Old Crow Flats were included to incorporate both tundra and boreal forest environments. Resampling to a daily frequency and linear interpolation were applied to compensate for the temporal irregularity of the data ensuring that each of the backscatter time-series for each year of data had the same length (161 values)

and gearing it for the deep learning classification (Pelletier et al., 2019; Valero et al., 2016). Although the lake ice lifecycle is non-linear, previous studies have shown that more complex interpolation methods have little influence on classification accuracy (Pelletier et al., 2019, Valero et al., 2016). Linear interpolation was performed utilizing python programming language and the tools of pandas module. Interpolation was performed individually on every time-series (backscatter value of each pixel traced through time). As a result, we obtained SAR image stacks consisting of 161 full coverage scenes, which were

subsequently input into the TempCNN to perform classification. In addition to the proper SAR processing and speckle filtering, further quality control was implemented by filling any missing or Not a Number (NaN) values, especially common for ERS1/2 and scene fringes, as part of the temporal interpolation process. The final labeled time-series consisted of 161 time steps (i.e., one time step per day) covering the time period between October 4 and March 13.

Figure 2 compares interpolated time-series by class and by sensor. One can notice that the three classes are quite distinct. For

a clearer presentation, each class is represented by a mean and a standard deviation of a randomly selected sample of 100 time series. As such, averaging masks the initial drop in backscatter as thin ice forms on the wind-roughened water surface (Duguay and Lafleur, 2003). However, Fig. 2 demonstrates that while floating ice backscatter gradually increases throughout the ice season, the bedfast ice backscatter peaks around the months of December and January and then decreases as the ice becomes bedfast (Duguay and Lafleur, 2003). Land, on the other hand, shows little variability throughout the ice season and is

characterized by a narrow range of backscatter values. Despite slight differences between the three sensors, the VV polarized S1 and ERS1/2 and HH polarized R1 have enough intraclass similarity and interclass distinctions to allow for successful temporal classification.

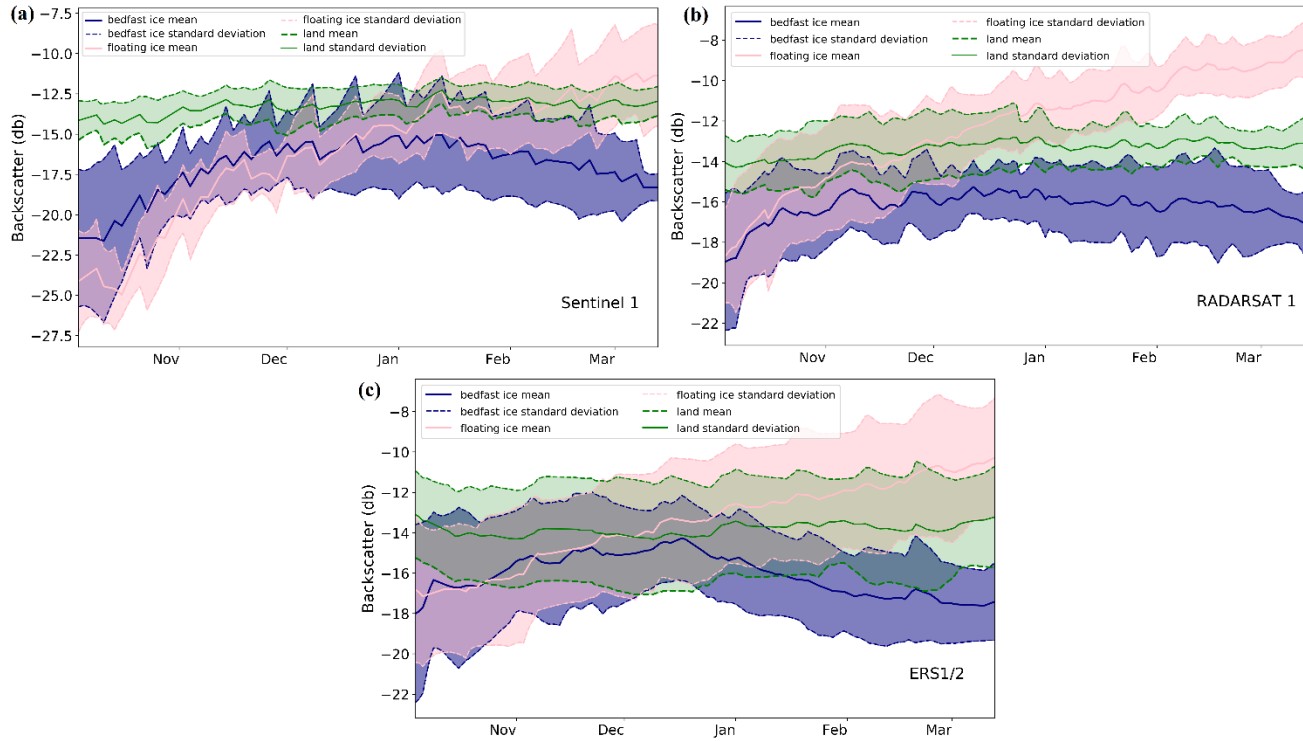

**Figure 2.** Comparison of the three classes by sensor: (a) Sentinel-1; (b) RADARSAT-1; (c) ERS-1/2. Each class is represented by a mean and a standard deviation of a sample of 100 randomly selected pixels per sensor. Means and standard deviations are identified by solid and dashed lines, respectively: pink – floating ice; dark blue – bedfast ice; green – land.

## 3.3. Temporal Convolutional Neural Network and Ice Regime Maps

### 3.3.1 TempCNN

In this work, the temporal dimension of ice backscatter evolution is employed by adapting a TempCNN proposed by Pelletier et al. (2019) for land cover classification from optical data to be used for lake ice classification from SAR. The study by Pelletier et al. (2019) shows that TempCNNs are more suitable for large scale studies due to their lower computational

complexity, higher classification accuracy, and shorter training time than recurrent neural networks that are traditionally used for sequential data (Minh et al., 2018; Ndikumana et al., 2018). A TempCNN learns from the data by applying different filters to the input time-series at the pixel level. A filter of a given size constitutes a 1-dimensional array of weights that slides across the time-series at the stride defined by the user; in each position the values of the filter are multiplied element-wise by the values of the time-series. The summed up resulting products become the part of the convolution output. The information

extracted by each filter depends on the values that constitute it. For instance, Fig. 3 demonstrates an input and an output for filters of size 5 known as the gradient filter and the lowpass filter that extract the information on increasing/decreasing gradient of the input time-series (Pelletier et al., 2019) and its trend, respectively. In a TempCNN, each input time-series undergoes a number of filters equal to the number of neurons (units) in a given convolutional layer. The best filters are learned by the network through the training process to allow for accurate mapping.

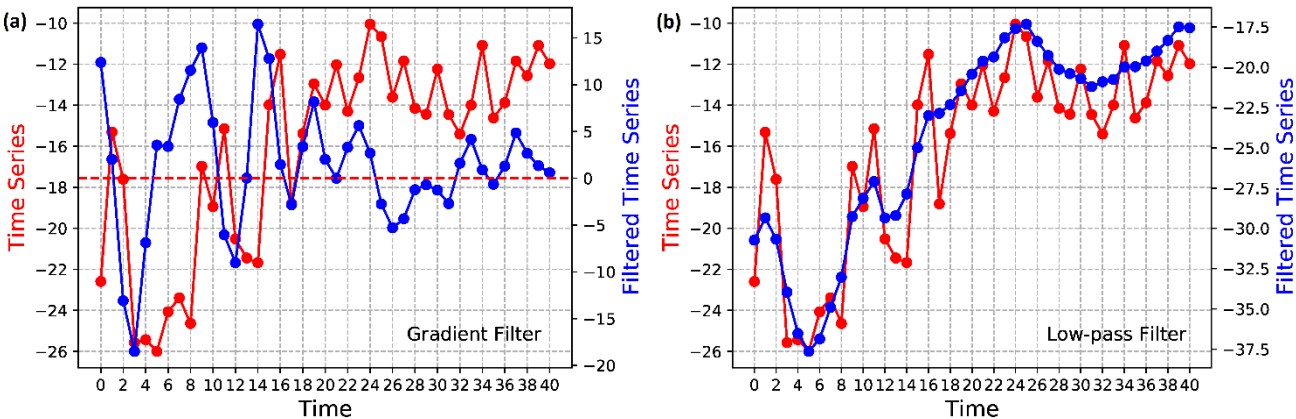

**Figure 3.** The two graphs illustrate application of 1-dimensional (1D) filters to time-series that can be used by convolutional layers of a TempCNN for extraction of temporal features. The red line represents original time-series, while the blue line denotes the filtered time-series: (a) a curve that resembles floating ice transformed by a gradient filter; the red dashed line indicates the origin of the filtered time series; the filtered series has positive values where the value of the original series is increasing, while the filtered series has negative values where the value of the original series is decreasing; (b) a curve resembling floating ice transformed by a low-pass filter.

### 3.3.2 TempCNN architecture

The TempCNN architecture used in this article is shown in Fig. 4. The network consists of three convolutional layers, each of which contains 64 different filters of size 5. Padding set to "same" in combination with the stride of the filter equal to 1 ensure that the input retains its size from one convolutional layer to the next. The three convolutional layers are followed by a fully connected layer with 256 units. This layer performs the final class assignment based on the final output of the convolution

process. Finally, the softmax layer converts the probabilities output by the fully connected layer into class probabilities that sum up to 1. Ultimately, this information can be either used to create probability maps or transformed using an argmax function that selects the highest probability and assigns a class to each pixel.

The network was trained using an activation function called Rectified Linear Unit (ReLU). Once the input has been transformed by a convolutional unit, it is subsequently transformed by an activation function that introduces non-linearity to the model. Overfitting is a common deep learning problem. Overfit models perform extremely well on the training data, but generalize poorly to unseen data. The TempCNN incorporates four techniques that ensure that the model is well-trained, but at the same time able to perform well on new data: batch normalization, early stopping (patience = 10 epochs) controlled by a validation set that constitutes 5% of the training set, the dropout technique (rate = 0.5), and L2-regularization (l2 = $1*10^{-6}$).

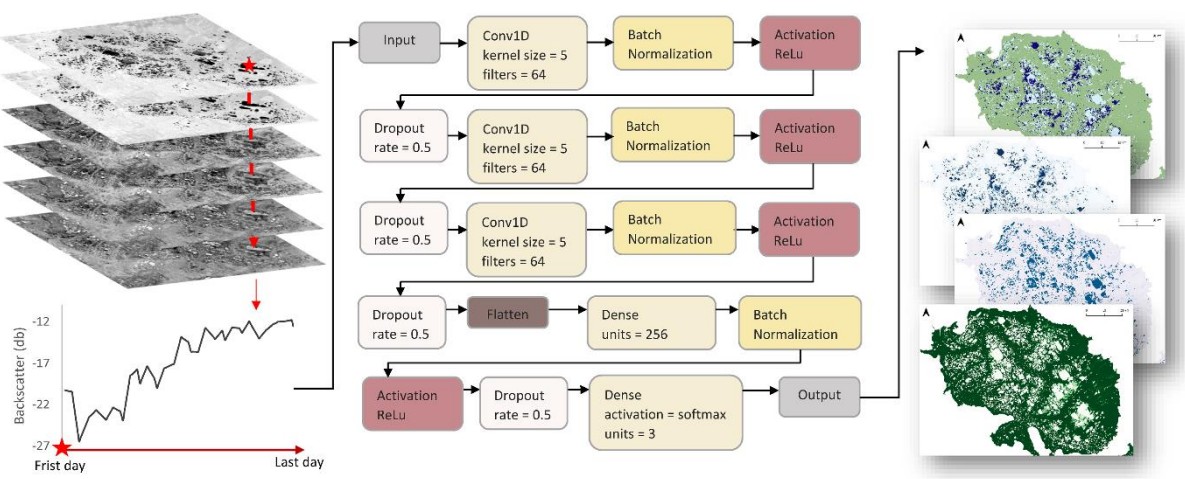

**Figure 4.** Structure of the trained TempCNN. The left portion represents an image stack that serves as an input to the neural network; the middle portion illustrates the composition of the TempCNN including layers, activation functions, and mechanisms for overcoming overfitting; the right portion represents the output created by the neural network that consists of a three-class (floating ice; bedfast ice; land) map, and three probability maps (one for each class).

### 3.3.3 TempCNN architecture selection

Out of 18 years of data, one year was reserved for final testing, while the remaining 17 years were used to identify the best model structure utilizing a cross-validation procedure. Cross-validation allowed to determine the optimal number of convolutional units (4, 8, 16, 32, 64, or 128) while keeping other elements of the structure constant (Fig. 4). TempCNN with each number of units was trained on 16 years of data and tested on one year of data 17 times each time leaving a different year out for testing. The overall accuracy summary of each of the six sets of experiments is shown in a box-plot form in Fig. 5. Based on the value of the median, interquartile range (IQR), length of the whiskers, and number of outliers, architecture with 64 convolutional units was identified as the best and was subsequently used for creating lake ice maps.

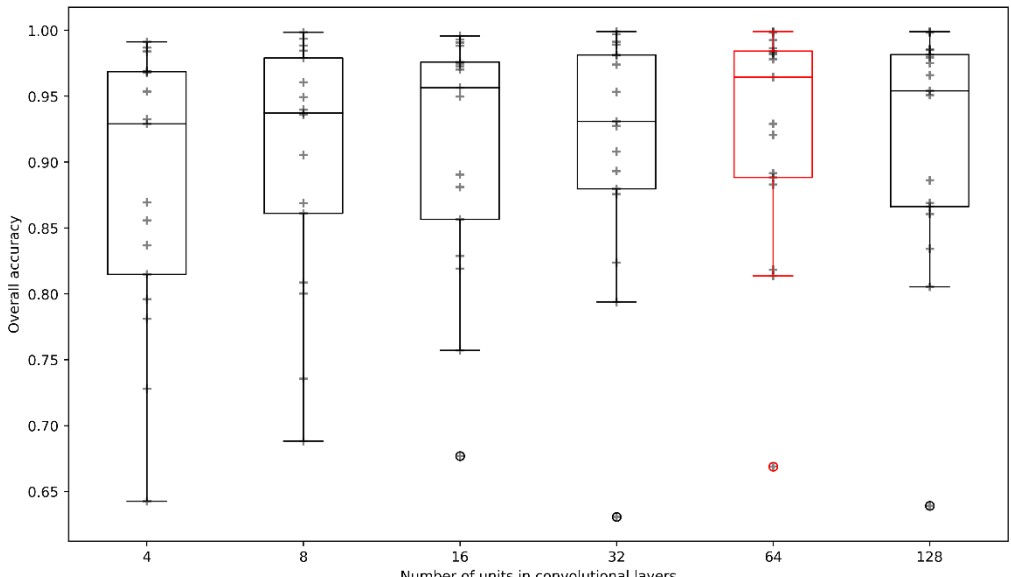

**Figure 5.** The graph illustrates temporal cross validation results using box-plots. Each box-plot contains 17 overall accuracy values and if read from left to right each box-plot corresponds to 4, 8, 16, 32, 64, and 128 convolutional units in each convolutional layer. Red highlights the best architecture with 64 convolutional units.

### 3.3.4 TempCNN training and testing

The classification accuracy of the selected TempCNN model was evaluated through 15 more experiments to account for the stochastic nature of the neural network training process (Pelletier et al., 2019). The training and testing were performed 15 times, consisting of three sets of five runs each split for training and testing differently. The first set randomly split the dataset into 80% for training and 20% for testing, this way incorporating data from all the years into both the training and the test set. The second set was trained on 15 years and tested on three years, where each test year was obtained by a different sensor. The third set was trained on 17 years of data and tested on the 18[th] year of S1, 2020/2021 reserved for validation. The model trained on points from all years of the SAR dataset was selected for the ice regime mapping. Although it is most likely worse at generalizing to unseen years and areas than other models, it produced better overall accuracy, and therefore is more likely to create high-quality maps for the purpose of OCF ice dynamics analysis.

### 3.4 Creation of ice regime maps using TempCNN

In order to transform SAR image stacks for each of the 18 years of data into lake ice regime maps using the trained TempCNN each stack had to be interpolated. Interpolation allowed to compensate for temporal resolution variability between different years such that each year's stack consisted of 161 scenes corresponding to a daily frequency from October 4 to March 13. Pixel-based linear interpolation was performed utilizing python programming language and the tools of pandas module.

Although the lake ice lifecycle is non-linear, previous studies have shown that more complex interpolation methods have little influence on classification accuracy (Pelletier et al., 2019; Valero et al., 2016). Once the SAR stacks for 18 years were interpolated and each consisted of 161 scenes, the trained TempCNN model was used to create ice regime classification maps consisting of three classes: floating ice, bedfast ice, and land (Shaposhnikova et al., 2022). Apart from outputting a classification map, TempCNN also provided probability maps for each class, where the value of each pixel corresponded to the probability of this pixel to belong to this specific class. For consistency, after classification all the maps were subset to the areas with an elevation below 330 m using a digital elevation model (DEM) granule included with the RTC S1 imagery from ASF. The DEM granule comes from the National Elevation Dataset at 2 arc seconds resolution (NED2) which is produced and distributed by USGS. This step allowed to exclude high elevation mountainous areas surrounding the OCF that appear very bright in SAR imagery and could be classified as floating ice.

### 3.5 Accuracy assessment

Apart from statistical classification based on the test set, accuracy of the ice regime maps was assessed using a set of 51 field observations. A set of bathymetry measurements collected in the OCF in late July of 2000, in addition to a small set of field measurements collected in taiga and tundra zones of the OCF in spring of 2009 and 2021, were available. Data collection included some or all of the following characteristics: ice thickness, ice regime, lake depth, temperature of the underlying sediment, and snow depth and density. Given the limited number of field measurements, lake ice thickness estimates were also obtained for comparison via simulations with the Canadian Lake Ice Model (CLIMo) (Duguay et al., 2003). CLIMo is a one-dimensional thermodynamic model designed for the simulation of lake ice formation, ice-growth and ice decay processes. The model has been shown to perform well for the simulation of ice dates and thickness on small (shallow) and large (deep) northern lakes (e.g., Brown and Duguay, 2011; Duguay et al., 2003; Gunn et al., 2015b; Jeffries et al., 2005a; Kheyrollah Pour et al., 2017; Menard et al., 2002; Surdu et al., 2014). In this study, CLIMo was forced with mean daily air temperature, wind speed, relative humidity, snow fall, and cloud cover produced from ERA5 (global atmospheric reanalysis data, produced using ECMWF model freely available through the Copernicus Climate Change Service). In the simulations, lake depth was specified as 2 m (small changes in depth do not impact ice thickness for the end of the season). Snow density was set to 175 kg m$^{-3}$ for lakes located in the taiga zone and 300 kg m$^{-3}$ for those in the tundra zone. The density values were chosen based on the combination of limited field measurements and range of typical values reported in the literature (Duguay et al., 2003; Jeffries et al., 2005).

Bathymetric measurements of July 2000 were matched with the corresponding TempCNN predicted classes for March 2000 based on geolocation and analyzed in the context of CLIMo simulated ice thickness for the same year. Specifically, if the depth of the data point (based on the bathymetric measurement) was shallower than the CLIMo simulated ice thickness for the corresponding vegetation type (taiga, tundra, or mixed – using Turner et al., 2014 OCF land cover classification) and the label created by TempCNN was "bedfast ice", the point was considered to be classified correctly. Analogously, if the depth of the

data point was greater than the CLIMo simulated ice thickness and the TempCNN label was "floating ice", the point was

considered to be classified correctly. Ice regime observations made in early April of 2009 and 2021 were also matched with the TempCNN classification output. In this case, both ice thickness and lake depth measurements were available. As such, if the lake depth was equal to the ice thickness, the point was considered to be bedfast, while if the lake depth exceeded the ice thickness measurement, the point was considered to be floating. The precision of the utilized field lake depth and ice depth measurements was 1- 2 cm. Finally, in order to assess the accuracy of all the classified ice regime maps, the ice dynamics over

the 29-year period of Husky Lake were analyzed in the context of CLIMo ice thickness simulations. Other factors, such as possible water level fluctuations, were also taken into account by visually analyzing Landsat optical imagery and looking at precipitation data provided by Environment and Climate Change Canada.

## 3.6 Comparison to thresholding

In order to benchmark the proposed method against commonly used techniques of lake ice regime classification, it was

compared to one of the most recent variations of the thresholding approach designed by Duguay and Wang (2019b) and applicable to S1 data acquired at HH and VV polarization as is illustrated in Fig. 6 below. This thresholding algorithm defines the backscatter threshold between floating and bedfast ice as a linear function of the local incidence angle. As such, lake ice regime of each lake pixel is determined in a two-step process: 1) a threshold value is calculated using the following equation: $f(\theta) = -0.257 * \theta - 6.933$; 2) if the backscatter value (VV or HH) of a specific lake ice pixel is greater than or equal to the

threshold it is classified as floating, if the value is less than the threshold it is classified as bedfast. Due to the fact that this approach is suitable only for lake ice pixels it is necessary to apply a lake mask to the SAR scene prior to the classification, as is also the case for other previously proposed thresholding approaches (see Section 1).

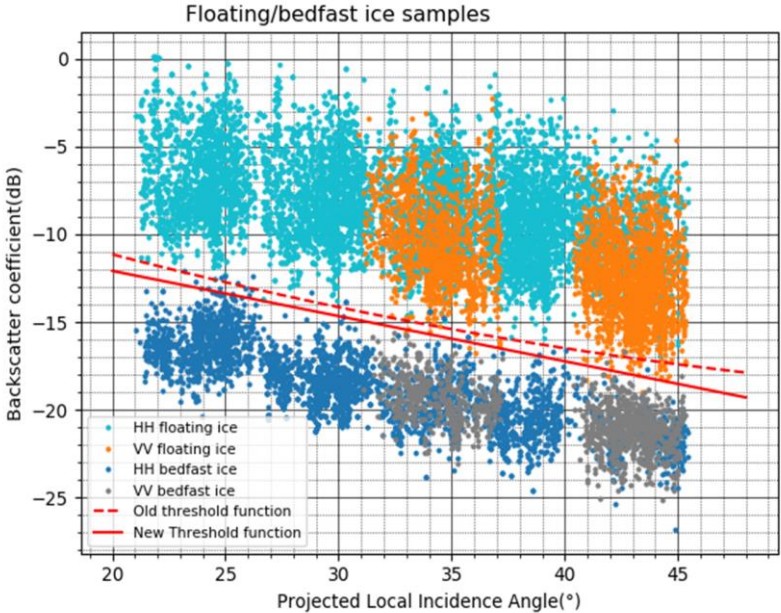

**Figure 6.** Relationship between HH and VV polarized backscatter and projected local incidence angle of floating and bedfast lake ice. The "New Threshold function" represents the threshold function proposed by Duguay and Wang (2019b) for lake ice classification and used in this work for the purpose of comparison. The Figure is adopted from Duguay and Wang, 2019b.

For the purpose of comparison between the thresholding approach and the temporal deep learning approach (TempCNN), lake ice regime maps were created for the four years of S1 data using thresholding: 2021 (March 15), 2020 (March 13), 2019 (March 14), 2018 (March 14). The thresholding algorithm linear function was applied to the local incidence angle layer obtained as part of the RTC level 2 products from ASF. The resulting threshold layer, where each pixel corresponded to the calculated threshold, was applied to classify the VV backscatter layer into either bedfast or floating ice based on whether the backscatter value was above or below the threshold for a given pixel. Next, it was necessary to apply a lake mask. Extraction of lakes is challenging in a wetland environment. As such, for simplicity, a single lake mask was created using an October 3, 2020, scene and a threshold of -16.5 dB identified experimentally by changing the threshold value in increments of 0.5dB, until lake boundaries where accurately captured. The four resulting lake ice regime maps were evaluated in terms of overall accuracy by utilizing the labelled dataset created as part of this work as ground truth. Results were then compared to those obtained from the TempCNN model for the same set of lakes.

**3.7 Lake ice dynamics analysis**

In order to assess ice regime changes that took place in the OCF over the past three decades, change detection between the first (1992/1993) and the last (2020/2021) years of the series was performed. The change map was created using a simple raster calculation: ("class on date 1" *10) + "class on date 2", which created an output where a two-digit number corresponded to

the original class (1st digit) and the subsequent class (2nd digit) (This approach to change detection was introduced by Dr. Jinfei Wang, Western University, London, Canada). In addition, bedfast and floating lake ice fractions were extracted for each year of data. For the ice fraction analysis, a lake mask discussed in the previous subsection was utilized in order to focus on the lake surface, rather than including all of the wetland. This will allow for future comparison to trends observed in other geographic areas of thermokarst lakes. Ideally, a different lake mask would be used for each year of data. However, due to the scope and timeframe of the project it was not feasible to create a different lake mask for each year of data. In order to establish a common baseline, a lake mask from fall of the most recent season in the dataset 2020/2021 (indication of the current status) was used, while acknowledging that this could lead to underestimation of the lake ice surface for years prior to major catastrophic drainage events. Prior to change detection analysis the ice regime maps created from S1 and ERS1/2 time-series were resampled to 50 m to match the largest pixel size of R1. Bedfast lake ice fraction at the time of maximum ice thickness was traced through the 29-year period (1992/1993-2020/2021) and the potential presence of a trend was explored using the Mann-Kendall statistical test in combination with Sen's slope (Sen, 1968). The Mann-Kendall test was selected due to its robustness against missing values in the time-series. The Mann-Kendall test in combination with Sen's Slope was successfully utilized in multiple lake ice studies (e.g., Surdu et al., 2014; Duguay et al., 2006). The python pyMannKendall package was used to perform the test (Hussain and Mahmud, 2019). The observed changes were analyzed in the context of CLIMo ice thickness simulations and known lake drainage and refilling events in the region (Lantz and Turner, 2015; e.g., Lake Zelma and Lake Netro).

## 4 Results and Discussion

### 4.1 TempCNN classification accuracy assessment

Our study shows that temporal deep learning offers a comprehensive framework that does not require a lake mask, as is the case for other studies on the topic of floating and bedfast ice mapping from SAR, and automatically classifies SAR imagery into three classes: floating ice, bedfast ice, and land. This approach offers high-quality ice regime maps for S1, ERS1/2, and R1. Figures 7a, 7b, 7c show examples of TempCNN classification output for S1, R1, and ERS1/2, respectively. Although all the years were classified using the same deep learning model, map quality is dependent on the quality of the input SAR data. Spatial and especially temporal resolution of imagery through the ice season have a significant impact on the output of temporal classification. S1 data having a pixel size of 30 m and a regular full coverage of the study area produced the best results, clearly separating the lake and the land classes. Older ERS1/2 and R1 imagery suffers from irregular and sparse temporal coverage resulting in decreased quality of the temporal classification output. In fact, available ERS1/2 scenes provided full coverage of the study area only 4-8 times in the lake ice season. Therefore, an extensive linear interpolation was employed to fill in the temporal gaps. On the other hand, ERS1/2 high original spatial resolution and small pixel size of 12.5 m allow to resolve finer details. Although R1 had a slightly better temporal coverage, the number of scenes still greatly varied from year to year (16-

45). Moreover, of the three platforms, R1 had the largest pixel size of 50 m, which led to coarser lake outlines in ice regime maps. It should be noted that years with higher temporal resolution and more regular frequency showed better results for all three sensors. Speckle noise is another factor that can significantly compromise the classification accuracy. Speckle filtering with size adjusted based on spatial resolution has allowed to reduce the noise effects.

As has been mentioned in Sect. 3, the TempCNN model classification accuracy was evaluated through 15 experiments. Table 2 presents the overall accuracy of the 15 experiments calculated as total number of correctly classified time-series in the training set divided by the total size of the training set and multiplied by 100%. The overall accuracy ranges from 91-99%. It should be noted that the model appears to be very sensitive to exclusion of certain years from that dataset. The sensitivity is likely attributed to temporal sparseness and irregular frequency of scenes. Especially for ERS1/2, backscatter time-series vary significantly from year to year in terms of shape based on the missing time-periods. Although linear interpolation fills in the gaps, the major features of the lifecycle can be lost.

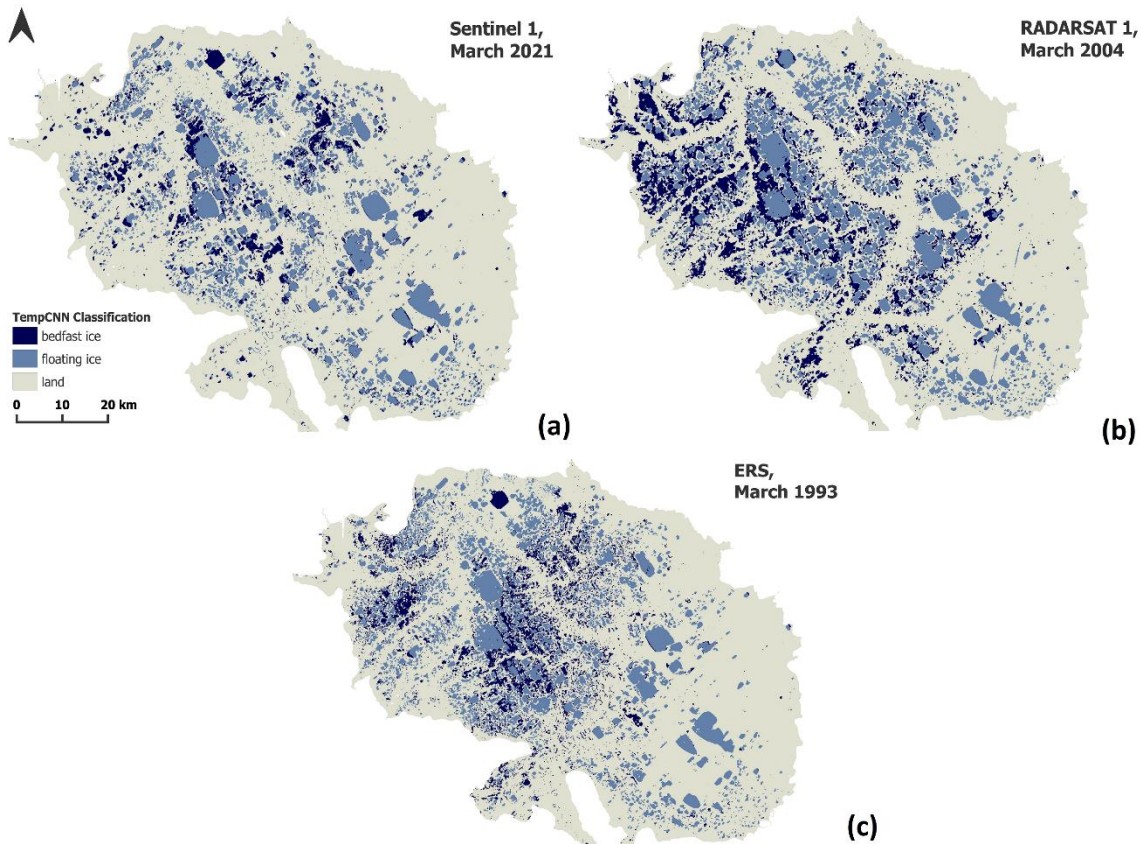

**Figure 7.** A sample of TempCNN classification output for OCF, Yukon, Canada: (a) Sentinel-1, March 2021; (b) RADARSAT-1, March 2004; (c) ERS, March 1993. Dark blue, light blue, and grey represent bedfast ice, floating ice, and land, respectively.

**Table 2. TempCNN overall classification accuracy for 15 experiments designed to test sensitivity of the network to removing certain years of data from the training set. Runs 1-5 correspond to the 20/80% split of the entire dataset, runs 6-10 were performed by training the network on 15 years of data and testing it on 3 each from a different sensor, runs 10-15 were carried out by training the network on 17 years of data and testing it on 1 year of data that was originally reserved and was not part of the cross validation procedure for determining the best architecture. Mean accuracy for each set of 5 runs as well as the mean of the three means can be found in the last two rows.**

| Test set | 20% | | | | | 3 years | | | | | 1 year | | | | |
|---|---|---|---|---|---|---|---|---|---|---|---|---|---|---|---|
| Run | 1 | 2 | 3 | 4 | 5 | 6 | 7 | 8 | 9 | 10 | 11 | 12 | 13 | 14 | 15 |
| Overall accuracy | 99.6 | 99.49 | 99.66 | 99.7 | 99.59 | 92.33 | 90.55 | 92.78 | 91.7 | 90.62 | 91.8 | 96.51 | 94.68 | 90.62 | 96.04 |
| Mean accuracy | 99.61 | | | | | 91.6 | | | | | 93.93 | | | | |
| Mean of means | 95.05 | | | | | | | | | | | | | | |

The following pattern emerged when analyzing confusion matrices from all experiments: the 80/20% models had very minimal misclassifications equally distributed between the three classes; the models tested on three years had most significant misclassification of land as bedfast ice; and the models tested on one year of S1 had most significant misclassifications of floating ice as bedfast ice. The better performance of the 80/20% model is believed to be attributed to the high inter-year temporal variability of available scenes. As such when time-series from each year of data are present in both the training and the test set higher accuracy is achieved. Land misclassification may be caused by the fact that the OCF is a wetland with varying distribution of surface water (water extent and level) that are frequently changing. Backscatter evolution of areas that have standing water will be very similar to that of bedfast lake ice making the distinction between the two classes challenging. It has been mentioned by a few researchers (e.g., Engram et al., 2018; Pointner et al., 2019), that deeper thermokarst lakes often display darker (low backscatter) patches of floating ice in their centers. Some lakes of the OCF manifest such patterns. Moreover, some of the years have characteristic dark spots in most of the lakes. The dark spots are particularly pronounced for R1 2001; ERS 1993, 1994. Some of the recent studies (e.g., Engram et al., 2020; Pointner and Bartsch, 2020) suggest that the dark spots pattern could be a result of local ice thinning or complete melt caused by methane ebullition. Although these patterns have been accounted for in the labeled dataset, the model still appears to struggle with some of those cases. In addition, Antonova et al. (2016) and Engram et al. (2018) also suggest that temperature of the underlying sediments, and whether they are fully frozen or not once the ice has become bedfast, could have an impact on the SAR backscatter.

## 4.2 Assessment against field observations and lake ice thickness simulations

Although an extensive accuracy assessment has been performed using the labeled dataset, the model prediction is only as accurate as the labeled data provided for its training and testing. Only extensive fieldwork can truly validate the accuracy of the created ice regime maps. However, most of the field work conducted in the OCF is performed in summer (Labrecque et al., 2009; Lantz and Turner, 2015; Turner et al., 2010). Winter lake ice observations are largely non-existent for the OCF so that little is known about the actual ice regime (floating and bedfast ice) patterns of its lakes. As such, due to the lack of ground

truthing, the labeled dataset used for model training was created from visual interpretation of SAR imagery and, therefore, could be subject to human error. In order to leverage the few field observations available to us and ice thickness simulated using CLIMo, classification results were compared against point lake depth measurements collected in 2000, observations of ice thickness and lake depth (or ice regime) at drilling holes collected in early April 2009 and 2021, and ice thickness model output from CLIMo. CLIMo simulated ice thickness has been successfully used in absence or in addition to limited field

observations to provide context for SAR-derived lake ice regime findings by Surdu et al. (2014), Antonova et al. (2016), and Nitze et al. (2020).

The lake ice regime is determined by multiple factors, including atmospheric conditions such as air temperature and snow accumulation on the ice surface (i.e., depth and density), which, in turn, is impacted by wind and the presence/absence of vegetation cover surrounding the lake (forest and shrubs) (Duguay et al., 2003; Jeffries et al., 2005). Another obvious control

on the lake ice regime is lake bathymetry; areas that exceed the depth of maximum ice thickness remain afloat, while shallower areas become bedfast (Arp et al., 2011). As such, point lake depth measurements collected in the OCF in late July 2000 were compared with the ice regime classification produced by the TempCNN model for the mid-March of 2000. Vegetation is another important factor. The OCF is a forest-tundra ecotone, and while some lakes are surrounded by tundra, namely dwarf shrubs and herbaceous plants, some are located in deciduous, coniferous, or mixed forest, and others are fringed by a

combination of tundra and forest environments (Turner et al., 2014). In tundra environments, wind controls snow depth and density, which have a greater influence on the ice thickness than air temperature (Brown and Duguay, 2010; Roy-Léveillée et al., 2014). For lakes surrounded by tundra the snow cover is expected to be thinner and denser leading to faster ice growth (Roy-Léveillée, 2014). In contrast, for taiga lakes surrounded by mixed and deciduous forest that facilitates greater loose snow accumulation providing insulation, ice growth is hindered (Roy-Léveillée, 2014). Figure 8 shows a plot of lake depths against

mid-March SAR backscatter where the color of the points indicates the class predicted by TempCNN, while the shape refers to the surrounding vegetation (tundra, taiga, or combination of the two). In addition, the horizontal lines indicate the ice thickness simulated with CLIMo for mid-March of 2000 for two different snow scenarios. The line labeled as "CLIMo taiga ice thickness" represents a scenario of 100% snow cover and 175 kg m$^{-3}$ snow density, while the line labeled as "CLIMo tundra ice thickness" represents a snow scenario of 50% snow cover and 300 kg m$^{-3}$ snow density (Fig. 9).

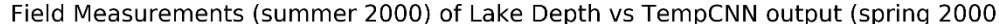

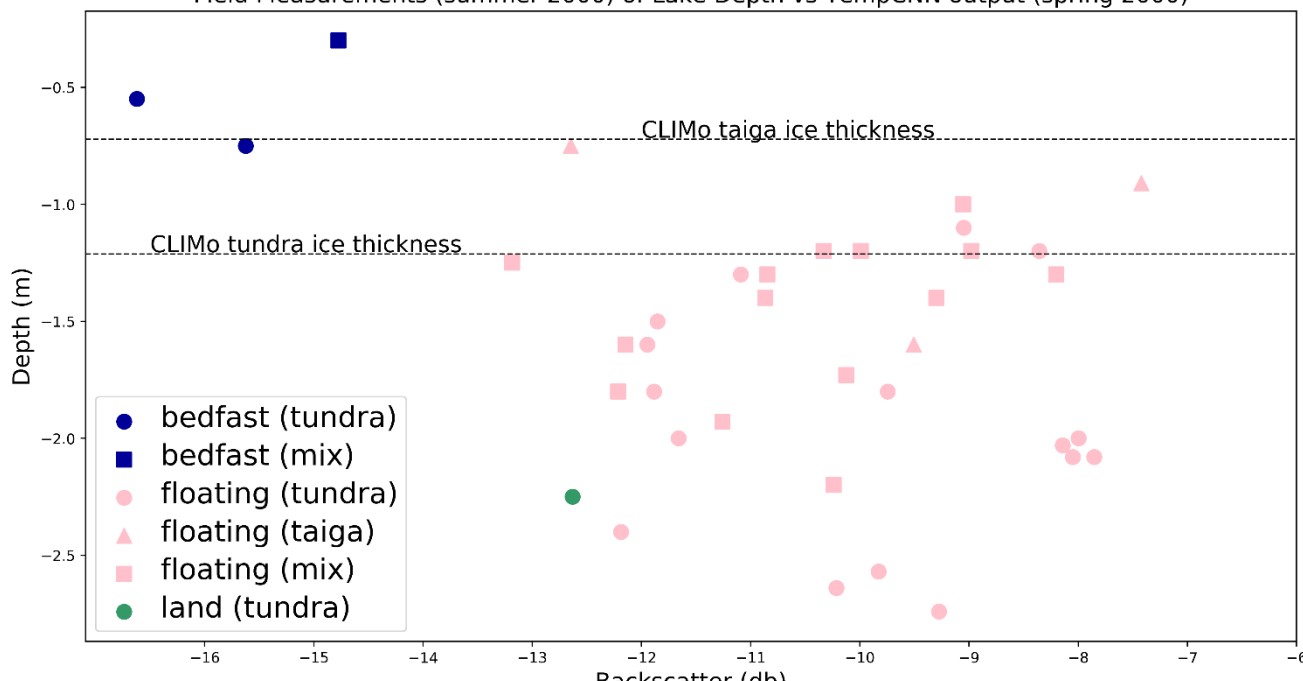

**Figure 8.** Matching TempCNN output with lake depths collected in 2000. Horizontal lines indicate CLIMo ice thickness predictions for taiga (0.72 m) and tundra (1.21 m) environments. RADARSAT-1 SAR 1999/2000 time-series were used for ice regime classification. The colour of points corresponds to labels assigned to each location by the TempCNN: dark blue – bedfast ice, pink – floating ice, green – land; the shape corresponds to the surrounding vegetation: circle – tundra, triangle – taiga; square – mixed assigned based on the OCF vegetation map created by Turner et al., 2014.

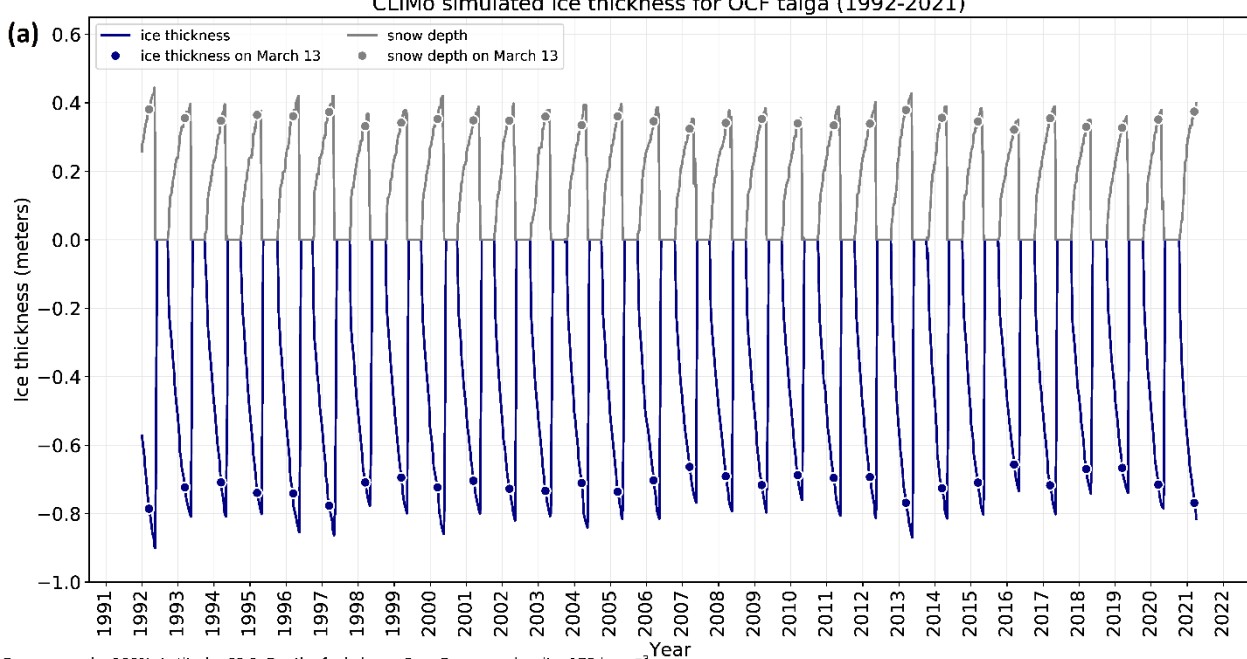

**(a)** CLIMo simulated ice thickness for OCF taiga (1992-2021)

Snow scenario: 100%, Latitude: 68.0, Depth of mix layer: 2 m, Dry snow density: 175 kg m$^{-3}$

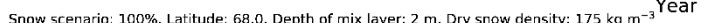

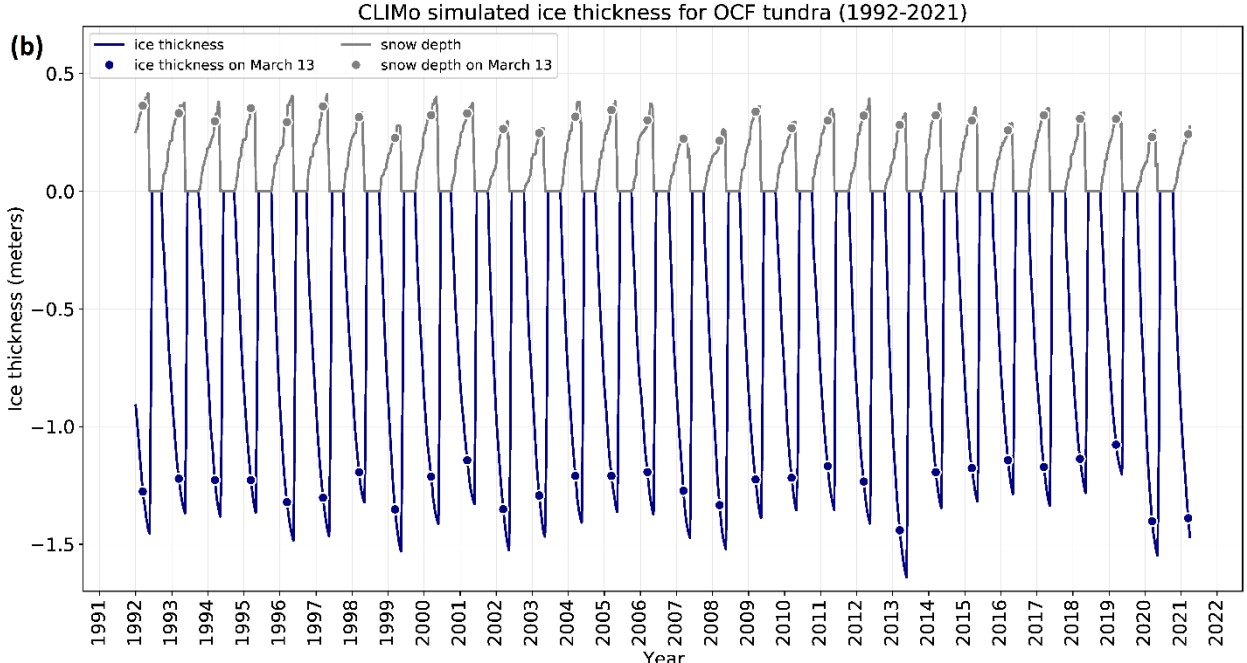

**(b)** CLIMo simulated ice thickness for OCF tundra (1992-2021)

Snow scenario: 50%, Latitude: 68.0, Depth of mix layer: 2 m, Dry snow density: 300 kg m$^{-3}$

**Figure 9.** CLIMo simulated ice thickness for OCF: (a) simulation for the taiga environment; (b) simulation for the tundra environment. Dark blue represents ice thickness and grey stands for snow depth. The grey and dark blue points mark the condition on March 13 (the last day of the time-series) for each year.

The three bedfast (dark blue) locations align well with the CLIMo results. The two tundra bedfast locations (dark blue circles) correspond to depths shallower than the CLIMo simulated ice thickness for tundra, while the one bedfast location surrounded by taiga (dark blue square) falls above the CLIMo simulated ice thickness for taiga. All of the tundra, taiga, and mixed vegetation locations with depths greater than the CLIMo simulated tundra ice thickness are classified as floating by the TempCNN indicating its accuracy. Out of the four floating ice points that fall in between the two CLIMo simulated values, two are surrounded by taiga (pink triangles) and logically fall below the CLIMo simulated taiga ice thickness, while the remaining two points are surrounded by tundra (pink circle) and mixed vegetation (pink square). It is difficult to claim with certainty whether the two latter points are a TempCNN misclassification or are a result of local snow cover and vegetation variations as CLIMo output is sensitive to the input values of snow depth and density. As to the one point classified as land (green circle), it belongs to a small pond that contains floating vegetation (Turner et al., 2014) whose backscatter signal was likely diluted in the surrounding land signal due to low spatial resolution and large pixel size of R1. Looking at Fig. 8 it can be noticed that the deeper the lake, the lower is the backscatter of floating ice. This relationship contributes to the challenge of distinguishing between floating and bedfast ice in deeper lakes noted by a few researchers (Duguay and Wang, 2019a; Engram et al., 2018; Pointner et al., 2019).

Apart from depth measurements, a small amount of ice thickness data was collected in OCF in April of 2009 and 2021. The results can be seen in Tables 3 and 4. For 2021, the ice regime of the four points matches the TempCNN classification output. The field-observed ice thickness is between 77-81 cm, which is consistent with the taiga CLIMo prediction for the early April of 2021 (Fig. 9). Table 4 also illustrates the effect of snow depth on the ice growth, as thicker snow above drill hole 3 corresponds to thinner ice, and thinner snow above drill hole 4 leads to thicker ice. For 2009, the first eight drill holes were made in tundra, where ice grows thicker (Table 3). All but one of the tundra drill hole locations were classified correctly by the TempCNN, with five floating ice locations (ice thickness less than lake depth), and two bedfast ice locations (ice thickness equal to lake depth). The first drill hole was misclassified as land. However, the land pixel is directly adjacent to the lake border and is likely a result of the drill hole being close to the lake shore which in combination with low spatial resolution leads to mixing of backscatter contributions from the lake and the surrounding taiga. In taiga (two last drill holes), the ice is noticeably thinner. While one of the points was correctly classified as floating, the last drill hole was misclassified as land for the reasons discussed above.

**Table 3. Field data collected in the OCF in April 2009. For each of the ten locations, UTM coordinates, ice thickness, lake depth, and ice regime recorded in the field are matched with the TempCNN output (bedfast, floating, land) and vegetation type (tundra or taiga).**

| Drill hole | UTM Easting/Northing (m) | Ice thick ness (cm) | Lake depth (cm) | Ice regime | TempCNN predicted ice regime | Vegetation |
|---|---|---|---|---|---|---|
| 1 | 565714/ 7532111 | 109 | 154 | floating | land | tundra |
| 2 | 565765/ 7532196 | 124 | 208 | floating | floating | tundra |
| 3 | 564146 7535012 | 125 | 145 | floating | floating | tundra |
| 4 | 564104/ 7535066 | 130 | 200 | floating | floating | tundra |
| 5 | 564646/ 7535556 | 140 | 232 | floating | floating | tundra |
| 6 | 566304/ 7535568 | 102.5 | 102.5 | bedfast | bedfast | tundra |
| 7 | 566278/ 7535604 | 151 | 151 | bedfast | bedfast | tundra |
| 8 | 564669/ 7539300 | 132.5 | 213 | floating | floating | tundra |
| 9 | 543100/ 7569101 | 77.5 | 232 | floating | floating | taiga |
| 10 | 543418/ 7568436 | 100 | 191 | floating | land | taiga |

**Table 4. Field data collected in April 2021 on a small lake located beside the drained basin of Zelma Lake. For each of the four locations, UTM coordinates, ice regime, ice thickness, snow depth, and sediment temperature are matched with the TempCNN output.**

| Drill hole | UTM Easting/Northing (m) | Ice regime | Ice thickness (cm) | Snow depth (cm) | Sediment temperature (°C) | Backscatter (db) | TempCNN predicted ice regime |
|---|---|---|---|---|---|---|---|
| 1 | 545779/ 7536142 | bedfast | 67 | 46 | <0 | -18 | bedfast |
| 2 | 545850/7536206 | bedfast | 79 | 37 | 0.5 | -16.4 | bedfast |
| 3 | 545924/7536277 | floating | 77 | 49 | 0.5 | -10.17 | floating |
| 4 | 545996/7536339 | floating | 81 | 39 | 0.5 | -9.36 | floating |

Finally, using an example of a lake traditionally known as Husky Lake, we can assess the TempCNN ice regime classification (Fig. 10) in the context of CLIMo simulated ice thickness over the 29-year period (1992/1993-2020/2021) (Fig. 9). Due to the fact that Husky Lake is surrounded mainly by dwarf shrubs and herbaceous vegetation, tundra CLIMo simulation results were used. Aligning CLIMo simulated ice thickness with TempCNN classification results it was noticed that all years with simulated ice thickness equal to or thinner than 1.14 m were classified as floating ice, while all years with simulated ice thickness equal to or greater than 1.29 m were classified as bedfast. Years with simulated ice thickness ranging between 1.21 m and 1.23 m fluctuated between the two ice regimes. This could be attributed to the fact that CLIMo has a general uncertainty of approximately 0.05-0.06 m (Duguay et al., 2003) which could explain the fluctuations. In addition, according to Tondu (2012), depth of Husky Lake is approximately 1 m based on 2010/2011 observations. As such, CLIMo results for tundra are most likely overestimating ice thickness, while taiga simulation is underestimating it. Consequently, years where CLIMo simulated ice thickness ranged between 1.21 m and 1.23 m were likely characterized by ice thickness very close to the depth of Husky

Lake, resulting in different ice regimes for different years. The two years where CLIMo results seemed to contradict the TempCNN predictions were 2006 (1.19 m CLIMo simulated ice thickness; bedfast label) and 1996 (1.32 m CLIMo simulated ice thickness; floating label). It is worth noting that ice thickness is not the only factor that dictates lake ice regime. Fluctuating lake water levels could result in different ice regimes given the same winter atmospheric conditions especially considering the shallow nature of OCF lakes (Surdu et al., 2014). Therefore, mismatch between CLIMo and TempCNN predictions is most likely rooted in the water level difference. Summer preceding spring of 2006 was characterized by lower water levels (drier year) based on visual analysis of Landsat optical imagery. Lower water level could potentially explain ice grounding in spite of thinner CLIMo simulated ice thickness. For 1996, on the other hand, the reason for floating ice regime, despite greater than usual CLIMo simulated ice thickness, was not identified. However, it is important to note that CLIMo simulations can be impacted by the quality of the atmospheric forcings input data. Visual analysis of mid-March SAR scenes in 1996 indicates widespread floating ice throughout the OCF.

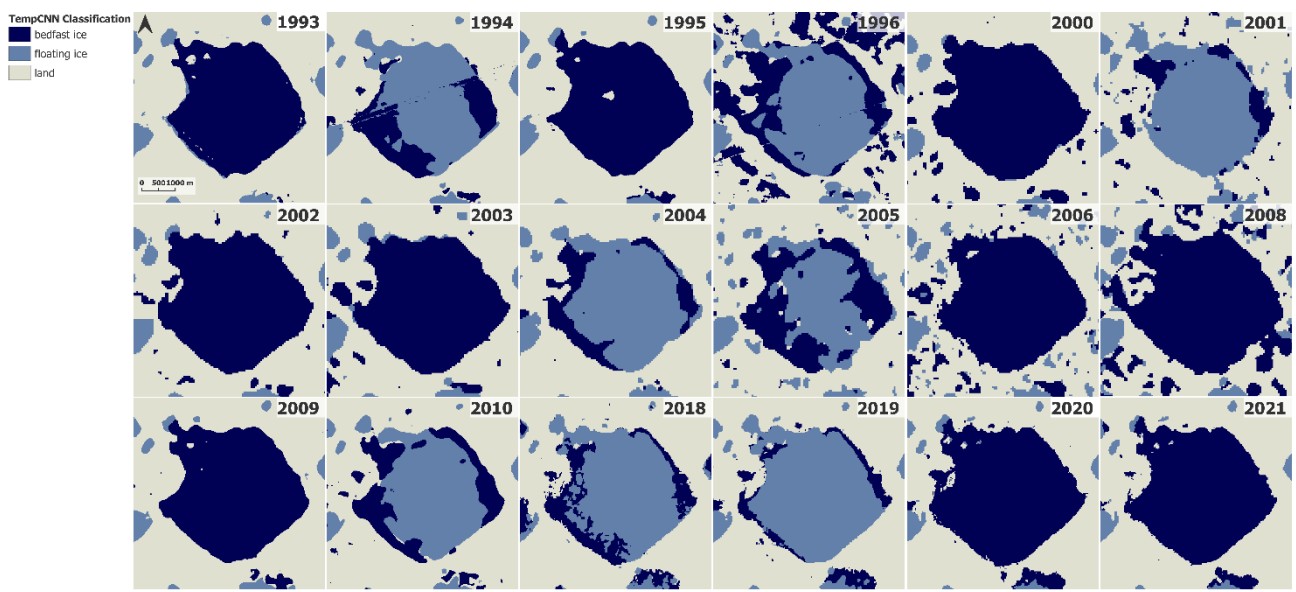

**Figure 10.** Husky Lake TempCNN predicted ice regime (1992/1993-2020/2021). Dark blue represents bedfast ice, light blue – floating ice, grey – land. Ice regime fluctuates between floating and bedfast depending on snow conditions, water level, and air temperature.

**4.3 Comparison to the state-of-the art thresholding approach**

To benchmark the proposed temporal deep learning approach against the state-of-the-art methods of lake ice regime classification from SAR, a brief comparison to the thresholding algorithm proposed by Duguay and Wang (2019b ) was carried out. The overall accuracy for each year was found to be as follows: 2018 – 87.8%; 2019 – 99.4%; 2020 – 98.8%; 2021 – 99.3%, with a mean accuracy of 96.3%. The mean overall accuracy of the TempCNN model with a 20/80% testing and training split which was used to create lake ice maps further employed for lake ice dynamics analysis of the OCF was 99.6 % (Table

2). It should be noted that the overall accuracy for TempCNN was carried out using the 18 years of data, while thresholding algorithm was evaluated using only 4 years of S1 data showing accuracies ranging from 87.8% to 99.4%. Figure 11 contains a side-by-side comparison of the lake ice maps created by the TempCNN (Fig. 11a) and the thresholding algorithm (Fig. 11b) for March 2021. Visual analysis of the results of both methods are rather similar. Nonetheless, let us summarize the benefits and shortcomings of both methods. The thresholding algorithm: (1) produces highly accurate results overall for the four years

examined; (2) is simple in implementation; however, (3) requires a lake mask; due to the dynamic nature of the wetlands a new mask would be needed for each year for the best results; (4) a local incidence angle layer is necessary; and (5) this algorithm has been designed to work with S1 data (VV and HH polarizations), while its suitability for other SAR platforms is yet to be explored. Temporal deep learning (TempCNN): (1) is more complex in implementation due to the requirement for time-series of scenes, rather than one scene; nonetheless, (2) produces highly accurate results; (3) does not require a lake mask

due to its ability to classify VV and HH backscatter into three classes: floating ice, bedfast ice, and land, which is critical for dynamic thermokarst landscapes; (4) does not require incidence angle information; (5) based on visual comparison of the lake ice maps, TempCNN is better at classifying lake ice in deeper portions of larger/deeper lakes than the thresholding algorithm (in Fig. 11b, some misclassifications of floating ice as bedfast ice are observed in large lakes), and (6) the approach is applicable to multiple SAR platforms (S1 - VV polarization, ERS1/2 - VV polarization, and R1 - HH polarization) as shown in this study.

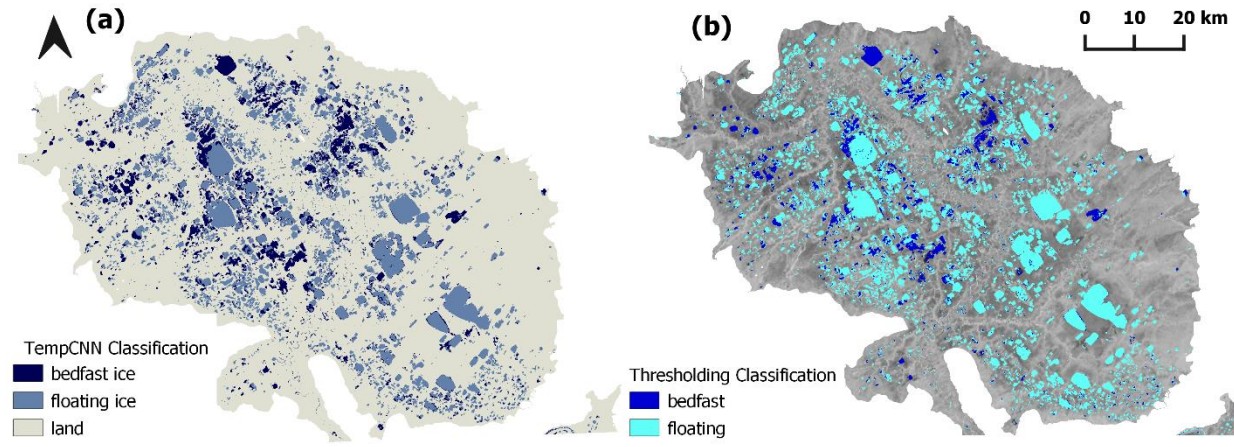

**Figure 11.** Comparison of the (a) TempCNN lake ice regime classification and (b) thresholding algorithm output which defines the threshold between the two classes as a linear function of the local incidence angle (Duguay and Wang, 2019b).

## 4.4 OCF lake ice dynamics analysis

With the goal of analyzing ice dynamics of lakes within the OCF, this study yielded 18 lake ice regime maps from 1992/1993
to 2020/2021 created using SAR image time-series and the trained TempCNN deep learning model. The classified maps
contain three classes, where two classes characterize the lake surface: bedfast ice and floating ice, and the remaining class
encompasses the surrounding land area, including various vegetation types (Turner et al., 2014).

Performing change detection between the first (1992/1993) and the last years of the dataset (2020/2021), reveals a transition
of 51 $km^2$ from bedfast to floating ice regime. However, 172 $km^2$ of floating ice shifted to a bedfast state. In addition, over the
29-year period, 256 $km^2$ transitioned from land to water reflecting active erosion processes (Roy-Léveillée and Burn, 2015),
out of which 106 $km^2$ transitioned from land to bedfast ice, and 159 $km^2$ transitioned from land to floating ice. Moreover, 450
$km^2$ shifted from water to land, including 341 $km^2$ bedfast ice to land and 109 $km^2$ floating ice to land. It is crucial to remember
that no lake mask was used and that OCF is a wetland. As such, a significant extent of area that transitioned from land to water
and water to land does not consist only of lake drainage, refilling, expansion, and shrinkage, but also captures the changing
nature of the wetland. Figure 12 highlights the lake ice regime transitions in dark (floating to bedfast) and light (bedfast to
floating) blue, and also shows areas that switched from floating ice to land in green. The three insets illustrate each of the
change types. According to ERA5 atmospheric data and CLIMo ice thickness simulations, the winter of 1993 was warmer
than the winter of 2021 with the maximum ice thickness of 0.81 m for taiga in both years, and 1.37 m and 1.47 m in 1993 and
2021, respectively, for tundra. In mid-March, the simulated ice thickness was 0.72 m for taiga and 1.22 m for tundra in 1993
and 0.77 m for taiga and 1.39 m for tundra in 2021. Based on the ERA5 atmospheric data displayed in Fig. 13 one can notice
that apart from the air temperature difference, 2021 was characterized by a significantly lower depth of the snow cover
compared to 1993 and most of the years in the dataset. In fact, the total snow accumulation over the period between October
4, 2020, and March 13, 2021, was 30% lower than the 29-year mean. Reduced snow insulation allowed thicker ice to develop
which is reflected in greater ice thickness simulated for 2021. As such, lower air temperature and reduced snow depth help
explain part of the transition from floating to bedfast regime as for shallow flat-bottomed OCF lakes a 5-17 cm difference in
ice thickness could allow for ice grounding. Moreover, as has been mentioned above, water levels play a significant role in
winter lake ice conditions. As indicated by Lantz and Turner (2015) who studied OCF lake area changes in response to
thermokarst processes and climate change between 1950 and 2009, the annual rate of lake drainage has increased 4-5 times in

the recent decades. Looking at the change map, one can notice multiple locations where the significant ice regime change was

indeed caused by significant changes in water level, in particular lake drainage and refills.

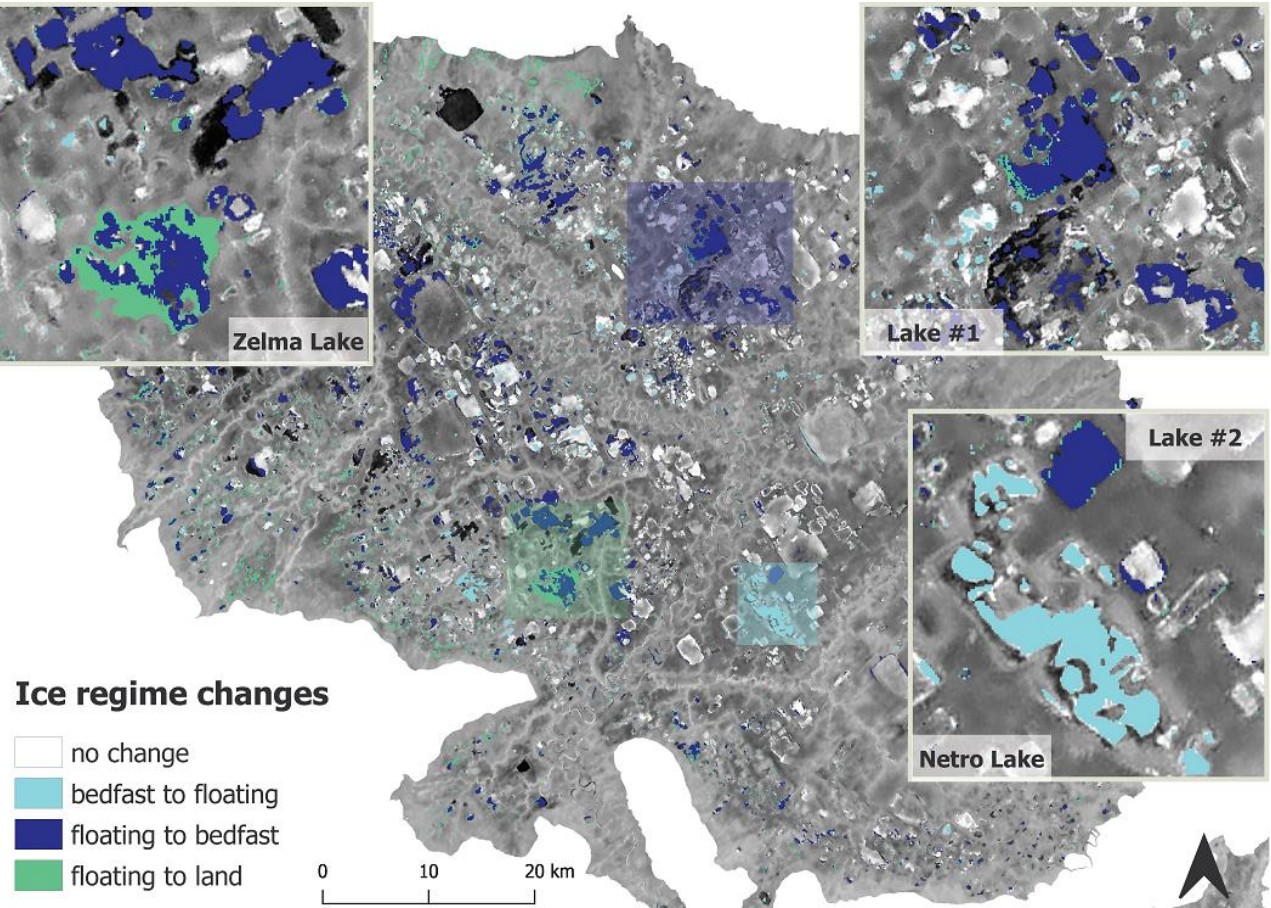

**Figure 12.** Change detection between 1992/1993 (ERS) and 2020/2021 (Sentinel-1) ice regime maps. Dark blue represents transition from floating to bedfast ice regime; light blue shows transition from bedfast to floating ice regime; green stands for transition from floating ice regime to land. The inset map in the top left corner is a zoomed in view of the area highlighted as a green rectangle on the main map; top right corner - dark blue rectangle; bottom right corner - light blue rectangle.

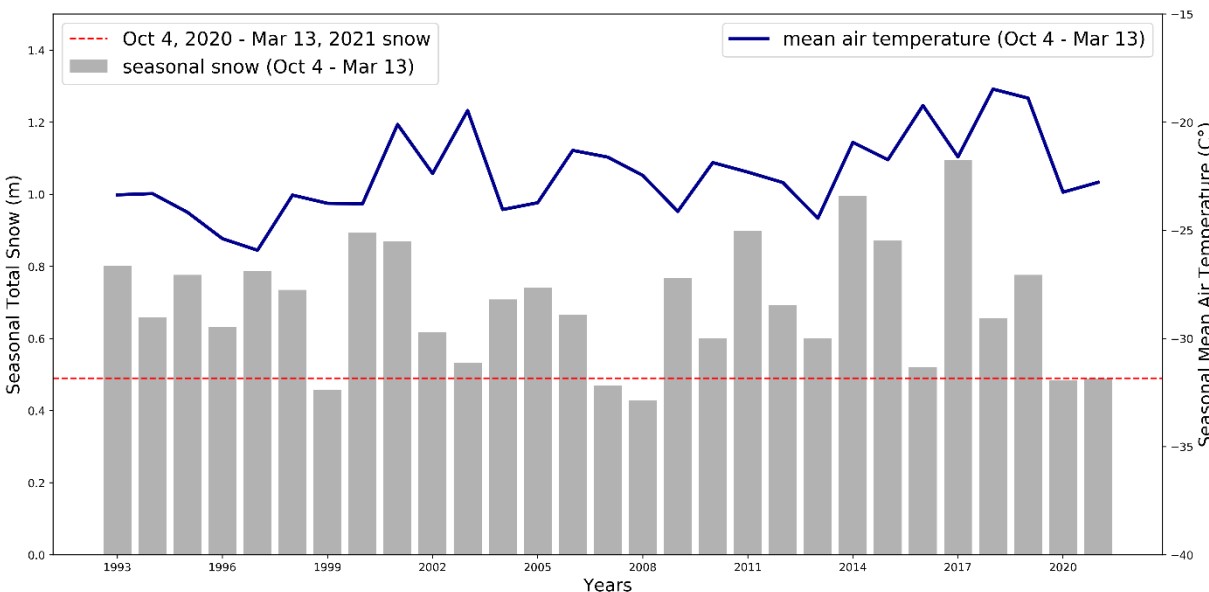

**Figure 13.** Total seasonal snow and mean seasonal air temperature in OCF from 1992/1993 to 2020/2021 (Oct 4- Mar 13) calculated based on the daily ERA5 (global atmospheric reanalysis data, produced using ECMWF model freely available through the Copernicus Climate Change Service). The horizontal dashed red line assists in visually comparing 2021 snow to all the other years.

Firstly, let us look in more detail at Netro Lake that represents the most significant transition to a floating ice regime. Figure 14 illustrates its gradual transition from mostly bedfast to a mostly floating ice regime. It has been reported by Labrecque et al. (2009) that Netro Lake drained catastrophically losing 10.51 km² between 1972 and 2001. However, it appears that Netro Lake has not only refilled, regaining a lot of its area back, but has also transitioned to a mostly floating ice regime. The later transition coincided with the catastrophic drainage of a neighboring lake which spilled in Netro Lake in 2019.

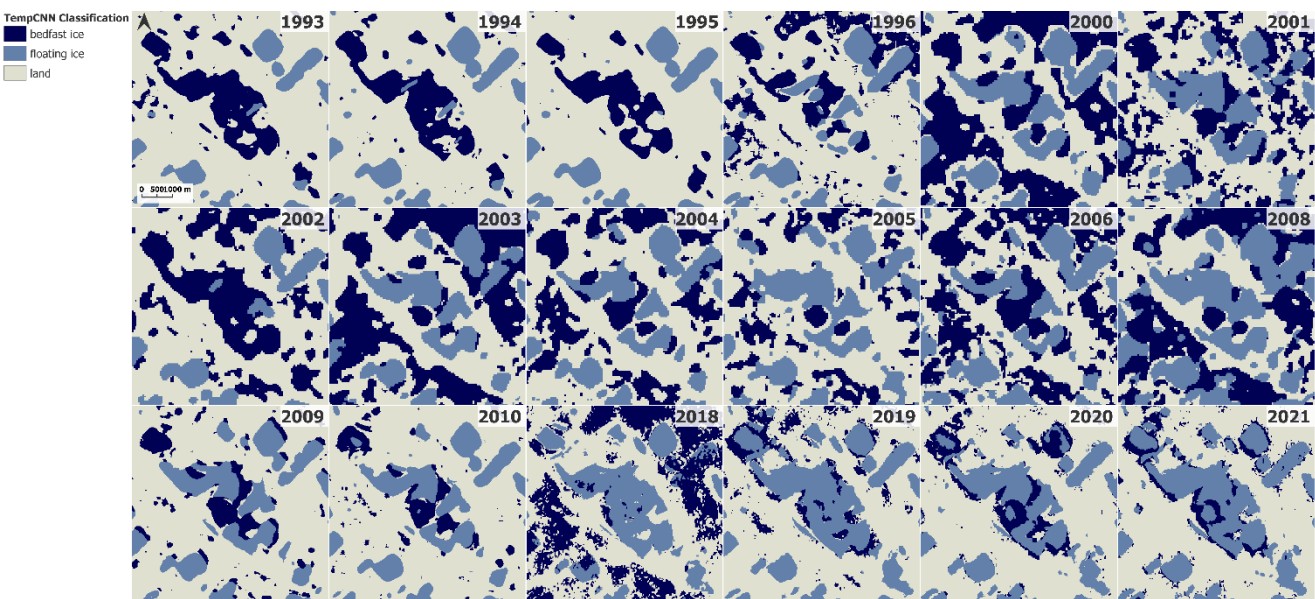

**Figure 14.** Netro Lake TempCNN predicted ice regime (1992/1993-2020/2021). Dark blue represents bedfast ice, light blue – floating ice, grey – land. Between 1992/1993 and 2020/2021 ice regime has transitioned from mostly bedfast to mostly floating.

The second example is Zelma Lake. Zelma Lake drained catastrophically in summer of 2007 through an outflow gully into a nearby creek losing 43% of its area over the course of a few weeks (Lantz and Turner, 2015; Turner et al., 2010). The drainage

exposed approximately 5.2 km² (Turner et al., 2010) of lakebed leaving behind shallow remnant ponds. Figure 15 illustrates ice regime dynamics over the course of the past 29 years. A significant reduction of lake area can clearly be seen between 2006 and 2008 maps. It should also be noted that the ice regime after the drainage has mostly become bedfast for years 2009, 2010, 2020, and 2021 following the drainage. The fact that 2018 and 2019 contain patches of floating ice in deeper areas can be explained by these two years being significantly warmer than the others resulting in thinner ice (Fig. 9 and 13).

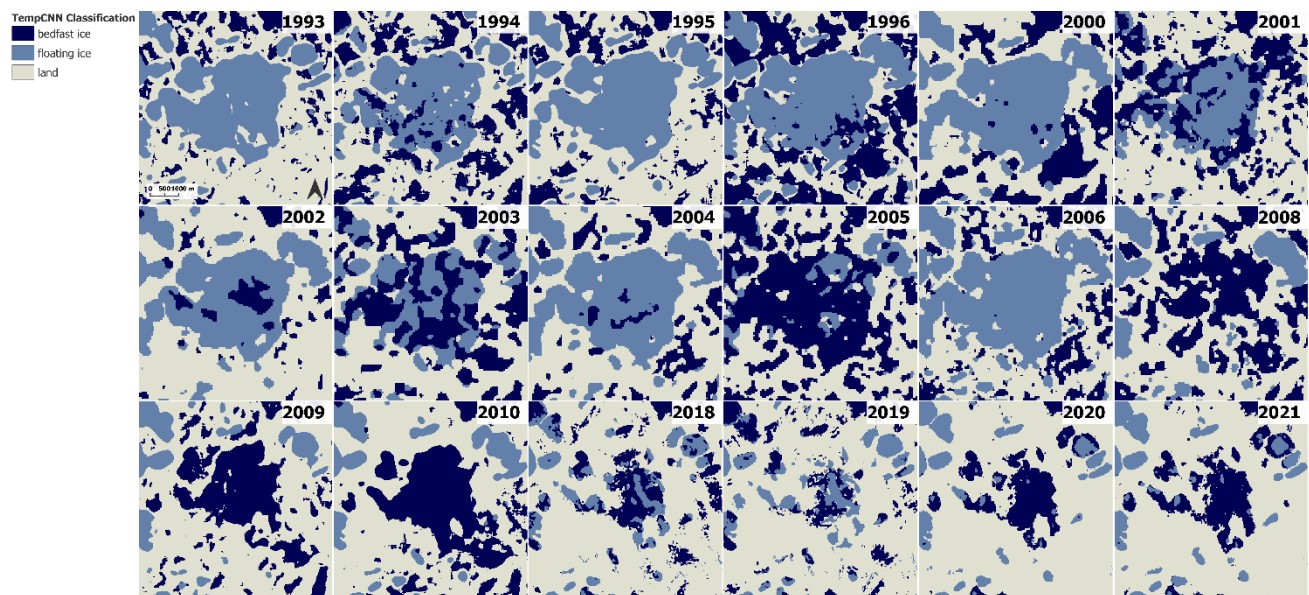

**Figure 15.** Zelma Lake TempCNN predicted ice regime (1992/1993-2020/2021). Dark blue represents bedfast ice, light blue – floating ice, grey – land. You can notice a significant reduction in water surface area and a transition to mostly bedfast ice regime following the 2007 catastrophic drainage event.

The two other lakes that experienced drainage events in recent years are referred to as Lake #1 and #2 in Fig. 12. Lake #1 and #2 presumably drained in 2017 and 2018, respectively, and therefore are not mentioned in the study of Lantz and Turner (2015). Figure 16 presents lake ice regime change detection between 2010 and 2018 for both lakes. Change maps reveal extensive transition to a bedfast lake ice regime (dark blue) as the lakes drained and became shallower. Moreover, a significant area at the lake fringes transitioned to land. Both change maps are supplemented with Landsat images that clearly demonstrate exposed land at lake edges. It is worth noting that extensive lake drainage (192 lakes) has been recently documented for nearby northwestern Alaska in 2018 and explained by winter 2017/2018 being the warmest and wettest on record (Nitze et al., 2020).

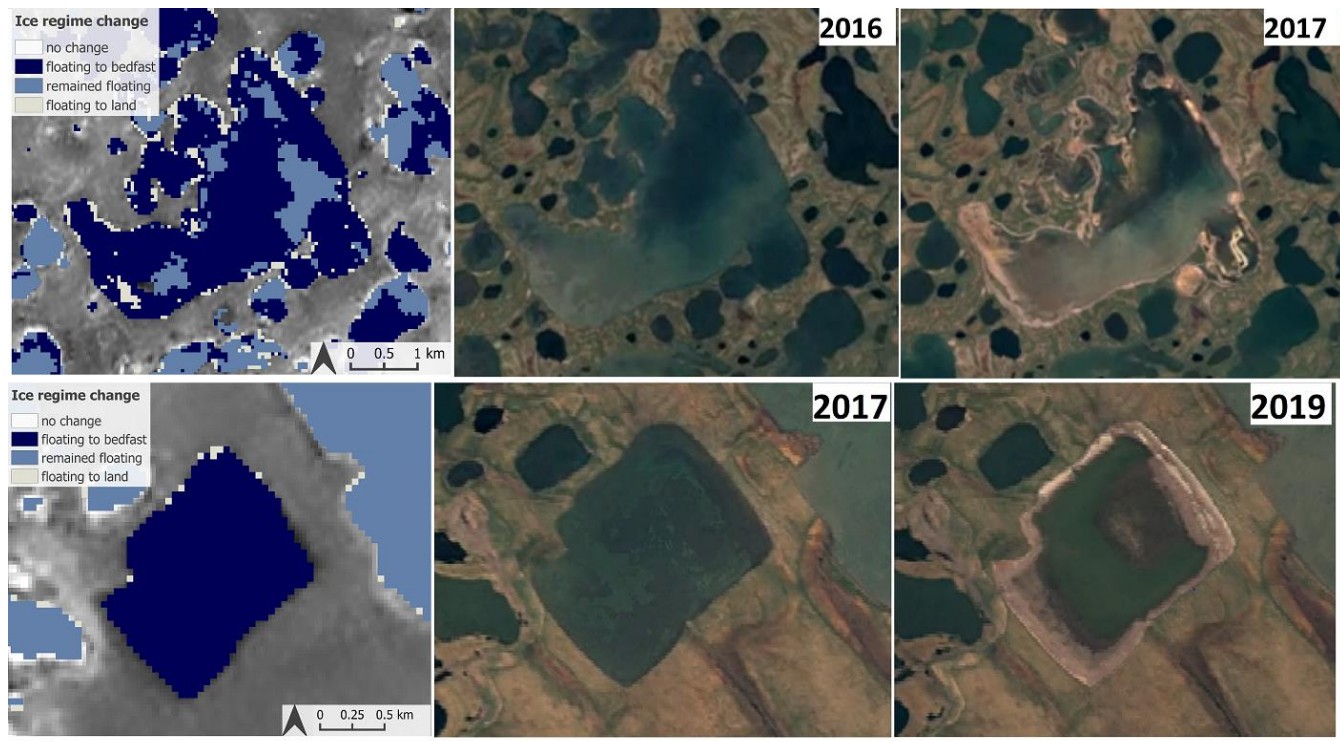

**Figure 16.** (top) Lake #1: Transition from floating to bedfast ice regime between 2010 and 2018 (left). Water level changes visible on Landsat imagery between summers of 2016 (center) and 2017 (right). (bottom) Lake #2: Transition from floating to bedfast ice regime between 2010 and 2018 (left). Water level changes visible on Landsat imagery between summers of 2017 (center) and 2019 (right). Landsat scenes were obtained from the Google Earth Timelapse tool (link: https://earthengine.google.com/timelapse/, accessed: August 15, 2021).

It is important to consider that a variety of simultaneous processes are working to modify the OCF landscape. For instance, while larger lakes drain losing surface area, smaller ponds are forming and expanding through permafrost thaw and erosion processes (Labrecque et al., 2009; Roy-Léveillée and Burn, 2010; Roy-Léveillée and Burn, 2015) as is illustrated in Fig. 12. However, based on the above examples, it appears that the OCF lake ice dynamics are significantly impacted by drainage events driven by a combination of climate change and thermokarst processes (Labrecque et al., 2009; Lantz and Turner, 2015).

To capture inter-annual lake ice dynamics, we have analyzed bedfast and floating lake ice fractions throughout the 29-year period (18 years of data). For simplicity, a lake mask created using an early October S1 SAR scene from 2020 was utilized to isolate lake areas. Table 5 shows ice dynamics in terms of ice fractions as well as area. We can observe significant inter-annual fluctuations, most likely driven by varying atmospheric conditions (air temperature and snow depth) and water level changes. However, over the 29-year period, a decrease in floating ice fraction and an increase in bedfast ice fraction is observed. It appears that the main drivers behind the observed trend are the lake drainage events, which lead to significant reduction in water level allowing for subsequent ice grounding and increasing frequency of dry years characterized by reduced snow cover,

negative water balances, and low lake water levels. The mean bedfast lake ice fraction for the period between 1993 and 1996 is 11%, for 2000 to 2006 is 15%, for 2008 to 2010 is 12%, and for 2018 to 2021 is 25%. It is expected that the extent of over- or underestimation of the bedfast ice fraction varies between the above-listed time periods due to the difference in spatial resolution. Based on the Mann-Kendall test, the bedfast lake ice fraction between 1992/1993 and 2020/2021 displayed an increasing trend at a 10% significance level. The trend line fitted using the Sen's slope test indicates an increase of about 11% in the fraction of bedfast ice over the 29-year period ($0.375\%$ $yr^{-1}$). 2018 and 2021 stand out among other years in the dataset due to a significantly higher bedfast ice fraction (Table 5). Nonetheless, more extensive field validation is required for drawing more definitive conclusions regarding the progression of bedfast ice fraction in the OCF.

**Table 5. Bedfast and floating lake ice fractions and area ($km^2$) in OCF from 1992/1993 to 2020/2021. A lake mask created using 2020/2021 lake extent was used to extract lake ice fractions.**

| Year | Floating ice fraction (%) | Bedfast ice fraction (%) | Floating ice area ($km^2$) | Bedfast ice area ($km^2$) |
|------|------|------|------|------|
| 1993 | 88 | 12 | 790 | 108 |
| 1994 | 86 | 14 | 779 | 127 |
| 1995 | 88 | 12 | 848 | 111 |
| 1996 | 92 | 8 | 909 | 82 |
| 2000 | 90 | 10 | 825 | 93 |
| 2001 | 81 | 19 | 753 | 175 |
| 2002 | 86 | 14 | 796 | 134 |
| 2003 | 79 | 21 | 746 | 203 |
| 2004 | 91 | 9 | 836 | 80 |
| 2005 | 77 | 23 | 687 | 207 |
| 2006 | 90 | 10 | 884 | 98 |
| 2008 | 89 | 11 | 826 | 101 |
| 2009 | 92 | 8 | 757 | 70 |
| 2010 | 83 | 17 | 792 | 160 |
| 2018 | 66 | 34 | 671 | 346 |
| 2019 | 79 | 21 | 798 | 206 |
| 2020 | 81 | 19 | 794 | 184 |
| 2021 | 75 | 25 | 762 | 252 |

**5 Conclusion**

In this work we explored a temporal deep learning approach for bedfast/floating ice regime mapping from C-band SAR time-series and used it to study lake ice dynamics of Old Crow Flats, Yukon, Canada. Scenes covering the period from October to mid-March for each year of data were obtained from Sentinel-1, ERS-1 and 2, and RADARSAT-1 and were used to create an extensive annotated dataset of time-series each labeled as either floating ice, bedfast ice, or land based on the ice regime on the last day of the series. A temporal convolutional neural network (TempCNN) was trained and tested using the annotated dataset. The trained network, in turn, was employed to create lake ice maps of the Old Crow Flats covering a period of 29 years from 1992/1993 to 2020/2021. The created ice maps allowed to perform ice dynamics analysis.

This study demonstrates the potential of a temporal deep learning approach to lake ice mapping from C-band SAR. TempCNN offers a comprehensive automated classification framework suitable for different polarizations (HH and VV), SAR platforms, and years. Comparison of the proposed approach to a state-of-the-art thresholding algorithm (Duguay and Wang, 2019b) has shown the lake ice classification results to be very similar with each algorithm offering its advantages. While both methods allow to produce accurate lake ice maps, thresholding is simpler in implementation as it requires a single SAR scene. TempCNN, on the other hand, is complexed by heavier data requirements, but does not require a lake mask or incidence angle information. In addition, due to extensive training on deeper portions of larger lakes, TempCNN is better at avoiding misclassifications of floating ice as bedfast ice in deeper portions of larger lakes. Although both methods are applicable to VV and HH polarizations, the thresholding algorithm used for comparison was designed to work with S1 HH and VV imagery and its applicability to other SAR platforms remains unexplored; although Engram et al. (2018) obtained an overall accuracy of 93% using a threshold-based algorithm with ERS1/2, RADARSAT-2, Envisat, and S1 SAR imagery evaluated over seven lake-rich regions in Arctic Alaska. The present study has shown TempCNN to also be applicable to S1 (VV), ERS1/2 (VV), and R1 (HH), achieving a mean overall accuracy of 95% in the classification of bedfast ice, floating ice and land, and 99.6% in the classification of bedfast ice and floating ice (using the TempCNN model trained on 80% of the labelled dataset). The TempCNN-generated ice map accuracy was validated using a small set of field measurements and CLIMo simulated ice thickness, which partially explain the inter-annual lake ice dynamics of OCF. Aside from ice thickness, which is in part controlled by snow accumulation, variations in lake water levels can impact the inter-annual variability of the bedfast/floating ice regime.

The proposed approach could be improved by extensive OCF field data collection. More field observations of lake bathymetry, ice regimes, and vegetation cover changes could contribute to more accurate ice mapping and understanding of the ice dynamics underlying causes. In addition, the current temporal deep learning model learns only from the temporal component of the data. In future work, incorporation of the spatial component would be beneficial. Breaks in the data record also pose a challenge. More frequent SAR acquisitions possible with the combination of Sentinel-1 and the recently launched RADARSAT Constellation Mission (RCM) could help in this respect. Future analyses of the bedfast/floating ice regime will also benefit from data of the Surface Water and Ocean Topography (SWOT) mission (launched on December 16, 2022; https://swot.jpl.nasa.gov/), which will allow for higher accuracy mapping of water level of lakes than is currently possible from current radar altimetry missions. This comprehensive approach will help reduce the uncertainty of whether the ice regime change is associated with air temperature, snowfall, water level changes or a combination of these factors.

Over the period between 1992/1993 and 2020/2021, lake ice fraction analysis from the available 18 years of data suggests that bedfast ice fraction in OCF has increased by 11% despite a warming climate (Fig. 12). This counter-intuitive result is linked to the increased frequency of catastrophic lake drainages affecting OCF (Lantz and Turner 2015), which result in drastic decreases in water depths and exposed portions of lake bottoms in several large lakes that were associated with a transition to bedfast ice (Fig. 15 and 16). The partial refilling of drained basins, in some cases linked to the catastrophic drainage of adjacent

lakes, was associated with a transition to floating ice (Fig. 14). As permafrost may be sustained or aggrade under shallow water with a bedfast ice regime in OCF (Roy-Léveillée and Burn 2017), this transition may impact permafrost sustainability and talik development beneath OCF lakes (Heslop et al., 2015). Change detection analysis revealed an almost 13% net decrease in the extent of water in OCF. This change was associated with the catastrophic drainages of several large lakes (Lantz & Turner, 2015) and with the shrinking or desiccation of several small water bodies located near the periphery of OCF, where snowmelt is an important component of the water balance of the lakes (Turner et al., 2014), leaving them vulnerable to desiccation when snow runoff is low (Bouchard et al. 2013), as was the case in 2021. The decrease in water area detected is partly compensated by transition of land to water. More than half of the area affected transitioned to floating rather than bedfast ice, despite low snow precipitation in winter 2020-21. This is associated with rapid permafrost degradation at these new water areas, as the lake or pond bottom subsides following talik development and the melting of ground ice beneath water bodies with a floating ice regime (Roy-Léveillée & Burn, 2017). While part of this land to water transition is associated with the expansion of existing lakes via the erosion of thermokarst lake shores (Roy-Léveillée and Burn, 2010; Roy-Léveillée and Burn, 2015), many of the new water areas detected are spread in between lakes and may be associated with the initiation and expansion of ponds (Labrecque et al., 2009) which were not detected on the early imagery, due to their appearance during the study period or to their size in 1993 in relation to the spatial resolution of the data available for that year. The detected areas transitioning from water to land and land to water are yet larger than expected based on known rates and occurrences of lake initiation, growth, drainage or desiccation, and refilling, because no lake mask was used in this analysis, and the changes detected and compiled also reflect changes in the wetland's surface wetness and the convolutional neural network's ability to detect these changing conditions using different data types. This study provides a landscape-level perspective on the combined impacts of climatic warming, interannual variations in precipitation, and accelerated lake drainages on the extent of bedfast and floating ice in the OCF. The unexpected overall increase in the fraction of bedfast ice and rapid transition of floating to bedfast ice following catastrophic lake drainage will inform ongoing analyses of permafrost distribution, recovery, and sustainability in drained lake basins.

This work provides a strong baseline for future thermokarst lake-ice dynamics analysis, a topic of circumpolar relevance as thermokarst lowlands cover approximately 20% of the northern permafrost regions (Jones et al., 2002) and contain globally significant stores of soil organic carbon (Olefeldt et al., 2015). Documenting transitions between bedfast and floating ice is crucial to understanding permafrost dynamics beneath shallow lakes and drained lake basins, with potential impacts on methane ebullition and the regional carbon balance.

*Data availability:* The data including lake ice maps of Old Crow Flats, Yukon, Canada and the annotated dataset of lake ice backscatter time-series created by the authors are available on PANGAEA at: https://doi.org/10.1594/PANGAEA.947789 (Shaposhnikova et al., 2022) .

*Author contributions:* All authors contributed to the data collection, dataset creation, algorithm development, analysis and presentation of results equally.

*Acknowledgements:* The authors would like to thank a team from Yukon University (Nina Vogt, Louis-Philippe Roy, Caleb Charlie, and Cathy Koot) who collected field data used by this work in April of 2021 in OCF. The authors also thank Kevin Turner for providing a map of OCF vegetation. Finally, the authors are grateful to Nastaran Saberi for support throughout the project.

*Financial support:* This research has been supported by Natural Sciences and Engineering Research Council (NSERC) Alexander Graham Bell Canada Graduate Scholarship and NSERC PermafrostNet.

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
