# Peer review of "Bedfast and Floating Ice Dynamics of Thermokarst Lakes Using a Temporal Deep Learning Mapping Approach: Case Study of the Old Crow Flats, Yukon, Canada"

_EGUsphere, 2022_

## Referee Comment (RC2)

General comments:

This paper focused on the lake ice mapping in the Old Crow Flats using a temporal deep learning approach from the C-band SAR time series. Lake ice maps labeled as floating ice, bedfast ice, or land Flats were created from 1993 to 2021. The created ice-map dataset could be a reference for future ice dynamics analysis. So, it is an important and interesting issue. In this paper, however, the lake ice dynamics analysis was too simple. This work can be published after some minor revisions. However, some methods, expressions and the lake ice dynamic analysis need to be clarified before considering publication. Thus, I want to recommend that the paper be published after a few (minor) modifications.

Specific comments:

1. In the paper, the main purpose was to detect the bedfast ice and the floating ice. The two types ice was the top topic in this paper, so detail information on these is needed. In the introduction, the difference between bedfast ice and floating ice was not mentioned a lot. Please make a supplement to the bedfast ice and floating ice in the different effects on the lake and climate.

2. Could the TempCNN can recognize the lake from the land when there is no ice in the lake? In this paper, it is deemed the land and lake ice are seamlessly connected. Did ice fully cover lakes during the study period?

3. Lake ice simulation is another key work in your paper. But the process of the simulation was not clear. It seems unreasonable to make the parameters unchanged as the land in the northern part and southern part are different.

4. The figure captions are not so clear. Some lines and background lack details to make the figure hard to understand.

Technical corrections:

Table 1: The dataset is not consistent because of the lack of data from 1996/1997 to 1998/1999. Please give detailed information about it.

Figure 1. please point out what the different color areas represent.

Figure 3. what's the dashed red line represent for?

Line 140 "which cover the time period between 1992 to 2021."; line 14 "Canada over the 1993 to 2021 period"; Please unify the time of data for the paper: it's better to unify the time as 1992/1993 to 2020/2021.

Line 250: please make an explanation about the interpolation of the SAR stacks.

Line 268: "lake depth was specified as 2m", is it reasonable, as the average depth of lakes in OCF is 1.5m.

Line 422:" Performing change detection between the first (1993) and the last years of the dataset (2021), reveals a transition of 51 $km^2$ from bedfast to floating ice regime. However, 172 $km^2$ of floating ice shifted to a bedfast state." How to detect the exchange between the bedfast ice and floating ice?

Table 5: maybe you can add a "total ice area (bedfast ice + floating ice)" list to the table. The fraction trends analysis is better to company with an area trends analysis.

---

## Author Comment (AC1)

This paper used a temporal deep learning mapping approach for the purpose of classifying bedfast and floating ice in the Old Crow Flats. The authors have clearly done some work, but the manuscript needs major revision and in my opinion several aspects of the study need to be revisited before it can be considered for publication.

*We would like to thank the referee for valuable comments which have substantially helped improve the clarity and quality of the manuscript and stimulated interesting and constructive discussion.*

Major comments:

1. Your method is different from those used in previous studies (thresholding method or data mining approach), why did you not perform a comparison between them? If it works better, it is logical to proceed by presenting the method. The authors could do a more careful job comparing the proposed method with the approaches used previously by using the same dataset. Right now, there is not enough information to understand if the new method is necessary. There is almost no information about how the training and testing data sets were selected, what is the time stamps and amount of them?

*Thank you for a valuable comment.*

- *The goal of this manuscript was not to compare methods, but rather present a new method not previously used for lake ice mapping describing its benefits for the Old Crow Flats. The method presented in our paper could, however, be compared to other previously published approaches in a follow-up study either in the same or other study area(s). Although, this method is not necessarily better than other methods it does present advantages, namely 1) it takes advantage of the temporal evolution of ice backscatter over the entire lake ice season and is not reliant on a single value, as is the case for other methods; as new SAR platforms are being launched with higher quality, spatial resolution, and denser temporal coverage it is only reasonable to look for ways to make use of the wealth of available data; 2) wetland landscapes are dynamic in nature and the presented TempCNN method does not requires a lake mask, which is necessary for all the previously presented methods; 3) a well-trained TempCNN can be applied to various SAR sensors, both HH and VV polarizations, and different spatial resolutions. The accuracy of the method was assessed using a training set and the overall accuracy values fall in the range reported by other ice regime mapping solutions.*
- *The annotated dataset was created through visual assessment of the ice regime/land on the last day of the tie-series for each season. Labels were assigned based on the following three factors: 1) backscatter values, 2) value of the projected incidence angle of the SAR pulse (Claude Duguay's research team has done previous work on the relation of backscatter values and incidence angle values for the two ice types; Duguay and Wang, 2019 ), and 3) location of the pixel within the scene (inside or outside a lake; at the lake edges or lake center; in a larger or smaller lake).*
- *The amount of full and partial time stamps for each year is included in Table 1 (please, see below). Each year of data was further interpolated to 161 time stamps which corresponds to a daily frequency from October 4 to March 14. There is a total of 555 scenes and including dates for each year of data would be two lengthy. However, the full table can be found in the following thesis of the lead author: https://uwspace.uwaterloo.ca/handle/10012/17414.*

**Table 1. Data used in the project (I.A. stands for Incidence Angle); number of scenes is indicated for each year, where full indicates # of full coverage scenes of OCF.**

| Instrument | Year | Polarization | Imaging Mode | Data Product | Pixel Size (m) | Spatial Resolution (m) | I.A |
|---|---|---|---|---|---|---|---|
| Sentinel - 1 | 2020/2021 (28 full); 2019/2020 (41 full); 2018/2019 (37 full); 2017/2018 (36 full); | VV | IW (Interferometric Wide Swath) IW covers an area of 250 km | L2 RTC product | 10 (30 after RTC) | 5x20 | 20-45 |
| ERS-1/2 | 2009/2010 (32; 7 full); 2008/2009 (26; 7 full); 1995/1996 (52; 9 full); 1994/1995 (34; 6 full); 1993/1994 (25; 4 full); 1992/1993 (30; 6 full) | VV (central frequency 5.6 cm) | STD (SAR Imaging Mode; swath width of 100 km) | L1 amplitude CEOS image | 12.5 | 26x6 | 23 at mid-swath |
| RADARSAT-1 | 2007/2008 (24 full); 2005/2006 (16 full); 2004/2005 (21 full); 2003/2004 (24 full); 2002/2003 (32 full); 2001/2002 (45 full); 2000/2001 (22; 16 full); 1999/2000 (30 full) | HH (central frequency 5.6 cm) | ScanSAR wide (SWB, swath width of 450 km) | L1 amplitude CEOS image | 50 | 100x100 | 20-46 |

*Duguay, C.R. and J. Wang, 2019. Arctic-wide ground-fast lake ice mapping with Sentinel-1. ESA Living Planet Symposium, Milan, Italy, 13-17 May.*

2. The sentences that were used to describe the implementation of the deep learning mapping approach is not enough. This cannot be used to judge whether the deep learning model was implemented correctly. How to determinate the model parameter values? Did you compare the model performances when using different parameter values? What is the loss curve of your model? More details are needed here to evaluate your model.

*Thank you for a valuable comment. This work closely follows the TempCNN approach presented for land cover mapping using optical imagery by Pelletier et al., 2019. In terms of model parameters, the number of convolutional units for each convolutional layer was selected to be 64 based on temporal cross validation using 4, 8, 16, 32, 64, and 128 units (Please, refer to Figure 5). Batch size was adjusted to achieve best results. In addition, the number of training epochs is controlled by an early stopping technique with a patience of 10 epochs and a validation set comprising 5% of the training set.*

*The training and validation loss and accuracy for the final model used for creating lake ice maps is shown in the figure below.*

[Figure]

*Parameter selection is discussed in the subsection 3.3.3. TempCNN architecture Selection.*

*"Out of 18 years of data, one year was reserved for final testing, while the remaining 17 years were used to identify the best model structure utilizing a cross-validation procedure. Cross-validation allowed to determine the optimal number of convolutional units (4, 8, 16, 32, 64, or 128) while keeping other elements of the structure constant (Fig. 4). TempCNN with each number of units was trained on 16 years of data and tested on one year of data 17 times each time leaving a different year 235 out for testing. The overall accuracy summary of each of the six sets of experiments is shown in a box-plot form in Fig. 5. Based on the value of the median, interquartile range (IQR), length of the whiskers, and number of outliers, architecture with 64 convolutional units was identified as the best and was subsequently used for creating lake ice maps."*

*The rest of the parameters were kept the same as in the Pelletier et al., 2019 study, from which the TempCNN architecture was adopted. Further parameter testing will be done in future work.*

*Pelletier, C., Webb, G. I., and Petitjean, F.: Temporal convolutional neural network for the classification of satellite image time series, Remote Sens., 11, 1–25, 2019*

3. Several types of SAR images were used in your paper, but the effect of their differences (e.g., spatial resolution) was not considered in your model.

*Thank you for a valuable comment. The authors believe that being trained on data of different spatial resolutions, polarizations (HH and VV), and temporal resolution the TempCNN is capable of recognizing time-series from the three C-band platforms: ERS1/2, Sentinel 1, RADARSAT 1.*

4. The author still analyzed the lake ice dynamics. However, as you pointed out in the manuscript, the lake area is also changing, how to consider the effect of this on the lake ice dynamics analysis.

*Thank you for a valuable comment. Changing lake area definitely affects lake ice dynamics analysis. This is why in this work we strived to present an approach that does not need a new lake mask for each year and can capture the fluctuating lake boundaries. The reason behind using a lake mask for extracting lake ice fractions was to focus on the lake surface, rather than including all of the wetland. This will allow for future comparison to trends observed in other geographic areas of thermokarst lakes. Ideally a different lake mask had to be used for each year of data. However, due to the scope and timeframe of the project it was not feasible to create a different lake mask for each year of data. In order to establish a common baseline, the authors have decided to use a lake mask from fall of the most recent season in the dataset 2020/2021 (indication of the current status), while acknowledging that this definitely does lead to underestimation of the lake ice surface for years prior to major catastrophic drainages.*

5. Most of the figure citations are short and simple, more details should be listed as in that case the readers can easily understand the figures.

*Thank you for a valuable comment. We have revised the figure captions as follows:*
*"Figure 1. Old Crow Flats, Yukon, Canada. The background image is an RGB Landsat 8 of May 31, 2020, downloaded from USGS Earth Explorer (link: https://earthexplorer.usgs.gov/, accessed: July 4, 2021). Most lakes are still ice covered at this time of the year and appear white, open water surface of the river and smaller lakes, as well as some of the ice fringes appear black, tundra has a brownish shade, while areas of boreal forest appear dark green."*

*"Figure 2. Comparison of the three classes by sensor: (a) Sentinel-1; (b) RADARSAT-1; (c) ERS-1/2. Each class is represented by a mean and a standard deviation of a sample of 100 randomly selected pixels per sensor. Means and standard deviations are identified by solid and dashed lines, respectively: pink – floating ice; dark blue – bedfast ice; green – land."*

*"Figure 3. The two graphs illustrate application of 1-dimensional (1D) filters to time-series that can be used by convolutional layers of a TempCNN for extraction of temporal features. The red line represents original time-series, while the blue line denotes the filtered time-series: (a) a curve that resembles floating ice transformed by a gradient filter – the dashed red line indicates the origin of the filtered time series, where the value of the original series is increasing the filtered series has positive vales, and where the value of the original series is decreasing the filtered series has negative values; (b) a curve resembling floating ice transformed by a low-pass filter."*

*"Figure 5. The graph illustrates temporal cross validation results using box-plots. Each box-plot contains 17 overall accuracy values and if read from left to right each box-plot corresponds to 4, 8, 16, 32, 64, and 128 convolutional units in each convolutional layer. Red highlights the best architecture with 64 convolutional units."*

*"Figure 6. A sample of TempCNN classification output for OCF, Yukon, Canada: (a) Sentinel-1, March 2021; (b) RADARSAT-1, March 2004;(c) ERS, March 1993. Dark blue, light blue, and grey represent bedfast ice, floating ice, and land, respectively."*

*"Table 2. TempCNN overall classification accuracy for 15 experiments designed to test sensitivity of the network to removing certain years of data from the training set. Runs 1-5 correspond to the 20/80% split of the entire dataset, runs 6-10 were performed by training the network on 15 years of data and testing it on 3 each from a different sensor, runs 10-15 were carried out by training the network on 17 years of data and testing it on 1 year of data that was originally reserved and was not part of the cross validation procedure for determining the best architecture. Subsequently, mean accuracy for each set of 5 runs was calculated and finally mean of the three means is shown in the last row of the table."*

*"Figure 7. Matching TempCNN output with lake depths collected in 2000. Horizontal lines indicate CLIMo ice thickness predictions for taiga (0.72 m) and tundra (1.21 m) environments. RADARSAT-1 SAR 1999/2000 time-series were used for ice regime classification. The colour of points corresponds to the label assigned to each location by the TempCNN: dark blue – bedfast ice, pink – floating ice, green – land; while the shape corresponds to the surrounding vegetation: circle – tundra, triangle – taiga; square – mixed assigned based on the OCF vegetation map created by Turner et al., 2014."*

*"Figure 8. CLIMo simulated ice thickness for OCF: (a) simulation of taiga environment; (b) simulation of tundra environment. Dark blue represents ice thickness and grey stands for snow depth. The grey and dark blue points mark the condition on March 13 (last day of time-series) for each year."*

*"Table 3. Field data collection in the OCF in April 2009. For each of the ten locations, UTM coordinates, ice thickness, lake depth, and ice regime collected in the field are matched with TempCNN (bedfast, floating, land) and vegetation type (tundra or taiga)."*

*"Table 4. Field data collected in April 2021 on a small lake located beside the drained basin of Zelma Lake. For each of the four locations, UTM coordinates, ice regime, ice thickness, snow depth, and sediment temperature are matched with the TempCNN output."*

*"Figure 9. Husky Lake TempCNN predicted ice regime (1993-2021). Dark blue represents bedfast ice, light blue – floating ice, grey – land. Ice regime fluctuates between floating and bedfast depending on snow conditions, water level, and air temperature."*

*"Figure 12. Netro Lake TempCNN predicted ice regime (1993-2021). Dark blue represents bedfast ice, light blue – floating ice, grey – land. Between 1993 and 2021 ice regime has transitioned from mostly bedfast to mostly floating."*

*"Figure 13. Zelma Lake TempCNN predicted ice regime (1993-2021). Dark blue represents bedfast ice, light blue – floating ice, grey – land. You can notice a significant reduction in water surface area and a transition to mostly bedfast ice regime following the 2007 catastrophic drainage event."*

Specific comments:

Line 12 "less lakes are expected to develop bedfast ice", Line 504 "lake ice fraction analysis from the available 18 years of data suggests that bedfast ice fraction has increased by 11% despite a warming climate". Is there a contradiction between these two sentences?

*Thank you for a valuable comment. Line 12 reflects an expectation based on studies carried out in other thermokarst regions, while line 504 discusses the results observed in through this study, so there is no contradiction.*

*The climate of OCF is continental with cold winters (mean January temperature of -31.1°C) and warm summers (mean July temperature of 14.6°C) (Lantz and Turner, 2015). Continental climate and glaciolacustrine deposits make this area unique and different from other thermokarst lake zones.*

*Although the area is underlain by continuous ice-rich permafrost, unlike other regions, such as Mackenzie River Delta, even shallow bedfast-ice lakes display mean lake-bed temperatures above zero due to high summer air temperatures (Roy-Léveillée, 2014). Lakes surrounded by boreal forest tend to accumulate more snow, and therefore, develop thinner ice, while tundra lakes have a thinner, denser snow which results in ice grounding (Roy-Léveillée, 2014, Duguay et al., 2003). In light of this region's unique climate and sediment type, active thermokarst processes, and climate-driven vegetation changes, such as shrubification of the tundra area (Wang et al., 2020), the analysis carried out in this study make a valuable addition to the existing body of knowledge.*

*Lantz, T. C. and Turner, K. W.: Changes in lake area in response to thermokarst processes and climate in Old Crow Flats, Yukon, J. Geophys. Res. Biogeosci., 120, 513–524, 2015.*

*Roy-Léveillée, P.: Permafrost and thermokarst lake dynamics in the Old Crow Flats, northern Yukon, Canada, Ph.D. thesis, Carleton University, Ottawa, Canada, 2014.*

*Duguay, C. R., Flato, G. M., Jeffries, M. O., Ménard, P., Morris, K., and Rouse, W. R.: Ice-cover variability on shallow lakes at high latitudes: model simulations and observations. Hydrol. Process., 17, 3465–3483, 2003.*

*Wang, J. A., Sulla-Menashe, D., Woodcock, C. E., Sonnentag, O., Keeling, R. F., and Friedl, M. A.: Extensive land cover change across Arctic Boreal Northwestern North America from disturbance and climate forcing. Global Change Biology, 26, 807-822, 2020.*

Line 20-21, do you have any evidences to support this point?

*Thank you for a valuable comment. It has been shown by other researchers (Lantz and Turner, 2015) that drainage events are on the rise in the Old Crow Flats. Effect of fluctuations in water levels and winter snowfall on the ice thickness is discussed in, for instance (Surdu et al., 2014, Duguay et al., 2003). The discussion that the referee is referencing here suggests that based on the ice maps produced in this study, ice regime of the Old Crow Flats is influenced by the above three phenomena. For instance, partial lake drainages lead to a transition from mostly floating to mostly bedfast (e.g., Zelma Lake, Lake #1 and # 2 shown in Figure 10). Moreover, years with less snow depth (e.g., 2021) exhibit more wide-spread bedfast ice (Please, refer to Table 5).*

*Lantz, T. C. and Turner, K. W.: Changes in lake area in response to thermokarst processes and climate in Old Crow Flats, Yukon, J. Geophys. Res. Biogeosci., 120, 513–524, 2015.*

*Surdu, C. M., Duguay, C. R., Brown, L. C., and Fernández Prieto, D.: Response of ice cover on shallow lakes of the North Slope of Alaska to contemporary climate conditions (1950–2011): radar remote-sensing and numerical modeling data analysis, The Cryosphere, 8, 167–180, 2014.*

*Duguay, C. R., Flato, G. M., Jeffries, M. O., Ménard, P., Morris, K., and Rouse, W. R.: Ice-cover variability on shallow lakes at high latitudes: model simulations and observations. Hydrol. Process., 17, 3465–3483, 2003.*

Line 22-23, more bedfast ice?

*Thank you for a valuable comment. Catastrophic drainage events lead to loss of water surface area and shallowing of the remaining lake portion. As such, lake drainage is often accompanied by a transition from a mostly floating lake to a mostly bedfast lake.*

Line 31-32, here you said that a subsequent decrease in the number of bedfast ice lakes has been noted by many researchers. However, in Line 504 "lake ice fraction analysis from the available 18 years of data suggests that bedfast ice fraction has increased by 11% despite a warming climate".

*Thank you for a valuable comment.The decrease in the number of bedfast ice lakes has been noted in other thermokarst regions (e.g., Arctic Alaska). However, to the best of the authors' knowledge, no ice regime trend analysis has been performed for the Old Crow Flats. Based on the results of this study an increase in the bedfast ice fraction is observed.*

Line 40, as you said there are many extensive studies, you should compare your method to theirs.

*Thank you for a valuable comment. The goal of this manuscript was not to compare methods, but rather present a new method not previously used for lake ice mapping describing its benefits for the Old Crow Flats. The method presented in our paper could, however, be compared to other previously published approaches in a follow-up study either in the same or other study area(s). Although, this method is not necessarily better than other methods it does present advantages, namely 1) it takes advantage of the temporal evolution of ice backscatter over the entire lake ice season and is not reliant on a single value, as is the case for other methods; as new SAR platforms are being launched with higher quality, spatial resolution, and denser temporal coverage it is only reasonable to look for ways to make use of the wealth of available data; 2) wetland landscapes are dynamic in nature and the presented TempCNN method does not requires a lake mask, which is necessary for all the previously presented methods; 3) a well-trained TempCNN can be applied to various SAR sensors, both HH and VV polarizations, and different spatial resolutions. The accuracy of the method was assessed using a training set and the overall accuracy values fall in the range reported by other ice regime mapping solutions.*

Line 45, 'only THREE lakes shifted from bedfast to a floating ice regime', why?

*Thank you for a valuable comment. This line discusses the work presented in Arp et al., 2012. This study classifies lakes as either bedfast, floating, or intermittent (lakes with bedfast ice in one or more, but not all years) in the period between 2003 and 2011. Then the results were compared to 1980. The study reports that on the Outer Arctic Coastal Plain of northern Alaska, only three lakes bedfast in 1980 fully transitioned to a floating ice regime in 2003-2011 period, and two experience an opposite transition likely due to drainage events. Please, visit reference below for more information.*

*Arp, C. D., Jones, B. M., Lu, Z., and Whitman, M. S.: Shifting balance of thermokarst lake ice regimes across the Arctic Coastal Plain of northern Alaska, Geophys. Res. Lett., 39, 1–5, 2012.*

Lien 49, 'one of the study areas', it should be more specific here.

*Thank you for a valuable comment. The authors are referring to the Fish Creek region on the Inner Arctic Coastal Plain of northern Alaska. The sentence in the manuscript has a typo and has been modified as follows:*

*"Nonetheless, one of the study areas, namely Fish Creek region on the Inner Arctic Coastal Plain of northern Alaska, exhibited strong trends towards floating ice regimes."*

Line 53, why did you choose the Old Crow Flats as your study area, more reasons are needed.

*Thank you for a valuable comment. Old Crow Flats is a wetland of international significance underlain by continuous permafrost. This landscape contains over 2500 thermokarst lakes and changes in their ice regime have the potential to significantly influence the underlaying permafrost and subsequently the climate of the area. However, as has been mentioned in the "Study Area" section, ice regimes of Old*

*Crow Flats remain largely unexplored. While ice regimes of other thermokarst lake areas, such as northern Alaska, USA, Lena River Delta, Russia, and Hudson Bay, Canada, got significant attention from the research community, ice regimes of the Old Crow Flats require investigating.*

Line 61-61, there is a distinct backscatter patterns for floating and bedfast ice, thresholding method may be enough for the lake ice classification.

*Thank you for a valuable comment. Thresholding approach is indeed very useful and definitely wins over the proposed approach due to its simplicity. However, the remaining challenges of the thresholding approach are as follows: 1) need for a lake mask – which is likely different for each year of data; 2) misclassification of floating ice as bedfast ice in the deeper sections of lakes; 3) reliance on a single backscatter value (one date/one scene) that could be affected by speckle noise. In this work we have explored a temporal deep learning approach which allows to classify SAR images into three classes, eliminating the need for a lake mask. In addition, TempCNN allows to minimize misclassification in deeper portions of lakes by including such areas into the training set and allowing the network to become familiar with them. Finally, the proposed method classifies each pixel in the image based on the evolution of the backscatter throughout the ice season, and therefore is less affected by the salt and pepper effect of the SAR imagery.*

Line 70, a comparison to threshold-based classification is needed to prove that your method is necessary.

*Thank you for a valuable comment. The goal of this manuscript was not to compare methods, but rather present a new method not previously used for lake ice mapping describing its benefits for the Old Crow Flats. The method presented in our paper could, however, be compared to other previously published approaches in a follow-up study either in the same or other study area(s). Although, this method is not necessarily better than other methods it does present advantages, namely 1) it takes advantage of the temporal evolution of ice backscatter over the entire lake ice season and is not reliant on a single value, as is the case for other methods; as new SAR platforms are being launched with higher quality, spatial resolution, and denser temporal coverage it is only reasonable to look for ways to make use of the wealth of available data; 2) wetland landscapes are dynamic in nature and the presented TempCNN method does not requires a lake mask, which is necessary for all the previously presented methods; 3) a well-trained TempCNN can be applied to various SAR sensors, both HH and VV polarizations, and different spatial resolutions. The accuracy of the method was assessed using a training set and the overall accuracy values fall in the range reported by other ice regime mapping solutions.*

Line 78-79, this accuracy is comparable to your method. So you need to clearly point out the advantage of your method.

*Thank you for a valuable comment. The benefit of the proposed method consists in 1) no need for a lake mask, unlike all the other proposed methods; 2) applicability of the same model to different sensor platforms (different incidence angles and both HH and HV polarization); 3) reliance on a seasonal evolution rather than a single value, which could be easily affected by speckle noise.*

Line 85, what are the remaining challenges?

*Thank you for a valuable comment. The remaining challenges are as follows:*

1. *Overestimation of bedfast ice in the middle of deeper lakes, as the backscatter signature of floating ice tends to be lower in such areas.*
2. *Misclassifications due to the presence of wet snow.*
3. *Need for an annual lake mask as all the existing methods are able to classify only lake ice.*

4. *Need for a new threshold value for each sensor and each scene.*

Line 88-89, does your method still work well for these lakes with different phenomena or ice properties?

*Thank you for a valuable comment. Further investigation is needed to determine whether TempCNN can tackle presence of wet snow and water salinity. In this work, the last day of the time-series was selected in mid-March to avoid any occurrences of early melt. However, it is our expectation that presence of wet snow or pools of standing water would still present a challenge for TempCNN as it does for other methods. Most likely isolated instances of wet snow prior to the last day of the time-series would be classified correctly. Nonetheless, more extensive surface melt at the end of season would likely look too similar to ice grounding due to low backscatter values of both wet snow and pools of water on top of the ice. Water salinity on the other hand, can likely be tackled by TempCNN as long as brackish lakes are part of the training dataset, which is not the case for the Old Crow Flats and most thermokarst lake regions.*

Line 104, did the data for different sensors/with different resolutions/acquired in different years affect the method performance? How to consider these?

*Thank you for a valuable comment. A major component of this work was to create lake ice maps for the Old Crow Flats for the longest timeframe possible. The other important consideration was to have a method that would take advantage of the temporal dimension (in the light of the more recent SAR platforms supplying a wealth of regular high-quality SAR coverage) and that would allow to create lake ice maps using data from 3 different available platforms with varying spatial and temporal resolutions, and two different polarizations (HH and VV). The authors believe that the quality of the output maps matches the quality of the input SAR data.*

Line 116, so your method is mainly used for shallow lakes?

*Thank you for a valuable comment. That is correct. This study was mainly concerned with shallow thermokarst lakes, as landscapes underlain by permafrost are more sensitive to changes in lake ice regime. However, the authors believe that the proposed approach could be used for lakes of any size. As has been noted in the manuscript, deeper lakes often display darker (low backscatter) patches of floating ice in their centers. As such, more extensive training including such areas would be required to ensure that the TempCNN is familiar with the temporal patters specific to the deeper lakes.*

Line 125, dose the decreasing lake area affect the analysis of bedfast ice dynamics?

*Thank you for a valuable comment. Changing lake area definitely affects lake ice dynamics analysis. This is why in this work we strived to present an approach that does not need a new lake mask for each year and can capture the fluctuating lake boundaries. The reason behind using a lake mask for extracting lake ice fractions was to focus on the lake surface, rather than including all of the wetland. This will allow for future comparison to trends observed in other geographic areas of thermokarst lakes. Ideally a different lake mask had to be used for each year of data. However, due to the scope and timeframe of the project it was not feasible to create a different lake mask for each year of data. In order to establish a common baseline, the authors have decided to use a lake mask from fall of the most recent season in the dataset 2020/2021 (indication of the current status), while acknowledging that this definitely does lead to underestimation of the lake ice surface for years prior to major catastrophic drainages.*

Lien 128-129, 'Despite the overall trend of decreasing water surface area, most lakes are increasing in surface area'??

*Thank you for a valuable comment. The authors are referring to the fact that although the Old Crow Flats landscape is dominated by catastrophic drainage events leading to an overall decrease in water surface area (1951 – 2007; further research is needed for more recent years; Lantz and Turner, 2015), lakes are subject to thermo-mechanical erosion caused by wave action (more pronounced in tundra areas of the Old Crow Flats where shores are not protected by vegetation cover) leading to rapid shoreline recession over time (Roy-Léveillée and Burn, 2015).*

*Roy-Léveillée, P. and Burn, C. R.: Geometry of oriented lakes in Old Crow Flats, northern Yukon, in: Proceedings, 68th Canadian Geotechnical Conference and 7th Canadian Permafrost Conference, 21–23, Quebec City, QC, 2015.*

Figure 1, longitude and latitude are suggested to use, more detailed descriptions about this figure are needed.

*Thank you for a valuable comment. The Figure 1 caption has been updated as follows:*

*"Figure 1. Old Crow Flats, Yukon, Canada. The background image is an RGB Landsat 8 of May 31, 2020, downloaded from USGS Earth Explorer (link: https://earthexplorer.usgs.gov/, accessed: July 4, 2021). Most lakes are still ice covered at this time of the year and appear white, open water surface of the river and smaller lakes, as well as some of the ice fringes appear black, tundra has a brownish shade, while areas of boreal forest appear dark green."*

Line 143, how to consider this effect on your model?

*Thank you for a valuable comment. The time-series was purposefully selected to end in mid-March to ensure that no melt was occurring. As such, the lake ice maps are not affected by wet snow. However, presence of wet snow or pools of standing water remain an open problem for the bedfast ice mapping as both lead to a darker signature (signal absorption and reflection, respectively).*

Line 157-158, 'Adjusting the filter size allowed to account for the pixel size differences.'. According to the resolutions listed in Table1, this may be not true.

*Thank you for a valuable comment. We have selected the speckle filter size to ensure that it covers approximately the same area for the three different SAR platforms. Please see the table below:*

| Platform | Pixel size (m) | Filter size (pixels) | Area covered by each filter ($m^2$) |
|---|---|---|---|
| Sentinel 1 | 30 x 30 | 7 x 7 | 44,100 |
| ERS1/2 | 12.5 x 12.5 | 16 x 16 | 40,000 |
| RADARSAT 1 | 50 x 50 | 4 x 4 | 40,000 |

Line 161, again, how to consider the effect of data inconsistence on your classification results, which may further affect the ice dynamics analysis.

*Thank you for a valuable comment. This paper explores taking a temporal approach to lake ice classification from SAR. A temporal approach definitely requires a regular temporal coverage of the ice season to produce the best results possible. The aim of this work is to demonstrate the potential of the TempCNN. Going forward, higher quality, spatial, and temporal resolution SAR imagery is available through Sentinel-1 and the new RADARSAT Constellation Mission (RCM)*

*which will allow for training a TempCNN and applying the trained model for ongoing monitoring of lake ice dynamics in the future.*

Line 167, what are the spatial distributions of these labeled pixels?

*Thank you for a valuable comment. Land and floating ice labeled pixels were spread out over each scene as evenly as possible. Labelled pixels with bedfast ice, on the other hand, being less prevalent were less spread out. However, we made sure to include areas in both the norther and the southern parts of the Old Crow Flats, when possible, to include both tundra and boreal forest environments.*

Line 169, what is the detailed process of the visual assessment, how to deal with the mixed pixels?

*Thank you for a valuable comment. Pixels at the border between floating ice and bedfast ice, as well as where land meets lake ice were avoided during the labeling process. This was done to provide TempCNN with the clearest and most representative time-series patterns to learn to distinguish between the classes. Nonetheless, the trained TempCNN outputs not only a classified map, but also probability maps for each of the three classes (land, floating ice, bedfast ice), where the value of each pixel corresponds to the probability of this pixel to belong to a specific class. As such, in the areas of mixed pixels probability values for multiple classes are high, with the final class being assigned based on the class with the highest probability. Probability maps have not been used in this work's analysis. However, their utility and potential applications will be thoroughly explored in future research.*

Lien 175-176, this process should be more quantified?

*Thank you for a valuable comment. The authors have made utmost efforts to select representative areas of bedfast and floating lake ice, as well as land for each year of available data. Different years contained scenes with different incidence angles. In terms of incidence angles, we were guided by previous work done by Claude Duguay's research team (Duguay and Wang, 2019). Please take a look at the graph below.*

[Figure]

*Duguay, C.R. and J. Wang, 2019. Arctic-wide ground-fast lake ice mapping with Sentinel-1. ESA Living Planet Symposium, Milan, Italy, 13-17 May.*

Lien 181, why using early fall scenes to identify land areas?

*Thank you for a valuable comment. Labels (bedfast ice, floating ice, land) were assigned to the time-series based on the condition of the ice on the last day of the time-series (mid-March). However, both land and floating ice have similar SAR signature (high backscatter) at this time of the season. Hence, to avoid confusion between the two classes, during the labeling process the authors made sure to refer to the early October scenes where land and lake surface are more easily distinguishable. Higher contrast between the ice-covered lake surface and the surrounding land in early fall is explained by the fact that thin ice covering the lakes creates a mirror-like surface, leading to specular reflection of the SAR signal away from the sensor, and consequently lower backscatter (lakes appear darker). Later in the season as the floating ice thickens its backscatter increases due to high dielectric discontinuity and roughness of the ice-water interface. Please, see an example of an early fall scene used for identifying lake outlines during the labeling process (October 14, 2017, Sentinel 1) below. The greyscale corresponds to sigma nought(db) values. Lake surfaces appear black, while the surrounding land is grey.*

[Figure]

Line 183-184, how many data were obtained from the interpolation?

*Thank you for a valuable comment. The temporal deep learning model trained and used in this study works with time-series of a specific length. It takes a backscatter time-series as input and assigns it a specific class (bedfast ice, floating ice, or land) which corresponds to the state of the pixel on the last day of the time-series. Each time-series represents the backscatter value of a single pixel traced through time from October 4 to March 13. However, temporal resolution of SAR image stacks (# of scenes available throughout the ice season) varied significantly between years (please, refer to Table 1). As such, prior to inputting a SAR image stack into the TempCNN for classification, it had to be interpolated. The model was trained to work with time-series consisting of 161 time stamps which corresponds to a daily frequency from October 4 to March 13 – excluding February 29 for leap years. Imagine that each SAR image stack is a collection of time-series where each pixel is represented by a time-series of backscatter values starting from its backscatter value on October 4 and ending with a backscatter value on March 13 (Please, refer to portion of Figure 4 presented on the right for visual illustration). To ensure that each of the backscatter time-series for each year of data had the same length (161 values), linear interpolation was applied. Although the lake ice lifecycle is non-linear, previous studies have shown that more complex interpolation methods have little influence on classification accuracy (Pelletier et al., 2019, Valero et al., 2016). Linear interpolation was performed utilizing python programming language and the tools of pandas module. Interpolation not only filled the temporal gaps, but also replaced any missing or Not a Number (NaN) values, especially common for ERS1/2 and scene fringes. Interpolation was performed individually on every time-series (backscatter value of pixel traced through time). As a result, we obtained SAR image stacks consisting of 161 full coverage scenes, which were subsequently input into the TempCNN to perform classification.*

[Figure]

*The manuscript briefly explains time-series interpolation in Subsection 3.2 Annotated dataset creation:*

*"Resampling to a daily frequency and linear interpolation were applied to compensate for the temporal irregularity of the data gearing it for the deep learning classification (Pelletier et al., 2019; Valero et al., 2016). The final labeled time-series consisted of 161 time steps (i.e., one time step per day) covering the time period 185 between October 4 and March 13."*

*As per the referee's request, Subsection 3.4 Creation of ice regime maps using TempCNN was extended to include more details on interpolation of SAR stacks as follows:*

*"In order to transform SAR image stacks for each of the 18 years of data into lake ice regime maps using the trained TempCNN each stack had to be interpolated. Interpolation allowed to compensate for temporal resolution variability between different years such that each year's stack consisted of 161 scenes corresponding to a daily frequency from October 4 to March 13. Pixel-based linear interpolation was performed utilizing python programming language and the tools of pandas module. Although the lake ice lifecycle is non-linear, previous studies have shown that more complex interpolation methods have little influence on classification accuracy (Pelletier et al., 2019; Valero et al., 2016). Once the SAR stacks for 18 years were interpolated and each consisted of 161 scenes, the trained TempCNN model was used to create ice regime classification maps consisting of three classes: floating ice, bedfast ice, and land."*

*Pelletier, C., Webb, G. I., and Petitjean, F.: Temporal convolutional neural network for the classification of satellite image time series, Remote Sens., 11, 1–25, 2019.*

*Valero, S., Pelletier, C., and Bertolino, M.: Patch-based reconstruction of high resolution satellite image time series with missing values using spatial, spectral and temporal similarities, in: 2016 IEEE International Geoscience and Remote Sensing Symposium (IGARSS), 2308–2311, 2016.*

Line 189-190, how to consider this effect on your method?

*Thank you for a valuable comment. The pattern described in the lines mentioned by the referee forms the basis of the proposed method. Training the deep learning model to recognize these temporal patterns allows it to distinguish between floating and bedfast ice and create lake ice maps.*

Figure 2, the colors of two types of ice are difficult to classify.

*Thank you for a valuable comment. The colours have been updated, please, see below:*

[Figure]

[Figure]

[Figure]

Line 218-224, how to determinate the optimal parameter values for your model?

*Thank you for a valuable comment. Parameter selection is discussed in a subsequent section. Please, refer to the subsection 3.3.3. TempCNN architecture Selection:*

*"Out of 18 years of data, one year was reserved for final testing, while the remaining 17 years were used to identify the best model structure utilizing a cross-validation procedure. Cross-validation allowed to determine the optimal number of convolutional units (4, 8, 16, 32, 64, or 128) while keeping other elements of the structure constant (Fig. 4). TempCNN with each number of units was trained on 16 years of data and tested on one year of data 17 times each time leaving a different year 235 out for testing. The overall accuracy summary of each of the six sets of experiments is shown in a box-plot form in Fig. 5. Based on the value of the median, interquartile range (IQR), length of the whiskers, and number of outliers, architecture with 64 convolutional units was identified as the best and was subsequently used for creating lake ice maps."*

*The rest of the parameters were kept the same as in the Pelletier et al., 2019 study, from which the TempCNN architecture was adopted. Further parameter testing will be done in future work.*

*Pelletier, C., Webb, G. I., and Petitjean, F.: Temporal convolutional neural network for the classification of satellite image time series, Remote Sens., 11, 1–25, 2019.*

Line 231, this subtitle seems to be subordinate to Section 3.3.2 (TempCNN architecture).

*Thank you for a valuable comment. The authors prefer to leave this section as a separate subsection to avoid having the fourth level of subsections.*

Line 242, why THREE SETS were used here?

*Thank you for a valuable comment. The 15 experiments described in the subsection 3.3.4 TempCNN training and testing aim to test how sensitive the model is to inclusion or exclusion of certain years of data. We have used three different ways to split the data into training and testing sets: 1) a random split of all data points into 80% for training and 20% for testing; 2) three complete years (each from a different SAR platform) were left out for testing, 15 years of data were used for training; 3) training was done using 17 years of data and testing using data from the 2020/2021 ice season which was originally reserved for validation and not used when determining optimal neural network architecture.*

Line 250, which dataset was used to train the TempCNN model?

*Thank you for a valuable comment. Please, refer to Line 246: "The model trained on points from all years of the SAR dataset was selected for the ice regime mapping."*

Line 254, what elevation data was used here?

*Thank you for a valuable comment. The radiometric terrain corrected (RTC) Sentinel-1 imagery downloaded from Alaska Satellite Facility is supplemented by a digital elevation model (DEM) granule which was used in this study. Cell values indicate elevation in meters and have been resampled to a pixel spacing of 30 m by Alaska Satellite Facility. The DEM granule comes from the National Elevation Dataset at 2 arc seconds resolution (NED2) which is produced and distributed by USGS.*

Line 257, what is the spatial distribution of these field data.

*Thank you for a valuable comment. Figure below displays the spatial distribution of the field data. There are 51 measurement points in total.*

[Figure]

Line 262, does the Canadian Lake Ice Model have the enough resolution for method evaluation? Because the resolution of ERA5 is quite coarse.

*Thank you for a valuable comment. CLIMo does provide enough resolution. CLIMo can use data from either nearby meteorological stations (not available in this study) or atmospheric reanalysis data such as ERA5 to provide forcing atmospheric field (air temperature, wind speed, relative humidity, cloud cover and snowfall or depth).*

Line 268, why 2 m? in Line 116 you mentioned an averaged depth of 1.5 m.

*Thank you for a valuable comment. The change in depth could impact the freeze-up date by at most 1 day. However, the ice thickness simulated by CLIMo for the end of the season and used in this study will not be impacted by such a small change in depth (no difference in the break-up date).*

Line 272-274, figure or table are suggested to use here to help readers understand your opinion.

*Thank you for a valuable comment. Figure 7 and 8, as well as Tables 3 and 4 in the Results and Discussion section illustrate the analysis referred to in Lines 272-274.*

Line 284, the lake area is still changing, a single lake mask may include some uncertainties.

*Thank you for a valuable comment. The referee is absolutely right, using a single lake mask is not ideal as over the years the lake area is changing, and catastrophic lake drainages are occurring. However, due to the scope and timeframe of the project it was not feasible to create a different lake mask for each year of data. The use of lake mask for extracting bedfast and floating lake ice fractions are explained by the fact that the Old Crow Flats is a dynamic wetland where areas of standing water outside of the lakes are also covered by ice in winter and get classified as such. In order to establish a common baseline, the authors have decided to use a mask to extract bedfast/floating lake ice fractions, while acknowledging that this definitely does lead to underestimation of the lake ice surface for years prior to major catastrophic drainages.*

Line 292-293, what are the known lake drainage and refilling events?

*Thank you for a valuable comment.Lantz and Turner, 2015 report that between 1951 and 2007 the Old Crow Flats experience a decline of ~6000 ha in total lake area (but gained 232 lakes). 38 large lakes drained. Examples include lake Zelma (catastrophically drained in 2007) and lake Netro (drained around 1972). In addition, results of our work suggest that Lakes referred to as Lake #1 and #2 appear to have lost some of their surface area in the recent years. Moreover, result of our study suggest that Lake Netro has been refilling.*

*Lantz, T. C. and Turner, K. W.: Changes in lake area in response to thermokarst processes and climate in Old Crow Flats, Yukon, J. Geophys. Res. Biogeosci., 120, 513–524, 2015.*

Line 300, so did you apply any quality control strategies for the SAR images?

*Thank you for a valuable comment. All the SAR imagery has been properly processed and speckle filtered. In addition, the interpolation process not only filled the temporal gaps but also replaced any missing or Not a Number (NaN) values, especially common for ERS1/2 and scene fringes.*

Line 302, when S1 data are not accessible, what is the coverage of other data in the study area?

*Thank you for a valuable comment. All the available C-band SAR platforms have been explored. ERS offers data between 1992-2011, RADARSAT 1 1996 – 2008, and Sentinel 1 2014 – present. All years of data with at least two scenes for each month of the ice season were included in the dataset. RADARSAT 2 (2007 – present) data is not freely accessible, while EnviSAT (2002 – 2012) did not have a reasonable coverage for any given year within the study period.*

Line 316, 'the major features of the lifecycle can be lost', why?

*Thank you for a valuable comment. TempCNN classifies SAR stacks based on the shape of the time-series it has learned during the training process. Generally, land in the Old Crow Flats has little variability*

*during the ice season compared to the lake ice (please, take a look at the figure below). For Sentinel 1, which has regular temporal coverage a clear lifecycle or pattern emerges and is similar between different years of data. However, for ERS1/2 where temporal coverage is sparser and more variable for different years and interpolation is more extensive, pattern/lifecycle of the three classes becomes more "unique" for a specific year, making it harder to extract and learn a general pattern.*

[Figure]

Figure 6, is ice area in 2021 larger than that in 2004?

*The Old Crow Flats is a wetland environment and has areas of standing water outside of the lake boundaries which are classified as lake ice by TempCNN. As such, fluctuations of the landscape wetness from year to year result in some years appearing to have more lake ice than others, but could be shallow surface water rather than "lake surface".*

Table 2, more detailed descriptions are need.

*Thank you for a valuable comment. The description of Table 2 has been modified as follows:*

*"Table 2. TempCNN overall classification accuracy for 15 experiments designed to test sensitivity of the network to removing certain years of data from the training set. Runs 1-5 correspond to the 20/80% split of the entire dataset, runs 6-10 were performed by training the network on 15 years of data and testing it on 3 each from a different sensor, runs 10-15 were carried out by training the network on 17 years of data and testing it on 1 year of data that was originally reserved and was not part of the cross validation procedure for determining the best architecture. Subsequently, mean accuracy for each set of 5 runs was calculated and finally mean of the three means is shown in the last row of the table."*

Line 321-323, some possible explanations are suggested to add here.

*Thank you for a valuable comment. The manuscript has been updated as follows:*

*"The following pattern emerged when analyzing confusion matrices from all experiments: the 80/20% models had very minimal misclassifications equally distributed between the three classes; the models tested on three years had most significant misclassification of land as bedfast ice; and the models tested on one year of S1 had most significant misclassifications of floating ice as bedfast ice.* ***The better performance of the 80/20% model is believed to be attributed to the high inter-year temporal variability of available scenes. As such when time-series from each year of data are present in both the training and the test set higher accuracy is achieved.****"*

Lien 328, why the dark spots are particularly pronounced for R1 2001, ERS 1993, 1994, are they related to sensor?

*Thank you for a valuable comment. A further investigation is necessary in order to say with certainty what the nature of the dark spots is. As is mentioned in the manuscript "Some of the recent studies (e.g., Engram et al., 2020; Pointner and Bartsch, 2020) suggest that the dark spots pattern could be a result of local ice thinning or complete melt caused by methane ebullition." The authors believe that it is not related to the sensor, as the patter appears clearly in most lakes, but not outside of them and the backscatter values could represent open water. Future research is definitely necessary to answer the referee's question and discover why the patter appeared in specific years only.*

Line 340-341, did you use the optical images in this study? Do you mean that your training dataset may contain erroneous labeling?

*Thank you for a valuable comment. The optical imagery was not used for the analysis and is not part of the dataset. However, Landsat imagery was occasionally consulted throughout the labeling process. As to the "erroneous labeling", authors are referring to the fact that field observations of ice regimes in Old Crow Flats are very scares and a more extensive field work would be beneficial to validate the ice regime patterns observed via SAR imagery analysis only. Moreover, as any dataset created manually, human error cannot be completely eliminated. In order to avoid the confusion, the sentence has been modified as follows:*

*"As such, due to the lack of ground truthing, the labeled dataset used for model training was created from visual interpretation of SAR imagery and, therefore, could be subject to human error."*

Line 381-392, how can the ice thickness data be used to evaluate the lake ice classification results?

*Thank you for a valuable comment. Ice thickness measured in the field was accompanied by lake depth measurement (please, refer to Table 3) or a record of ice regime (floating/bedfast) (please, refer to Table 4). Where ice thickness is equal to the lake depth, lake ice regime is bedfast, whereas in areas where ice thickness is less than the lake depth, lake ice regime is floating. This way, using ice thickness and lake depth or a direct field record of lake ice regime can be compared to the classification output of the TempCNN.*

Line 1.29, what about the ice whose thickness between 1.14 and 1.29 m?

*Thank you for a valuable comment. The aim of the discussion that the reviewer is referring to is to draw general parallels between CLIMo predicted ice thickness for the tundra area and the ice regime classification output by TempCNN for the Husky Lake (~ 68.33º N, 140.17º W). Below we are including a table which shows CLIMo simulated ice thickness for each of the years in the dataset along with the TempCNN ice regime class for the majority of the Husky Lake surface. Looking at the table below you can notice that for years where CLIMo predicted ice thickness is 1.08-1.14 m (thinner ice) the TempCNN classified ice regime is floating (highlighted in blue). For years where CLIMo simulated ice thickness is 1.29-1.40 m (thicker ice), TempCNN ice regime is bedfast (except for 1996) (highlighted in black), while the for the rest of the years with ice thickness 1.19 – 1.23 m the TempCNN ice regime fluctuates between floating and bedfast. It should be noted that CLIMo has a general uncertainty of ~ 0.05-0.06 m (Duguay and Lafleur, 2003).*

| Year | 1993 | 1994 | 1995 | 1996 | 2000 | 2001 | 2002 | 2003 | 2004 |
|---|---|---|---|---|---|---|---|---|---|
| CLIMo predicted ice thickness (meters) | 1.22 | 1.23 | 1.23 | 1.32 | 1.21 | 1.14 | 1.35 | 1.29 | 1.21 |
| TempCNN Ice regime | bedfast | floating | bedfast | floating | bedfast | floating | bedfast | bedfast | floating |
| Year | 2005 | 2006 | 2008 | 2009 | 2010 | 2018 | 2019 | 2020 | 2021 |
| CLIMo predicted ice thickness | 1.21 | 1.19 | 1.33 | 1.22 | 1.22 | 1.14 | 1.08 | 1.40 | 1.39 |
| TempCNN Ice regime (meters) | floating | bedfast | bedfast | bedfast | floating | floating | floating | bedfast | bedfast |

*Duguay, C. R. and Lafleur, P. M.: Determining depth and ice thickness of shallow sub-Arctic lakes using space-borne optical and SAR data, Int. J. Remote Sens., 24, 475–489, 2003.*

Line 474-475, does the changing lake area affect the ice dynamics analysis?

*Thank you for a valuable comment. Changing lake area definitely affects lake ice dynamics analysis. This is why in this work we strived to present an approach that does not need a new lake mask for each year and can capture the fluctuating lake boundaries. The reason behind using a lake mask for extracting lake ice fractions was to focus on the lake surface, rather than including all of the wetland. This will allow for*

*future comparison to trends observed in other geographic areas of thermokarst lakes. Ideally a different lake mask had to be used for each year of data. However, due to the scope and timeframe of the project it was not feasible to create a different lake mask for each year of data. In order to establish a common baseline, the authors have decided to use a lake mask from fall of the most recent season in the dataset 2020/2021 (indication of the current status), while acknowledging that this definitely does lead to underestimation of the lake ice surface for years prior to major catastrophic drainages.*

Line 482-483, how to reduce the effect from spatial resolution difference?

*Thank you for a valuable comment.*

*The authors are referring to the fact that in courser spatial resolution imagery (e.g., R1) smaller bedfast ice areas are likely to be missed due to larger pixels and signal dilution by the surrounding land/floating ice. Nothing can be done about this; courser spatial resolution will always lead to some information loss. However, going forward, higher spatial and temporal resolution C-band SAR (Sentinel-1, RADARSAT Constellation Mission (RCM)) and TempCNN method could be advantages.*

Table 5, how many areas of floating ice transferred to bedfast ice? How many areas of bedfast ice→ floating ice, ice→ land, land→ ice? You can add this information into this table.

Thank you for a valuable comment. Unfortunately, we did not understand the question quite clearly. There could be many (thousands) areas that transitioned from floating to bedfast ice and vice versa. Based on reference to Table 5 we think that the referee likely meant change in square km, we will add this information to the revised manuscript.

Line 499-500, so the accuracy of lake ice maps in different periods are variable. Does this affect the final ice dynamics analysis?

*Thank you for a valuable comment. To avoid confusion, the following sentence has been removed:*

**

*The accuracy assessment has been done for all the points in the dataset and not for different instruments separately. As such, we cannot definitively say if accuracy of one of the platforms is lower than the others. Overall accuracy for all the three platforms together is quite high and comparable to the results achieved by other methods. In lines 499-500 we were refereeing to the fact that ERS 1/2 had a much lower temporal coverage than the other two platforms, and a lot of its scenes did not cover the Old Crow Flats in their entirety. As such, we assume that accuracy of the ERS lake ice maps might be affected. Nonetheless, overall accuracy for the annotated dataset is high and comparable to other studies. Moreover, this study is the first to investigate a temporal approach to lake ice mapping and further investigation is definitely needed to determine the temporal sampling necessary to achieve optimal results.*

Line 505-507, more evidences are needed here.

*Thank you for a valuable comment. The manuscript has been modified as follows:*

*"Over the period between 1993 and 2021, lake ice fraction analysis from the available 18 years of data suggests that bedfast ice fraction has increased by 11% despite a warming climate (Fig. 10). This counter-intuitive result is linked to the increased frequency of catastrophic lake drainages affecting OCF (Lantz and Turner 2015), which result in drastic decreases in water depths and exposed portions of lake*

*bottoms in several large lakes that were associated with a transition to bedfast ice (Fig. 13, 14). The partial refilling of drained basins, in some cases linked to the catastrophic drainage of adjacent lakes (Fig. 12), was associated with a transition to floating ice (Fig. 12). As permafrost may be sustained or aggrade under shallow water with a bedfast ice regime in OCF (Roy-Léveillée and Burn 2017), this transition is likely to have implications for permafrost sustainability and talik development beneath OCF lakes, with potential implications for methane ebullition and the regional carbon budget (Helsop et al., 2015)."*

---

## Author Comment (AC2)

General comments:
This paper focused on the lake ice mapping in the Old Crow Flats using a temporal deep learning approach from the C-band SAR time series. Lake ice maps labeled as floating ice, bedfast ice, or land Flats were created from 1993 to 2021. The created ice-map dataset could be a reference for future ice dynamics analysis. So, it is an important and interesting issue. In this paper, however, the lake ice dynamics analysis was too simple. This work can be published after some minor revisions. However, some methods, expressions and the lake ice dynamic analysis need to be clarified before considering publication. Thus, I want to recommend that the paper be published after a few (minor) modifications.

*We would like to thank the referee for valuable comments which have substantially helped improve the clarity and quality of the manuscript and stimulated interesting and constructive discussion. \*Please, refer to the supplement for all tables and figures.*

Specific comments:

In the paper, the main purpose was to detect the bedfast ice and the floating ice. The two types ice was the top topic in this paper, so detail information on these is needed. In the introduction, the difference between bedfast ice and floating ice was not mentioned a lot. Please make a supplement to the bedfast ice and floating ice in the different effects on the lake and climate.

*Thank you for a valuable comment. The introduction section has been modified as follows to include a more detailed description of the impact of bedfast and floating ice presence on the permafrost:*

*"Many shallow arctic lakes and ponds of thermokarst origin freeze to bed in the winter months, allowing lake-bottom temperatures to drop below 0°C and frost to penetrate the lake bottom sediment. Permafrost is sustained beneath the lake bottom where the freezing-degree-days at the ice-sediment interface are sufficient to counterbalance the thawing that takes place while lake-bottom temperatures are above 0°C (Roy-Léveillée and Burn, 2017). Where lake bottom conditions are too warm to sustain permafrost, for instance where ice does not reach the lake bottom or where the period of ice contact is brief, permafrost will degrade and a bulb of unfrozen ground or talik will develop and expand beneath the lake bottom. Such talik development contributes to positive feedbacks as it promotes lake deepening via subsidence of the lake bottom (Roy-Léveillée and Burn 2016), further reducing the occurrence of bottom-fast ice, and increases the ebullition of potent greenhouse gases such as methane from the thawing and decomposition of organic matter beneath the lake bottom (Arp et al., 2012; Engram et al., 2020). However, lake ice thinning and a subsequent decrease in the extent and duration of bedfast ice lakes has been noted by many researchers (Engram et al., 2018; Labrecque et al., 2009; Surdu et al., 2014). Hence, monitoring and quantifying thermokarst lake ice dynamics is critical for understanding changes in sub-lake permafrost stability and expected changes in methane ebullition patterns in thermokarst lowlands. Bedfast ice mapping, in particular, has a variety of other applications, including climate monitoring (Arp et al., 2012), permafrost studies (Arp et al., 2011), bathymetric mapping (Duguay and Lafleur, 2003; Kozlenko and Jeffries, 2000), overwintering fish habitat (Brown et al., 2010), and winter water withdrawal (Hirose et al., 2008; Jeffries et al., 1996)."*

*Roy-Léveillée, P. and Burn, C. R.: A modified landform development model for the topography of drained thermokarst lake basins in fine-grained sediments. Earth Surface Processes and Landforms, 41, 1504-1520, 2016. DOI: 10.1002/esp.3918*

Could the TempCNN can recognize the lake from the land when there is no ice in the lake? In this paper, it is deemed the land and lake ice are seamlessly connected. Did ice fully cover lakes during the study period?

*Thank you for a valuable comment. Generally speaking, land surface appears bright (high backscatter values) in SAR imagery due to roughness of its surface and vegetation volume scattering and open water surface is dark most of the time (unless it is roughened by wind) (Huang et al., 2018, Duguay and Lafleur, 2003). As such, many studies have successfully explored water-body segmentation using SAR mainly using machine learning and deep learning techniques (Guo et al., 2022). To the best of the authors knowledge, no studies have applied temporal deep learning to this task. However, applying TempCNN to extracting water bodies would probably be unnecessarily complex, due to the need to create an extensive labelled dataset.*

*On the other hand, mapping bedfast and floating ice in presence of open water (cracks in the ice or local melt caused by methane ebullition (Engram et al., 2020) or temporarily warming) is a challenge, as both open water and bedfast ice are characterized by low backscatter (Duguay and Lafleur, 2003). In this work, the time-series for each year of data tracked backscatter from early October to mid-March, and the final classification for each year was representative of the state of ice on the last day of the time-series (mid-March). Although, some open water was likely present in early October, mid-March end date was selected specifically to avoid open water presence (based on air temperature information).*

*Huang, W., DeVries, B., Huang, C., Lang, M. W., Jones, J. W., Creed, I. F., and Carroll, M. L.: Automated extraction of surface water extent from Sentinel-1 data, Remote Sens., 10, 797, 2018.*

*Duguay, C. R. and Lafleur, P. M. (2003). Determining depth and ice thickness of shallow sub-Arctic lakes using space-borne optical and SAR data. International Journal of Remote Sensing, 24(3):475-489.*

*Guo Z, Wu L, Huang Y, Guo Z, Zhao J, Li N. Water-Body Segmentation for SAR Images: Past, Current, and Future. Remote Sensing. 2022; 14(7):1752. https://doi.org/10.3390/rs14071752*

*Engram, M., Anthony, K. M. W., Sachs, T., Kohnert, K., Serafimovich, A., Grosse, G., and Meyer, F.: Remote sensing northern lake methane ebullition, Nat. Clim. Chang., 10, 511–517, 2020.*

Lake ice simulation is another key work in your paper. But the process of the simulation was not clear. It seems unreasonable to make the parameters unchanged as the land in the northern part and southern part are different.

*Thank you for a valuable comment. The CLIMo simulation results were used to access the accuracy of lake ice maps. The CLIMo simulation was run for two scenarios: taiga and tundra to capture the fact that northern part of Old Crow Flats is characterized by polygonal tundra, while the southern part has subarctic boreal forest. Parameters were set as follows: taiga – snow depth 100%, snow density 175 kg/m3; tundra - snow depth 50%, snow density 300 kg/m3. Please refer to subsection 3.5 Accuracy assessment. The snow depth and density values used for the taiga vs tundra simulations are typical to those documented in other studies (Duguay et al., 2003; Sturm and Liston, 2003).*

*Duguay, C. R., Flato, G. M., Jeffries, M. O., Ménard, P., Morris, K., and Rouse, W. R.: Ice-cover variability on shallow lakes at high latitudes: model simulations and observations, Hydrol. Process., 17, 3465–3483, 2003.*

*Sturm, M. and Liston, G. E.: The snow cover on lakes of the Arctic Coastal Plain of Alaska, USA., J. Glaciol., 49, 370–380, 2003.*

The figure captions are not so clear. Some lines and background lack details to make the figure hard to understand.

*Thank you for a valuable comment. The figure captions have been revised and more detailed descriptions have been added to make figures more reader friendly as shown below.*

[revised manuscript text omitted]

Technical corrections:

Table 1: The dataset is not consistent because of the lack of data from 1996/1997 to 1998/1999. Please give detailed information about it.

*Thank you for a valuable comment. In order to build a reliable time-series a more or less even coverage throughout the ice season is necessary. 18 years of data were chosen for this study as they offered a minimum of two scenes for each month throughout the ice season as is mentioned in subsection 3.1 SAR imagery. ERS-1/2 availability was quite limited and seasons 1996/1997 1997/1998, and 1998/1999 had either none or an insufficient number of scenes available.*

Figure 1. please point out what the different color areas represent.

*Thank you for a valuable comment. Figure 1 caption has been updated as follows:*

*"Figure 1. Old Crow Flats, Yukon, Canada. The background image is an RGB Landsat 8 of May 31, 2020, downloaded from USGS Earth Explorer (link: https://earthexplorer.usgs.gov/, accessed: July 4, 2021). Most lakes are still ice covered at this time of the year and appear white, open water surface of the river and smaller lakes, as well as some of the ice fringes appear black, tundra has a brownish shade, while areas of boreal forest appear dark green."*

Figure 3. what's the dashed red line represent for?

*Thank you for a valuable comment. Figure 3 caption has been updated as follows:*

*"Figure 3. The two graphs illustrate application of 1-dimensional (1D) filters to time-series that can be used by convolutional layers of a TempCNN for extraction of temporal features. The red line represents original time-series, while the blue line denotes the filtered time-series: (a) a curve that resembles floating ice transformed by a gradient filter – the dashed red line indicates the origin of the filtered time series, where the value of the original series is increasing the filtered series has positive values, and where the value of the original series is decreasing the filtered series has negative values; (b) a curve resembling floating ice transformed by a low-pass filter."*

Line 140 "which cover the time period between 1992 to 2021."; line 14 "Canada over the 1993 to 2021 period"; Please unify the time of data for the paper: it's better to unify the time as 1992/1993 to 2020/2021.

*Thank you for a valuable comment. The time period has been made consistent throughout the manuscript.*

Line 250: please make an explanation about the interpolation of the SAR stacks.

*Thank you for a valuable comment. The temporal deep learning model trained and used in this study works with time-series of a specific length. It takes a backscatter time-series as input and assigns it a specific class (bedfast ice, floating ice, or land) which corresponds to the state of the pixel on the last day of the time-series. Each time-series represents the backscatter value of a single pixel traced through time from October 4 to March 13. However, temporal resolution of SAR image stacks (# of scenes available throughout the ice season) varied significantly between years (please, refer to Table 1). As such, prior to inputting a SAR image stack into the TempCNN for classification, it had to be interpolated. The model was trained to work with time-series consisting of 161 time stamps which corresponds to a daily frequency from October 4 to March 13 – excluding February 29 for leap years. Imagine that each SAR image stack is a collection of time-series where each pixel is represented by a time-series of backscatter values starting from its backscatter value on October 4 and ending with a backscatter value on March 13 (Please, refer to portion of Figure 4 presented on the right for visual illustration). To ensure that each of the backscatter time-series for each year of data had the same length (161 values), linear interpolation was applied. Although the lake ice lifecycle is non-linear, previous studies have shown that more complex interpolation methods have little influence on classification accuracy (Pelletier et al., 2019, Valero et al., 2016). Linear interpolation was performed utilizing python programming language and the tools of pandas module. Interpolation not only filled the temporal gaps, but also replaced any missing or Not a Number (NaN) values, especially common for ERS1/2 and scene fringes. Interpolation was performed individually on every time-series (backscatter value of pixel traced through time). As a result, we obtained SAR image stacks consisting of 161 full coverage scenes, which were subsequently input into the TempCNN to perform classification.*

[Figure]

*The manuscript briefly explains time-series interpolation in Subsection 3.2 Annotated dataset creation:*

*"Resampling to a daily frequency and linear interpolation were applied to compensate for the temporal irregularity of the data gearing it for the deep learning classification (Pelletier et al., 2019; Valero et al., 2016). The final labeled time-series consisted of 161 time steps (i.e., one time step per day) covering the time period 185 between October 4 and March 13."*

*As per the referee's request, Subsection 3.4 Creation of ice regime maps using TempCNN was extended to include more details on interpolation of SAR stacks as follows:*

*"In order to transform SAR image stacks for each of the 18 years of data into lake ice regime maps using the trained TempCNN each stack had to be interpolated. Interpolation allowed to compensate for temporal resolution variability between different years such that each year's stack consisted of 161 scenes corresponding to a daily frequency from October 4 to March 13. Pixel-based linear interpolation was performed utilizing python programming language and the tools of pandas module. Although the lake ice lifecycle is non-linear, previous studies have shown that more complex interpolation methods have little influence on classification accuracy (Pelletier et al., 2019; Valero et al., 2016). Once the SAR stacks for 18 years were interpolated and each consisted of 161 scenes, the trained TempCNN model was used to create ice regime classification maps consisting of three classes: floating ice, bedfast ice, and land."*

*Pelletier, C., Webb, G. I., and Petitjean, F.: Temporal convolutional neural network for the classification of satellite image time series, Remote Sens., 11, 1–25, 2019.*

*Valero, S., Pelletier, C., and Bertolino, M.: Patch-based reconstruction of high resolution satellite image time series with missing values using spatial, spectral and temporal similarities, in: 2016 IEEE International Geoscience and Remote Sensing Symposium (IGARSS), 2308–2311, 2016.*

Line 268: "lake depth was specified as 2m", is it reasonable, as the average depth of lakes in OCF is 1.5m.

*Thank you for a valuable comment. The change in depth could impact the freeze-up date by at most 1 day. However, the ice thickness simulated by CLIMo for the end of the season and used in this study will not be impacted by such a small change in depth (no difference in the break-up date).*

Line 422:" Performing change detection between the first (1993) and the last years of the dataset (2021), reveals a transition of 51 km2 from bedfast to floating ice regime. However, 172 km2 of floating ice shifted to a bedfast state." How to detect the exchange between the bedfast ice and floating ice?

*Thank you for a valuable comment. The ice regime maps created by TempCNN for each year of the available data allow to detect the exchange between the two ice regime classes using the following formula: ("class on date 1" *10) + "class on date 2". The resulting GIS layer of data will be a raster where each cell contains a two-digit number with the first digit being the class on date 1 and the second digit being the class on date 2. Figure 10 is an example of a change detection between 1993 and 2021. Using the ice maps created in this study the change detection can be performed between any two years of data.*

Table 5: maybe you can add a "total ice area (bedfast ice + floating ice)" list to the table. The fraction trends analysis is better to company with an area trends analysis.

*Thank you for a valuable comment. Total ice area has been added to Table 5 as is shown below. However, the authors prefer not to include it into the manuscript as the ice fraction analysis were done*

*using a lake mask from October 2020 and as such the total ice area for the years prior to 2020 is likely underestimated.*

**Table 5. Bedfast and floating lake ice fractions and area (km²) in OCF from 1993 to 2021. A lake mask created using 2020/2021 lake extent was used to extract lake ice fractions.**

| **Year** | Floating ice fraction (%) | Bedfast ice fraction (%) | Floating ice area (km²) | Bedfast ice area (km²) | Total ice area (km²) |
|---|---|---|---|---|---|
| 1993 | 88 | 12 | 790 | 108 | 898 |
| 1994 | 86 | 14 | 779 | 127 | 906 |
| 1995 | 88 | 12 | 848 | 111 | 959 |
| 1996 | 92 | 8 | 909 | 82 | 991 |
| 2000 | 90 | 10 | 825 | 93 | 918 |
| 2001 | 81 | 19 | 753 | 175 | 928 |
| 2002 | 86 | 14 | 796 | 134 | 930 |
| 2003 | 79 | 21 | 746 | 203 | 949 |
| 2004 | 91 | 9 | 836 | 80 | 916 |
| 2005 | 77 | 23 | 687 | 207 | 894 |
| 2006 | 90 | 10 | 884 | 98 | 982 |
| 2008 | 89 | 11 | 826 | 101 | 927 |
| 2009 | 92 | 8 | 757 | 70 | 827 |
| 2010 | 83 | 17 | 792 | 160 | 952 |
| 2018 | 66 | 34 | 671 | 346 | 1017 |
| 2019 | 79 | 21 | 798 | 206 | 1004 |
| 2020 | 81 | 19 | 794 | 184 | 978 |
| 2021 | 75 | 25 | 762 | 252 | 1014 |

---

## Author Comment (AC3)

Thank You for the interesting topic and manuscript. The manuscript will still require reasoning and clarification of the applied methodology and presentation of the results. Therefore, I propose a major revision.

*We would like to thank the referee for valuable comments which have substantially helped improve the clarity and quality of the manuscript and stimulated interesting and constructive discussion. \*Please, refer to the supplement for all tables and figures.*

Major comments:

My major comments are related to the applied methodology and presentation.

1) As pointed out by the other reviewers, the input data comes from different instruments and they even have different polarizations. The effect of this should be analyzed. Could e.g. the instrument type be fed into the NN as one additional input, or separate NN's used for different instruments? How much would the result improve by taking separate instrument into account (if any)? Also some kind of analysis of the backscattering of the separate instruments for the lake ice (and surrounding land) would be interesting (how do they differ or are they very similar statistically).

*Thank you for a valuable comment. The goal of this study was to present a method capable of working with different C-band SAR platforms and different polarizations. One of the strengths of a deep learning approach is the ability of the network to be trained on different types of data, learn all of its different aspects, and then be able to recognize and classify them correctly. It has been shown by Claude Duguay's research team (Duguay and Wang, 2019) that HH and VV backscatter of floating and bedfast ice is comparable and similar enough. Please, refer to the graph below.*

*In addition, as you can see in the three graphs below, although different instruments do have differences, the backscatter patterns for bedfast ice, floating ice, and land are quite comparable between the three platforms.*

*Duguay, C.R. and J. Wang, 2019. Arctic-wide ground-fast lake ice mapping with Sentinel-1. ESA Living Planet Symposium, Milan, Italy, 13-17 May.*

[Figure]

[Figure]

2) Selection of TempCNN as a method and parametrization TempCNN was selected to be used but it would be useful to compare the performance of TempCNN to some simple method (thresholding) to give evidence that it performs better (ow much better?). Also more detailed reasoning of the selected structure would be useful to be included, e.g. why there are just three convolutional layers etc.? The selections could also be reasoned by referring to publications where the selections have been justified.

*Thank you for a valuable comment.*

- *The goal of this manuscript was not to compare methods, but rather present a new method not previously used for lake ice mapping describing its benefits for the Old Crow Flats. The method presented in our paper could, however, be compared to other previously published approaches in a follow-up study either in the same or other study area(s). Although, this method is not necessarily better than other methods it does present advantages, namely 1) it takes advantage of the temporal evolution of ice backscatter over the entire lake ice season and is not reliant on a single value, as is the case for other methods; as new SAR platforms are being launched with higher quality, spatial resolution, and denser temporal coverage it is only reasonable to look for ways to make use of the wealth of available data; 2) wetland landscapes are dynamic in nature and the presented TempCNN method does not requires a lake mask, which is necessary for all the previously presented methods; 3) a well-trained TempCNN can be applied to various SAR sensors, both HH and VV polarizations, and different spatial resolutions. The accuracy of the method was assessed using a training set and the overall accuracy values fall in the range reported by other ice regime mapping solutions.*

- *This work closely follows the TempCNN approach presented for land cover mapping using optical imagery by Pelletier et al., 2019. In terms of model parameters, the number of convolutional units for each convolutional layer was selected to be 64 based on temporal cross validation using 4, 8, 16, 32, 64, and 128 units (Please, refer to Figure 5). Batch size was adjusted to achieve best results. In addition, the number of training epochs is controlled by an early stopping technique with a patience of 10 epochs and a validation set comprising 5% of the training set.*

*Pelletier, C., Webb, G. I., and Petitjean, F.: Temporal convolutional neural network for the classification of satellite image time series, Remote Sens., 11, 1–25, 2019.*

3) Use of data, division to training and test data sets: It should be better reasoned why the model training setup was such complicated. On what are the selections and divisions based? Cross validation is a good way to train and test if there is little data. If You want to continue the time series (with the same training) it would also be good to have a training with

good generalization property. In any case, give reasoning for the use of the data sets and division to training and test. Would a more simple approach be feasible or even better?

*Thank you for a valuable comment. The 15 experiments described in the subsection 3.3.4 TempCNN training and testing aim to test how sensitive the model is to inclusion or exclusion of certain years of data. We have used three different ways to split the data into training and testing sets: 1) a random split of all data points into 80% for training and 20% for testing; 2) three complete years (each from a different SAR platform) were left out for testing, 15 years of data were used for training; 3) training was done using 17 years of data and testing using data from the 2020/2021 ice season which was originally reserved for validation and not used when determining optimal neural network architecture.*

*To clarify this issues, caption of Table 2 has been updated as follows:*

*"Table 2. TempCNN overall classification accuracy for 15 experiments designed to test sensitivity of the network to removing certain years of data from the training set. Runs 1-5 correspond to the 20/80% split of the entire dataset, runs 6-10 were performed by training the network on 15 years of data and testing it on 3 each from a different sensor, runs 10-15 were carried out by training the network on 17 years of data and testing it on 1 year of data that was originally reserved and was not part of the cross validation procedure for determining the best architecture. Subsequently, mean accuracy for each set of 5 runs was calculated, and finally, mean of the three means is shown in the last row of the table."*

4) The effect of speckle should be evaluated by comparing the results without and with speckle filtering. Could the filtering be included in the neural network model? Could e.g. a small neighborhood around each pixel be used instead of single pixel values (applying a 2-D convolution)?

*Thank you for a valuable comment.*

*- Speckle filtering is a standard processing step when working with C-band SAR (e.g., Surdu et al., 2014; Arp et al., 2012; Bartsch et al., 2017; Engram et al., 2018; Duguay and Wang, 2019) and is always performed prior to any environmental analysis. Without speckle filtering the salt and pepper effect is so strong that it would not be possible to extract any useful information.*

*- This work considered pixel-based classification. However, patch-based classification will be the next step in our research.*

5) The reference data are not very good. Is there any way to evaluate the accuracy of the reference data e.g. w.r.t. the existing field measurements?

*Thank you for a valuable comment. To the best of the authors' knowledge, lake (bedfast and floating) ice regimes have not been previously investigated in the Old Crow Flats. As such, field data is very scarce. All the available field measurements have been used in the study. If possible, more ice regime field work will be carried out in the future.*

6) Analysis results: There are a lot of details and figures of selected subregions. What I miss would be a clear conclusion of the analysis indicating by a few numbers of one figure the most essential results of the analysis for the tundra and taiga lakes (in general) and possibly estimated uncertainty estimates. These could be given in a separate shortish subsection

*Thank you for a valuable comment. We appreciate the suggestion. However, we have not done the analysis to compare tundra and taiga ice regimes. The two ecotones are likely responding differently, and a follow-up study would be necessary to analyse and compare lake ice regime dynamics in taiga and tundra areas of the Old Crow Flats.*

Some detailed comments:

L70-74: There are some studies using the separate between static and drifting sea ice based on ice drift or correlation estimated from (SAR) image pairs. Such method has been applied e.g. in Makynen, M.; Karvonen, J.; Cheng, B.; Hiltunen, M.; Eriksson, P.B. Operational Service for Mapping the Baltic Sea Landfast Ice Properties. Remote Sens. 2020, 12, 4032. https://doi.org/10.3390/rs12244032 Also provide a reference to this kind of approach where backscattering is not directly used would complement the manuscript.

*Thank you for a valuable comment. The following sentence has been added to the Introduction section:*

*"Nonetheless, not all bedfast mapping approaches rely directly on the SAR backscatter, for instance, some sea ice studies identified bedfast ice using SAR interferometry (Dammann et al., 2018) and landfast ice using SAR image pairs (Makynen et al., 2020)."*

*Dammann, D.O., Eriksson, L.E.B., Mahoney, A.R., Stevens, C.W., Van der Sanden, J., Eicken, H., Meyer, F.J., and Tweedie, C.E.: Mapping Arctic Bottomfast Sea Ice Using SAR Interferometry, Remote Sens., 10, 720, https://doi.org/10.3390/rs10050720, 2018.*

*Makynen, M., Karvonen, J., Cheng, B.; Hiltunen, M., and Eriksson, P.B.: Operational Service for Mapping the Baltic Sea Landfast Ice Properties, Remote Sens., 12, 4032. https://doi.org/10.3390/rs12244032, 2020.*

Table 1: This indicates that VV mode has been used, except for RS-1. Were there not HH mode data available (e.g. S-1 EW mode data in HH/HV)? Would including cross-polarized channel improve the detection (or has this been studied by anyone)? This is interesting because there exist a lot of HH/HV or VV/VH data acquired by RS-2 and S-1.

*Sentinel-1 EW mode HH/HV imagery was available only for the season of 2016/2017. However, the earliest scene for that season was October 14. Starting the time-series in mid-October could have caused missing the major component of the lake ice lifecycle, namely the initial drop of backscatter as thin ice forms over the water surface. For the remaining years, there was no HH/HV coverage (all the available scenes were located slightly to the north and north-east of the study area).*

*HH and VV polarizations are commonly used for bedfast lake ice mapping (e.g., Engram et al., 2018). To the best of the author's knowledge, cross-polarized SAR imagery is not used for this task. There is a study that performs a classification of decaying ice and open water using HH and HV channels (Geldsetzer et al., 2010).*

*In addition, early C-band SAR data is available only as co-polarized (e.g., ERS1/2). As such, the authors were interested in developing a method that would be able to extend back and use the early platforms as well as take advantage of the new C-band SAR platforms.*

*Engram, M., Arp, C. D., Jones, B. M., Ajadi, O. A., and Meyer, F. J.: Analyzing floating and bedfast lake ice regimes across 580 Arctic Alaska using 25 years of space-borne SAR imagery, Remote Sens. Environ., 209, 660–676, 2018.*

*Geldsetzer, T., Sanden, J., and Brisco, B.: Monitoring lake ice during spring melt using RADARSAT-2 SAR, Can. J. Remote Sens., 36(sup2):S391-S400, 2010.*

L155: Give reference to Lee filter used (there exist variants of Lee filter). On what is the 7x7 window size based?

*Thank you for a valuable comment.*

*Speckle filtering is part of the Radiometric Terrain Correction process carried out by the Alaska Satellite Facility for Sentinel-1. They use Enhanced Lee filter (reference has been added: Banerjee et al., 2021) to remove speckle while preserving edges. When applied, the filter is set to a dampening factor of 1, with a box size of 7x7 pixels and 180 looks. The speckle filter size for the other two sensors was adjusted based on the spatial resolution to cover a similar area as the 7x7 window used for Sentinel-1. Please, refer to the table below.*

| Platform | Pixel size (m) | Filter size (pixels) | Area covered by each filter ($m^2$) |
|---|---|---|---|
| Sentinel 1 | 30 x 30 | 7 x 7 | 44,100 |
| ERS1/2 | 12.5 x 12.5 | 16 x 16 | 40,000 |
| RADARSAT 1 | 50 x 50 | 4 x 4 | 40,000 |

*Banerjee, S., Sinha Chaudhuri, S., Mehra, R., Misra, A.: A Survey on Lee Filter and Its Improved Variants, in: Advances in Smart Communication Technology and Information Processing, edited by Banerjee, S., Mandal, J.K., Lecture Notes in Networks and Systems, vol 165. Springer, Singapore, 2021 https://doi.org/10.1007/978-981-15-9433-5_36*

Figure 2: It would be more clear to show e.g. average and deviation for the classes, now the many curves are shadowing each other.

*Thank you for a valuable comment. Following the referee's suggestion, the means and standard deviation corridors have been plotted instead. Please, see the modified graphs below:*

[Figure]

L254: 330m, how was this elevation threshold selected. Give some reasoning for this selection.

*The threshold was selected by attempting to define an area that would encompass all of the wetland with the majority of lakes, but exclude the surrounding mountainous areas.*

L257: Give the number(s) of field measurements here.

*Thank you for a valuable comment.The manuscript has been modified as follows:*

*"Apart from statistical classification based on the test set, accuracy of the ice regime maps was assessed using a set of 51 field observations."*

*In addition, we are providing a map that demonstrates the distribution of field measurements for the referee's reference.*

[Figure]

L282: (Date1*10)+Date2, You probably mean class(date1)*10 + class(date2) or something similar?

*Thank you for a valuable comment.The formula has been replaced with:*

*("class on date 1" \*10) + "class on date 2"*

L285-286: experimentally defined threshold, be more specific with this. A simple way to define a threshold statistically (experimentally) would be the Bayesian approach based on class distributions. Was this approach used or how was the threshold defined experimentally?

*Thank you for a valuable comment.The threshold was identified by trial and error. We have tried multiple backscatter values moving by 0.5 dB. The selected threshold (-16.5 dB) captured the lake boundaries the best. The manuscript has been modified as follows:*

*"The lake mask was created using an early October 2020 scene and a threshold of -16.5 dB identified through the process of trial and error."*

L291: pyManKendall, give a reference to this python package.

*Thank you for a valuable comment. A reference for the python package has been added:*

*"Hussain et al., (2019). pyMannKendall: a python package for non parametric Mann Kendall family of trend tests. Journal of Open Source Software, 4(39), 1556, https://doi.org/10.21105/joss.01556"*

L305: Is linear interpolation really feasible. To get evidence, You could compare linear interpolation of the periods with data to get error estimates of the linear interpolation.

*Thank you for a valuable comment. The authors are applying linear interpolation following the lead of Pelletier et al., 2019 who propose the TempCNN method for land cover classification using optical imagery. The above-mentioned study*

*indicates that upon investigation they came to a conclusion that more complex interpolation methods have little influence on classification accuracy. In our work linear interpolation does work as is shown by the accuracy assessment. However, following referee's suggestion we will definitely investigate more complex interpolation techniques in the future research.*

Table 2: Please, give formulas /explanations of the accuracy measures (in the text).

*Thank you for a valuable comment. The text of the manuscript has been modified as follows:*

*"As has been mentioned in Sect. 3, the TempCNN model classification accuracy was evaluated through 15 experiments. Table 2 presents the overall accuracy of the 15 experiments calculated as total number of correctly classified time-series in the training set divided by the total size of the training set and multiplied by 100%."*

Fig 7: Better distinguishable colors could be used for different classes. Now the blue tones are difficult to distinguish in a printed version.

*Thank you for a valuable comment. Colours have been updated, please see below:*

[Figure]

Fig.8: Are these time series really informative? Maybe only ice thickness and snow depth for certain day(s) could be shown, now it seems these figures include too much information.

*Thank you for a valuable comment. However, authors would like to keep both ice thickness and snow depth for the full year. We believe that having the maximum value, as well as the seasonal trajectory along with the value on March 13 creates helpful context for the ice regime analysis, comparison of different years, and might be useful for the reader.*

*We wanted to illustrate to the reader the interannual variability and evolution of the snow and ice thickness during the ice growing season relative to the last date of the time-series for which the classification was performed.*

L534: SWOT, can You give any reference to this?

*Thank you for a valuable comment. SWOT (Surface Water and Ocean Topography) mission is due for launch in 1 month and 20 days. The manuscript has been modified as follows to include the link of the mission webpage that contains all the details:*

*"Future analyses of the bedfast/floating ice regime of the OCF will also benefit from data of the upcoming Surface Water and Ocean Topography (SWOT) mission (due for launch in November 2022; https://swot.jpl.nasa.gov/), which will allow for higher accuracy mapping of water level of lakes than is currently possible from current radar altimetry missions."*

---

## Referee Report (RR1)

Dear TC editor and authors of the revised manuscript egusphere-2022-388,

The manuscript has been improved significantly from the previous version and the reviewer comments have been responded and mostly taken into account.

It would be useful to include the information related to using different polarizations and instruments You provided in the response to the reviewer(s) in the manuscript. Also the figure included in the response could be included in the manuscript.

After this minor revision and finalizing the layout in co-operation with the TC editor/typesetter the manuscript will be ready for publication.

Thank You.

---

## Author Response (AR2)

**Point by Point Reply for the manuscript titled: "Bedfast and Floating Ice Dynamics of Thermokarst Lakes Using a Temporal Deep Learning Mapping Approach: Case Study of the Old Crow Flats, Yukon, Canada."**

Maria Shaposhnikova, Claude Duguay, and Pascale Roy-Léveillée

*We would like to thank the referees and the Editor for valuable comments which have substantially helped improve the clarity and quality of the manuscript and stimulated interesting and constructive discussion. The major changes to the manuscript are as follows:*

1. *A brief comparison to a state-of-the art thresholding algorithm with the TempCNN has been carried out and added to the manuscript.*
2. *The advantages of the two approaches (threshold-based and TempCNN) have been emphasized and clearly documented in both the comparison subsection of the Results and Discussion, as well as the Conclusion section.*
3. *Most responses from the previous point-by-point response have been incorporated into the manuscript.*
4. *The manuscript has been revised and edited and all of the referees' questions/comments have been addressed below.*

Referee #1

Dear representatives of the TC editorial board and authors of the revised manuscript egusphere-2022-388,

Thank You for Your response to the reviewer comments. Now many of the reviewer comments have been responded, but the manuscript has not been changed much. Therefore, I still propose a major revision. Including most of the information and references given in the responses to reviewers will already improve the manuscript significantly. A typical reader will very likely not check the responses to reviewer comments, even though they were available, but wants to see all the necessary information in the paper.

*Thank you for a valuable comment.*

*The manuscript has been revised to incorporate the majority of the information provided in the previous point-by-point response. Please, refer to the marked-up version of the manuscript which identifies the modified portions.*

Major comments:

1) There are many explanations, additional information and references in the responses to the reviewers. However, the manuscript has not been changed much. I recommend the authors to include most of the useful information and references given in the responses in the manuscript.

This would significantly improve the manuscript and this is easy to implement as the text has already been written.

*Thank you for a valuable comment.*

*The manuscript has been revised to incorporate the majority of the information provided in the previous point-by-point response. Please, refer to the marked-up version of the manuscript which identifies the modified portions.*

2) Even though the authors emphasize that this is a proof of concept study, it is necessary to make some kind of comparison to some existing method or at least clearly indicate the advantages of the proposed method including reasoning to the advantages. A comparison to another method would be very informative, the comparison could also include comparison of the properties of the algorithms in addition

to the classification accuracy. It does not make sense if the method does not provide any improvement compared to the earlier methods. Even in a proof of concept paper some kind of reference is needed even though the comparison would not be very thorough.

*Thank you for a valuable comment.*

*Based on the request of both referees a comparison to a state-of-the-art thresholding algorithm designed by Duguay and Wang (2019b) has been carried out. Consequently, the Data and Methods, Results and Discussion, and Conclusion sections of the manuscript have been modified to include the comparison. Please, see the details below (lines of the updated manuscript: 317-337; 487-510; 592-603):*

*"3. Data and Methods*

[revised manuscript text omitted]

3) More evidence of the performance for different instruments and polarizations would be needed. Would it be possible to provide classification accuracies for different instruments (polarizations) separately and compare the results with each other and possible performance measures published earlier. Now it is just mentioned that "the authors believe..." in the response to reviewers. I don't think this is very scientific, a more concrete evidence would be needed. If there are some earlier studies with classification accuracies for similar classification problems then the accuracies could be compared directly to give even some idea of the performance. Of course it would be better to apply some of already published algorithms to the same data but if this will require too much resources, at least some kind of comparison would improve the manuscript significantly.

*Thank you for a valuable comment. A comparison to one of the state-of-the-art thresholding algorithms (Duguay and Wang, 2019b) has been carried out and is described in response to the previous comment, as well as included in the manuscript.*

*At this point, it is not possible to provide separate accuracies for different instruments and polarization due to the way that the dataset has been split and used for training and testing. However, using different instruments and polarizations within the same classification algorithms has been done by multiple other researchers. For instance, Duguay and Wang (2019b) have developed a thresholding algorithm for Sentinel-1 and have demonstrated comparability of VV and HH polarized C-band SAR imagery for the purpose of classifying lake ice regimes (Please, see the graph below where New Threshold function shown as solid red line corresponds to the equation used for the comparison). A study by Engram et al., (2018) proposed an interactive threshold classification method to analyze floating and bedfast lake ice regimes across Arctic Alaska using 25-year time-series (1992-2016) of C-band SAR images from different platforms with both HH and VV polarizations. Engram et al. (2018) obtained an overall accuracy of 93% using an interactive threshold-based algorithm with ERS1/2, RADARSAT-2, Envisat, and S1 SAR imagery (including HH and VV polarizations) evaluated over seven lake-rich regions in Arctic Alaska.*

[Figure]

*Duguay, C.R. and J. Wang, 2019b. Arctic-wide ground-fast lake ice mapping with Sentinel-1. ESA Living Planet Symposium, Milan, Italy, 13-17 May.*

*Engram, M., Arp, C. D., Jones, B. M., Ajadi, O. A., and Meyer, F. J.: Analyzing floating and bedfast lake ice regimes across Arctic Alaska using 25 years of space-borne SAR imagery, Remote Sens. Environ., 209, 660–676, 2018.*

Responses to my comments:

Lee filter size: it has not been reasoned why 7x7 filter size was used for S-1, I guess ASF has some kind of reasoning for the size (or number of looks) used in the filtering? How is it better than e.g. 5x5 or 9x9? For the other instruments the filter sizes have been selected to cover approximately the same area, how many looks these correspond?

Also include all this information in the manuscript. There is also a problem with the even size of the filter (4x4 and 16x16) because their center falls between pixels, for this reason an odd number as a size of a filter is recommended, as it has a center pixel.

*Thank you for a valuable comment. ASF does not provide any reasoning for the speckle filter size. However, it does appear that a 7x7 kernel size effectively removes the speckle without loosing valuable information for a given pixel size. The reviewer is absolutely right about the odd filter size. After looking into the issue, we have realized that in fact the filter sizes used during the processing were as follows:*

*S1: 7x7 (area of 44,100 $m^2$)*

*ERS1/2: 17x17 (area of 45,156 $m^2$)*

*R1: 5x5 (area of 62,500 $m^2$)*

*The manuscript has been updated as follows (lines of the updated manuscript:170-173):*

*"Therefore, to match the RTC S1 products filtered using a 7x7 Lee Filter (the filter kernel covers approximately 44,100 $m^2$) with a dampening factor of 1 and 180 looks, ERS1/2 and R1 were speckle filtered using a 17x17 (45,156 $m^2$) and a 5x5 (62,500 $m^2$) Lee Filter, respectively. Adjusting the filter size allowed to account for the pixel size differences."*

The experimentally defined threshold of -16.5 dB: I think it would be better to say "experimentally" than by "trial and error" in the manuscript. Also include the explanation of the method in the manuscript, it is now only in the response to my comments. Also indicate which data were used to experimentally define the threshold (training data set?).

*Thank you for a valuable comment. The manuscript has been updated as follows (lines of the updated manuscript:332-335):*

*"Next, it was necessary to apply a lake mask. Extraction of lakes is challenging in a wetland environment. As such, for simplicity, a single lake mask was created using an October 3, 2020 scene and a threshold of -16.5 dB identified experimentally by changing the threshold value in increments of 0.5dB, until lake boundaries where accurately captured."*

Linear interpolation: You say that more complex interpolation has a little influence on classification. How about even more simple interpolation i.e. nearest neighbor interpolation then? There is no evidence that nature is linear, often changes in nature are quite sudden and fast. Please, include the text of Your response and the reference in the manuscript.

*Thank you for a valuable comment. The temporal deep learning classification method proposed in this work for lake ice regime classification from SAR closely follows the method described in Pelletier et al., 2019, which used linear interpolation to fill the temporal gaps. In future work, we will definitely consider exploring other methods of interpolation, such as nearest neighbor as has been suggested by the referee.*

*The manuscript has been updated as follows (lines of the updated manuscript:204-214):*

*"Resampling to a daily frequency and linear interpolation were applied to compensate for the temporal irregularity of the data ensuring that each of the backscatter time-series for each year of data had the same length (161 values) and gearing it for the deep learning classification (Pelletier et al., 2019; Valero et al., 2016). Although the lake ice lifecycle is non-linear, previous studies have shown that more complex*

*interpolation methods have little influence on classification accuracy (Pelletier et al., 2019, Valero et al., 2016). Linear interpolation was performed utilizing python programming language and the tools of pandas module. Interpolation was performed individually on every time-series (backscatter value of each pixel traced through time). As a result, we obtained SAR image stacks consisting of 161 full coverage scenes, which were subsequently input into the TempCNN to perform classification. In addition to the proper SAR processing and speckle filtering, further quality control was implemented by filling any missing or Not a Number (NaN) values, especially common for ERS1/2 and scene fringes, as part of the temporal interpolation process. The final labeled time-series consisted of 161 time steps (i.e., one time step per day) covering the time period between October 4 and March 13."*

Check all the selected parameters and threshold and give reasoning to their values, also indicate on which data set the parameter selections are based or give references to publications where similar selections have been used.

*Thank you for a valuable comment.*

*The selection of parameters closely follows Pelletier et al., 2019, as has been indicated in the manuscript. The number of convolutional units was selected through a cross-validation procedure using the labelled dataset created as part of this work with the inclusion of 2020/2021 season imagery, reserved for final testing.*

Referee #2

The manuscript has been improved significantly since the first submission, some of the technical issues have been addressed and a number of ambiguities have been clarified. I think that the method presented here has the potential to provide a reference for the lake ice research.

However, I still worry about the novelty of this manuscript. Deep learning method has been widely used in the Earth Science and usually performs better than previous methods. In the original and current versions of manuscript, the comparison to the previous methods is still missing. I understand that the main purpose of this study is just to present a new Deeping learning method for lake ice research. However, it is difficulty to determinate whether the presented method significantly outperform the existing methods and therefore does not provide a substantial added value. The authors admitted that the thresholding approach is indeed very useful and definitely wins over the proposed approach due to its simplicity. Therefore, was this method performed because it could be or because it should be? The latter needs to have demonstrated scientific value. The authors should provide more quantified evidence rather than theory.

*Thank you for a valuable comment. Based on the request of both reviewers a comparison to the state-of-the-art thresholding algorithm designed by Duguay and Wang (2019b) has been carried out. Consequently, the Data and Methods, Results and Discussion, and Conclusion sections of the manuscript have been modified to include the comparison. Please, see the details below (lines of the updated manuscript: 317-337; 487-510; 592-603):*

*"3. Data and Methods*

[revised manuscript text omitted]

Some specific comments:

1) As also pointed out by another reviewer, the input data come from different instruments and they even have different polarizations, the effect of this should be analyzed. Usually, it may be more difficulty to construct a Deeping learning method by using multiple sources of satellite data. However, the accuracy of classification result is still quite high, this may imply that the lake classification by using SAR backscatter is not very complexed. In author's response, the authors provided some counterarguments for this, they thought 'One of the strengths of a deep learning approach is the ability of the network to be trained on different types of data, learn all of its different aspects, and then be able to recognize and classify them correctly.'. However, the obtention of the accurate classification result is not the only aim, the potential mechanism is also worth to explore.

*Thank you for a valuable comment. The reviewer is right, the classification problem is indeed not very complex. However, the proposed approach does resolve multiple problems not tackled by the existing state-of-the-art lake ice regime mapping approaches. The proposed temporal deep learning approach is able to recognize 3 classes: bedfast ice, floating ice, and land, a property not offered by any other method. SAR signatures of land and floating ice become very similar towards end of the season and are not easily separated without applying a temporal approached, proposed in this work. The ability to work*

*with 3 classes, alleviates the need for a lake mask, which is a significant advantage in an environment where lake boundaries are constantly changing. In addition, a method that analysis temporal evolution rather than a single scene is more robust as instead of relying on a single value, it makes a classification decision based on a seasonal backscatter evolution.*

*Moreover, the contribution of the manuscript is not only in proposing a new method of lake ice regime classification, but also in creating and analyzing a time series of lake ice regime maps for Old Crow Flats, Yukon, Canada, - a wetland of international significance - which has not been done by other researchers and has the potential to benefit numerous researchers carrying out other research activities for this study area.*

*In terms of polarization differences, it has bee shown by other approaches that VV and HH polarized C-band imagery is comparable for the purposes of lake ice mapping. For instance, the thresholding approach proposed by Duguay and Wang, 2019b and used by us for comparison (please, see above), is suitable for both HH and VV polarised Sentinel-1 as is demonstrated in the graph below.*

[Figure]

*Duguay, C.R. and J. Wang, 2019. Arctic-wide ground-fast lake ice mapping with Sentinel-1. ESA Living Planet Symposium, Milan, Italy, 13-17 May.*

*In addition, Engram et al. (2018) obtained an overall accuracy of 93% using an interactive threshold-based algorithm with ERS1/2, RADARSAT-2, Envisat, and S1 SAR imagery (including HH and VV polarizations) evaluated over seven lake-rich regions in Arctic Alaska.*

2) The authors used ice thickness data to evaluate the lake ice classification results: 'where ice thickness is equal to the lake depth, lake ice regime is bedfast, whereas in areas where ice thickness is less than the lake depth, lake ice regime is floating.'. What are the precisions of these two datasets, is the ice thickness exactly equal to the lake depth?

*Thank you for a valuable comment. The precision of the field lake depth/ice depth measurements is 1- 2 cm. The ice thickness was determined to be equal to lake depth where ice was determined to have frozen to bed as indicated in our previous sentence.*

---

## Author Response (AR3)

**Point by Point Reply for the manuscript titled: "Bedfast and Floating Ice Dynamics of Thermokarst Lakes Using a Temporal Deep Learning Mapping Approach: Case Study of the Old Crow Flats, Yukon, Canada."**

Maria Shaposhnikova, Claude Duguay, and Pascale Roy-Léveillée

*We would like to thank the referees and the Editor for a very fruitful review process which substantially helped improve the clarity and quality of the manuscript. The minor changes to the manuscript are as follows:*

1. *A more extended discussion of the use of different polarizations and instruments has been added to the Data and Methods section.*
2. *The figure that demonstrates comparability of HH and VV backscatter for floating and bedfast ice has been included in the manuscript.*
3. *The scientific value of the study has been highlighted in the Abstract, Introduction, and Conclusion sections of the manuscript.*
4. *The process of using ice thickness field data to evaluate lake ice thickness classification results has been explained in further detail in the Data and Methods section of the manuscript.*

Referee #1

Dear authors of the revised manuscript egusphere-2022-388,

The manuscript has been improved significantly from the previous version and the reviewer comments have been responded and mostly taken into account.

*Thank you for your comments and a fruitful review process.*

It would be useful to include the information related to using different polarizations and instruments You provided in the response to the reviewer(s) in the manuscript. Also the figure included in the response could be included in the manuscript.

After this minor revision and finalizing the layout in co-operation with the TC editor/typesetter the manuscript will be ready for publication.

Thank You.

*Thank you for a valuable comment. The discussion related to the use of different polarizations and instruments has been added to the Data and Methods section (subsection 3.1. SAR imagery) and the figure has been added to the subsection 3.6 Comparison to thresholding.*

*The manuscript has been updated as follows:*

Under sub-section 3.1:

*"Although using different SAR instruments and polarizations within the same classification algorithm presents its challenges, previous research has shown that such combination is suitable for the mapping of bedfast and floating ice regimes. For instance, Duguay and Wang (2019b) have developed a thresholding algorithm for Sentinel-1 adjusted for incidence angle and have demonstrated comparability of VV and HH polarized C-band SAR imagery for the purpose of classifying lake ice regimes. Engram et al. (2018) also proposed an interactive threshold classification method to analyze floating and bedfast lake ice regimes across Arctic Alaska using 25-year time-series (1992-2016) of C-band SAR images from different platforms (ERS1/2, RADARSAT-2, Envisat, and S1) with both HH and VV polarizations."*

Under sub-section 3.6:

*"In order to benchmark the proposed method against commonly used techniques of lake ice regime classification, it was compared to one of the most recent variations of the thresholding approach designed by Duguay and Wang (2019b) and applicable to S1 data acquired at HH and VV polarization as is illustrated in Fig.6 below."*

[Figure]

**Figure 6.** Relationship between HH and VV polarized backscatter and projected local incidence angle of floating and bedfast lake ice. The "New Threshold function" represents the threshold function proposed by Duguay and Wang (2019b) for lake ice classification and used in this work for the purpose of comparison. The Figure is adopted from Duguay and Wang, 2019b.

Referee #2

The manuscript has been improved a lot and I am overall happy with revision. The paper can be accepted for publication with some edits identified below.

*Thank you for your comments and a fruitful review process.*

1. The scientific value of this study should be emphasized more.

*Thank you for a valuable comment. The scientific value of this work has been highlighted in the Abstract, Introduction, and the Conclusion sections of the manuscript as follows:*

Under Abstract:

*"The proposed lake ice regime mapping approach allowed to assess the combined impacts of warming, drainage, and changing precipitation patterns on transitions between bedfast and floating ice regimes, which is crucial to understanding evolving permafrost dynamics beneath shallow lakes and drained basins in thermokarst lowlands such as OCF."*

Under Introduction:

*"Documenting transitions between bedfast and floating ice regimes in relation to climatic trends is crucial to understanding permafrost dynamics beneath shallow water in thermokarst plains such as OCF, with potential implications for methane emissions and the regional carbon balance."*

Under Conclusion:

*"This study provides a landscape-level perspective on the combined impacts of climatic warming, interannual variations in precipitation, and accelerated lake drainages on the extent of bedfast and floating ice in the OCF. The unexpected overall increase in the fraction of bedfast ice and rapid transition of floating to bedfast ice following catastrophic lake drainage will inform ongoing analyses of permafrost distribution, recovery, and sustainability in drained lake basins.*

*This work provides a strong baseline for future thermokarst lake-ice dynamics analysis, a topic of circumpolar relevance as thermokarst lowlands cover approximately 20% of the northern permafrost regions (Jones et al., 2002) and contain globally significant stores of soil organic carbon (Olefeldt et al., 2015). Documenting transitions between bedfast and floating ice is crucial to understanding permafrost dynamics beneath shallow lakes and drained lake basins, with potential impacts on methane ebullition and the regional carbon balance."*

2. How to use the ice thickness data to evaluate lake ice classification results? This processing progress should be explained in more details.

*Thank you for a valuable comment. The process of using ice thickness data to evaluate lake ice classification results has been further detailed in the Data and Methods section (subsection 3.5 Accuracy assessment) of the manuscript as follows:*

Under sub-section 3.5:

*"Bathymetric measurements of July 2000 were matched with the corresponding TempCNN predicted classes for March 2000 based on geolocation and analyzed in the context of CLIMo simulated ice thickness for the same year. Specifically, if the depth of the data point (based on the bathymetric measurement) was shallower than the CLIMo simulated ice thickness for the corresponding vegetation type (taiga, tundra, or mixed – using Turner et al., 2014 OCF land cover classification) and the label created by TempCNN was "bedfast ice", the point was considered to be classified correctly. Analogously, if the depth of the data point was greater than the CLIMo simulated ice thickness and the TempCNN label was "floating ice", the point was considered to be classified correctly. Ice regime observations made in early April of 2009 and 2021 were also matched with the TempCNN classification output. In this case, both ice thickness and lake depth measurements were available. As such, if the lake depth was equal to the ice thickness, the point was considered to be bedfast, while if the lake depth exceeded the ice thickness measurement, the point was considered to be floating. The precision of the utilized field lake depth and ice depth measurements was 1- 2 cm."*